# Designing Affine-Invariant Neural Networks for Photometric Corruption Robustness and Generalization

**Mounir MESSAOUDI**[1*]**, Quentin RAPILLY**[1*]**, Sébastien HERBRETEAU**[2]**,**
**Anaïs BADOUAL**[1] **and Charles KERVRANN**[1]
[1] Inria Center at University of Rennes, France, SAIRPICO Team
U1339 INSERM, Institut Curie, Chemical Biology of Cancer Unit, Paris, France
{mounir.messaoudi,quentin.rapilly,anais.badoual,charles.kervrann}@inria.fr
[2] Univ Rennes, Ensai, CNRS, CREST—UMR 9194, F-35000 Rennes, France
[*] corresponds to equal contribution.

## Abstract

Standard Convolutional Neural Networks are notoriously sensitive to photometric variations, a critical flaw that data augmentation only partially mitigates without offering formal guarantees. We introduce the *Scale-Equivariant Shift-Invariant* (*SEqSI*) model, a novel architecture that achieves intensity scale equivariance and intensity shift invariance by design, enabling full invariance to global intensity affine transformations with appropriate post-processing. By strategically prepending a single shift-invariant layer to a scale-equivariant backbone, *SEqSI* provides these formal guarantees while remaining fully compatible with common components like ReLU. We benchmark *SEqSI* against *Standard*, *Scale-Equivariant* (*SEq*), and *Affine-Equivariant* (*AffEq*) models on 2D and 3D image-classification and object-localization tasks. Our experiments demonstrate that *SEqSI* architectural properties provide certified robustness to affine intensity transformations and enhances generalization across non-affine corruptions and domain shifts in challenging real-world applications like biological image analysis. This work establishes *SEqSI* as a practical and principled approach for building photometrically robust models without major trade-offs.

## 1 Introduction

Despite their widespread success, standard Convolutional Neural Networks (CNNs) are notoriously sensitive to photometric variations. Semantically irrelevant changes in image contrast or brightness can drastically alter a model predictions (Guan & Liu, 2022; Hendrycks & Dietterich, 2019; Torralba & Efros, 2011), a critical flaw for real-world applications. Data augmentation, mitigates this issue but acts as a brute-force solution, offering no formal guarantee of robustness.

In this work, we argue for a more principled approach: encoding robustness directly into the network architecture. We introduce the *Scale-Equivariant Shift-Invariant* (*SEqSI*) model, a novel architecture whose outputs are **provably invariant to global intensity shifts and equivariant to global intensity scales by construction**. This design provides a robust foundation that can be rendered fully invariant to affine intensity transformations with appropriate post-processing. Crucially, *SEqSI* achieves this guarantee while remaining practical; it is compatible with standard activations and introduces negligible computational overhead. Our main contributions are the following ones:

- We propose a novel architecture (*SEqSI*) which is by design intensity *Scale Equivariant* and intensity *Shift invariant*.

- We address the challenge of achieving invariance from networks whose logits[1] are either affine-equivariant (i.e., scale+shift equivariant) or scale-equivariant and shift-invariant. We

---

[1]We call **logits** the raw output of the network, before any task-specific post-processing.

show that while this is straightforward for argmax-based tasks (e.g., classification), standard pipelines for threshold-based tasks (e.g., object localization) are incompatible with such architectures. We resolve this by introducing a coherent framework pairing output standardization at inference with a novel Z-scored Mean Squared Error (ZMSE) loss.

- We benchmark *SEqSI* against *Standard*, *Scale-Equivariant* (*SEq*) (Mohan et al., 2019), and the more restrictive *Affine-Equivariant* (*AffEq*) (Herbreteau et al., 2023) models. We show that *SEqSI* architectural design provides certified robustness to affine transformations and enhances generalization to non-affine corruptions.

- We demonstrate that, unlike normalization pre-processing which only handles global transformations, the architectural properties of *SEqSI* makes it inherently robust to a range of spatially-varying affine intensity transformations.

- We demonstrate the advantages of *SEqSI* on challenging biological imaging tasks, including macromolecule classification in Cryo-Electron Tomography (Cryo-ET) and object localization in fluorescence microscopy. In these two application fields, where severe and naturally-occurring photometric shifts cause *Standard* models to fail, *SEqSI* architectural guarantees provide robust out-of-distribution generalization, while maintaining high accuracy where baselines collapse (see Fig. 1).

This paper is organized as follows. In Section 2, we give formal definitions. In Section 3, we present related work. In Section 4, we describe the *SEqSI* architecture and its theoretical guarantees. Furthermore, we present an extensive experimental validation, starting with classification benchmarks (Section 5), demonstrating higher out-of-distribution generalization capabilities on challenging cryo-ET data (Section 6), and concluding with its application to object localization (Section 7).

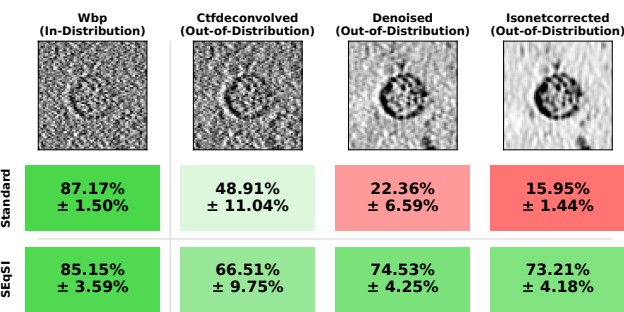

Figure 1: *SEqSI* **is robust to real-world domain shifts for macromolecule classification on Cryo-ET (CZI) data.** Trained on one data domain, Weighted Back-Projection (WBP), and tested on three unseen ones, *SEqSI* generalizes to all unseen domain, while the *Standard* model collapses. Mean accuracy (% ± std over 5 seeds) is color-coded (green: success, red: fail).

## 2 PRELIMINARIES

In this section, we present our framework: the group of photometric transformations of interest and the key properties of equivariance and invariance, which are the mathematical basis for building networks with provable robustness. Let us define a network as a function $f : \mathcal{X} \to \mathcal{Y}$ that maps an input image $x \in \mathcal{X}$ to an output $y \in \mathcal{Y}$, also called logits. The input space $\mathcal{X}$ encompasses 2D or 3D images, which can be single or multi-channel. In the following, we focus on the large family of CNN, and introduce constraints that ensure desired properties. In the paper, we refer to an unconstrained architecture as a *Standard* network, commonly designed as the composition of linear layers (either fully connected or convolutional layers, $x \mapsto Wx + b$, where $W$ is a weight matrix and $b$ is an additive bias) and non-linear activations.

**Photometric transformations.** These corruptions modify the intensities values of an image without altering its spatial structure. A photometric transformation is **global** if the same operator is applied to every pixel. This operator may be be non-linear, such as the gamma correction ($x \mapsto x^\gamma$). For a spatially-varying (**local**) transformations, the intensity variation depends on pixel coordinates (e.g., non-uniform illumination). As shown by Hendrycks & Dietterich (2019), these variations can significantly degrade CNN performance.

Our work focuses on the group of **global affine intensity transformations**, which apply a uniform scaling and shifting operation to every pixel, modeling changes in contrast and brightness. This group, denoted $\mathcal{T}_{\text{aff}}$, is composed of scaling ($\mathcal{T}_{\text{scale}}$) and shifting ($\mathcal{T}_{\text{shift}}$) operations:

$$\mathcal{T}_{\text{aff}} = \{T_{\lambda,\mu} : x \mapsto \lambda x + \mu \mid \lambda \in \mathbb{R}_+^*, \mu \in \mathbb{R}\}, \tag{1}$$

where $\mathcal{T}_{\text{scale}} = \{a_\lambda : x \mapsto \lambda x \mid \lambda \in \mathbb{R}_+^*\}$ and $\mathcal{T}_{\text{shift}} = \{b_\mu : x \mapsto x + \mu \mid \mu \in \mathbb{R}\}$. The condition $\lambda > 0$ ensures that the order of intensities is preserved. While our theoretical guarantees apply

specifically to $\mathcal{T}_{\text{aff}}$, we also empirically evaluate robustness against a broader set of photometric corruptions.

**Equivariance and Invariance.** These notions are fundamental properties for designing robust machine learning models, especially in computer vision. They describe how a function output behaves under input transformations: **equivariance** implies a predictable transformation of the output, whereas **invariance** means that the output remains unchanged. We formally define these concepts bellow.

Let $f : \mathcal{X} \to \mathcal{Y}$ be a function (e.g., a neural network) and $\mathcal{T}$ be a group of transformations defined on both, $\mathcal{X}$ and $\mathcal{Y}$. The function $f$ is:

- $\mathcal{T}$-**equivariant** if, for any $T \in \mathcal{T}$, applying the transformation before or after the function $f$ yields the same results: $\forall T \in \mathcal{T}, f(T(\boldsymbol{x})) = T(f(\boldsymbol{x}))$.

- $\mathcal{T}$-**invariant** if, for any $T \in \mathcal{T}$, the output of the function remains unchanged after applying the transformation to the input: $\forall T \in \mathcal{T}, f(T(\boldsymbol{x})) = f(\boldsymbol{x})$.

## 3 RELATED WORK

**Approximate Robustness.** Common strategies for robustness include data augmentation (Krizhevsky et al., 2012; Hendrycks et al., 2020) and regularization techniques that penalize inconsistencies under transformations in their loss function (Chen et al., 2022; Midtvedt et al., 2022). While effective, these methods only encourage approximate invariance and offer no formal guarantee, which is the main contribution of the paper.

**Robustness by Architectural Design.** A more principled approach, which we adopt, is to design architectures that are equivariant by construction, building on the theory of Group-Equivariant CNNs (Cohen & Welling, 2016). For photometric transformations, various categories of work exist. Some focus on equivariance to color changes (Lengyel et al., 2023; Yang et al., 2024). Others, like our work, address intensity variations with *Scale-Equivariant (SEq)* networks (Mohan et al., 2019) and, more recently, *Affine-Equivariant (AffEq)* networks (Herbreteau et al., 2023). These approaches build on a few key principles. *SEq* networks are constructed by simply removing biases from all layers, making them equivariant to scaling and compatible with standard activations like ReLU. To achieve full affine equivariance, the more restrictive *AffEq* model additionally constrains convolutional weights to sum to one[2] and replaces standard activations with a specialized 'SortPool' layer. 'SortPool' operates on pairs of features $(c_1, c_2)$ by sorting their values at each spatial location: SortPool : $(c_1, c_2) \mapsto (\min(c_1, c_2), \max(c_1, c_2))$. While this guarantees full affine equivariance, it introduces a significant trade-off: constraints on weights and activations make *AffEq* computationally expensive.

## 4 METHOD

Our goal is to design neural networks whose final predictions are provably invariant to global affine photometric transformations ($\mathcal{T}_{\text{aff}}$). Unlike standard approaches, which rely on a fragile input normalization step, we build robustness directly into the model architecture. Our core strategy is a two-step process: first, we design a network whose output logits, $f(\boldsymbol{x})$,

Table 1: **Architectural constraints** for the four model families. These principles can be applied to any standard backbone. Weights constraint are applied channel-wise.

| Model | Activation | Bias | Weights constraint |
|---|---|---|---|
| *Standard* | ReLU | Yes | None |
| *SEq* | ReLU | No | None |
| *AffEq* | SortPool | No | $\sum w = 1$ (for all layers) |
| *SEqSI (ours)* | ReLU | No | $\sum w = 0$ (1st layer only) |

are shift invariant and scale equivariant by design; second, we apply a simple, task-specific postprocessing function to these logits to guarantee a final invariant prediction. This principled approach ensures robustness by construction, as detailed in the following sections.

---

[2]Constraints applied to convolutions apply analogously to linear layers.

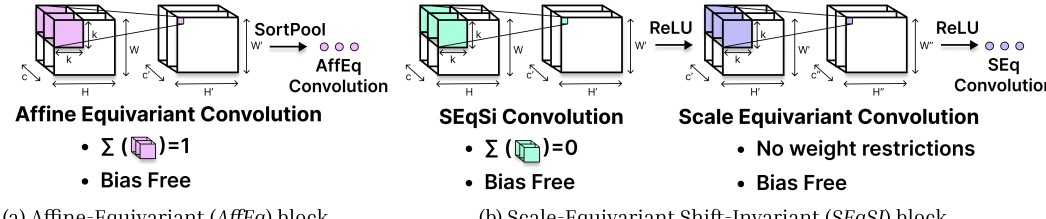

(a) Affine-Equivariant (*AffEq*) block.     (b) Scale-Equivariant Shift-Invariant (*SEqSI*) block.

Figure 2: Comparison of equivariant architectural strategies. **(a)** The *AffEq* model from Herbreteau et al. (2023) enforces strict equivariance in every layer, requiring constrained convolutions and a specialized 'Sort-Pool' activation. **(b)** Our proposed *SEqSI* architecture establishes robustness by prepending a single shift-invariant convolution to a backbone of scale-equivariant layers (bias-free convolution with ReLU). This design simplifies the construction of robust networks with minimal constraints.

## 4.1 PROPOSED ARCHITECTURE

We introduce the ***Scale-Equivariant Shift-Invariant (SEqSI)*** model (Fig. 2), a practical and efficient architecture that achieves robustness through a minimal, strategic design. Its construction follows two principles, following Proposition 2 (see Supp. B.1). First, to ensure scale-equivariance, all layers in the network are made scale-equivariant by using bias-free layers, scale-equivariant activations like ReLU, and standard pooling layers, which are inherently affine-equivariant (see Supp. B.3). Second, to achieve shift-invariance, the initial layer is also made shift-invariant by constraining its weights to sum to zero[3]. As shift-invariance is established by this single initial layer, the rest of the network remains a standard scale-equivariant backbone, while keeping compatible with high-performance components. This composition yields logits that are provably scale-equivariant and shift-invariant ($f(\lambda\boldsymbol{x} + \mu) = \lambda f(\boldsymbol{x})$), providing a robust foundation for achieving full affine invariance with an appropriate task specific post-processing. In Supp. E.3.2 we even show that our approach provide weak[4] invariance guarantees for spatially varying affine transformations such as piecewise constant intensity shifts or intensity shifts varying linearly in space.

The design of *SEqSI* is motivated by the end-task. While tasks like denoising require full *equivariance* at every layer, which forces the use of restrictive architectures like *AffEq*, classification or localization require final *invariance*. By establishing shift-invariance at the very first layer, *SEqSI* frees the following layers of the network from strict shift-equivariance constraints, making our approach more practical. The constraints for all benchmarked models are summarized in Table 1. This avoids the significant computational overhead of *AffEq*, which is over 50% slower and requires 3× more memory. *SEqSI* adds theoretical guarantees at no computational cost and trains 5% faster than the *Standard* baseline with an identical memory footprint (see Supp. C.1). Furthermore, unlike *AffEq*, *SEqSI* is compatible with transfer learning from *Standard* architectures (see Supp. C.10).

## 4.2 TASK SPECIFIC POST-PROCESSING

**Post-processing for Argmax-based tasks.** For any task for which the final prediction is derived from an 'argmax' operation on the logits such as classification or segmentation achieving affine invariance is straightforward. The core principle is that the 'argmax' operation is inherently invariant to any transformation that preserves the order of its inputs. Both models, *SEqSI* and *AffEq* are designed such that an affine transformation on the input image induces an order-preserving transformation on the output logits. Consequently, the 'argmax' of the transformed logits remains unchanged, yielding an invariant prediction. This property proven in Supp. C.3, holds even when a strictly monotonic increasing function (such as 'softmax') is applied before the 'argmax' operation.

**Post-processing for tasks requiring a thresholding.** The classical way to solve object localization consists in converting the logits in a score-map $\hat{\boldsymbol{z}}$, generally via a sigmoid function (to scale results in $[0, 1]$). The network is trained to generate local maxima of this map where objects are located. During inference, the maxima exceeding a chosen threshold are considered as object locations. This thresholding step is incompatible with an equivariant network; affine transformation

---

[3]To our knowledge, scale-invariance of a convolution is impossible to obtain (see Supp. B.2).

[4]We call "weak invariance" result that are not fully invariant, e.g., invariant in image sub-parts only.

on the input image produces transformed logits whose range is shifted and/or scaled, rendering any fixed threshold meaningless and breaking end-to-end invariance.

We address this issue by introducing an alternative way to generate the score-map that becomes invariant to affine transformation when paired with an equivariant network. It consists in standardizing the network output, as follows:

$$\hat{z} = \mathcal{Z}(\boldsymbol{y}) = \frac{\boldsymbol{y} - \mathbb{E}[\boldsymbol{y}]}{\sigma(\boldsymbol{y})}, \tag{2}$$

where $\mathbb{E}[.]$ and $\sigma(.)$ denote the mean and standard deviation (std) operators, respectively. The thresholding of $\hat{z} \geq \gamma$ consists in considering only the voxels whose score is higher than the mean by $\gamma$ times the std of the entire map. In Supp. E.1, we provide the invariance proof and further considerations of this thresholding strategy.

The usual training strategy dedicated to object localization, based on the Binary Cross Entropy (BCE) loss function, requires input standing in the range [0,1]. Hence, it is incompatible with our new score map generation. Accordingly, we introduce a new loss function, named Z-scored Mean Squared Error (ZMSE):

$$\mathcal{L}(\hat{z}, \boldsymbol{z}) = \mathrm{MSE}\Big(\hat{z}, \mathcal{Z}(\boldsymbol{z})\Big) = \mathrm{MSE}\Big(\mathcal{Z}(\boldsymbol{y}), \mathcal{Z}(\boldsymbol{z})\Big), \tag{3}$$

where $\boldsymbol{z}$ is the ground-truth (GT) score map. The GT is usually defined by assigning 0 to the background (outside the objects), and values decreasing linearly from 1 at the object's barycenter to 0 at its boundaries. The ZMSE forces the network to learn the relative spatial distribution of score rather than the absolute values, and do not impose the distributions to lie in the same range, as we are only interested in saving the order and locating maxima. This strategy can be adapted to other tasks requiring a thresholding such as binary segmentation.

---

All experiments presented in this paper are repeated with 5 different seeds to ensure statistical significance, resulting in 5 independently trained model instances.

---

# 5 EXPERIMENT 1: IMAGE CLASSIFICATION

In this section, we demonstrate the robustness of *SEqSI* to photometric corruption for image classification.

## 5.1 EXPERIMENTAL SETUP

**Architectures.** We benchmark four model families summarized in Table 1, based on a ResNet-20 backbone (He et al., 2015), presented in Supp. C.1: *Standard*, *SEq*, *AffEq*, and our proposed *SEqSI* models. For the *SEqSI* and *AffEq* models, an invariant prediction is obtained by applying a 'softmax' followed by an 'argmax' operation to the logits, as detailed in Section 4.2.

**Dataset and Training.** All models are trained on CIFAR-10 (Krizhevsky et al., 2009). We select the best checkpoint based on validation accuracy and report the mean performance across the runs (see Supp. Table 10 for full results). To isolate architectural contributions to robustness, our default training pipeline uses only geometric data augmentation (e.g., random flips and crops), deliberately excluding any photometric augmentations. This ensures that robustness to photometric corruptions is a direct result of architectural design, not training data (see details in Supp. C.1).

Table 2: **Summary of empirical invariance tests** in Supp. Table 7. A model is marked as invariant (✓) if it achieves 0% prediction error for all transformations within a class (Shift, Scale, Affine).

| Model | Shift | Scale | Affine |
|---|---|---|---|
| *Standard* | ✗ | ✗ | ✗ |
| *SEq* | ✗ | ✓ | ✗ |
| *SEqSI* | ✓ | ✓ | ✓ |
| *AffEq* | ✓ | ✓ | ✓ |

## 5.2 RESULTS AND DISCUSSION

**Certified Invariance.** We begin by empirically validating the theoretical properties of the models by measuring their **prediction invariance error** the percentage of predictions on the CIFAR-10

Table 3: **Robustness to corruptions on the CIFAR-10 test set.** Test accuracy (%) of four architectures under various photometric corruptions. Models are grouped by the training augmentation strategy used: none (Ø), affine (**Aff.**), non-affine (**NAff.**), or all combined (**All**). Each row corresponds to a different perturbation applied at evaluation. Within each training strategy (column group), the best result is highlighted in gray. The overall best accuracy for each perturbation is in **bold**. Full results with standard deviations are given in Supp. Table 10.

| Corruption / Model | Train Aug. = Ø | | | | Train Aug. = Aff | | | | Train Aug. = NAff | | | | Train Aug. = All | | | |
|---|---|---|---|---|---|---|---|---|---|---|---|---|---|---|---|---|
| | Stand. | SEq | SEqSI | AffEq | Stand. | SEq | SEqSI | AffEq | Stand. | SEq | SEqSI | AffEq | Stand. | SEq | SEqSI | AffEq |
| Original | 91.7 | 91.3 | 91.2 | 89.6 | 91.2 | 90.6 | 91.0 | 88.9 | 92.1 | 92.1 | 91.5 | 90.5 | 92.0 | 91.6 | 91.8 | 90.6 |
| **Affine transformations (Aff.)** | | | | | | | | | | | | | | | | |
| Shift | 51.1 | 53.4 | 91.2 | 89.6 | 91.2 | 90.3 | 91.0 | 88.9 | 88.5 | 88.7 | 91.5 | 90.5 | 91.4 | 90.7 | **91.8** | 90.6 |
| Scale ($< 1$) | 68.0 | 91.3 | 91.2 | 89.6 | 90.2 | 90.6 | 91.0 | 88.9 | 88.8 | 92.1 | 91.5 | 90.5 | 90.6 | 91.6 | 91.8 | 90.6 |
| Scale ($> 1$) | 91.0 | 91.3 | 91.2 | 89.6 | 91.1 | 90.6 | 91.0 | 88.9 | 91.6 | 92.1 | 91.5 | 90.5 | 92.0 | 91.6 | 91.8 | 90.6 |
| Affine | 64.1 | 65.1 | 91.2 | 89.6 | 90.7 | 89.6 | 91.0 | 88.9 | 87.0 | 88.1 | 91.5 | 90.5 | 90.9 | 90.1 | **91.8** | 90.5 |
| **Non-Affine transformations (NAff.)** | | | | | | | | | | | | | | | | |
| Shift saturated | 72.0 | 73.4 | 79.0 | 76.7 | 79.3 | 79.0 | 78.4 | 76.1 | **87.3** | 86.9 | 86.4 | 85.0 | 87.1 | 86.1 | 86.4 | 84.4 |
| Scale ($> 1$) saturated | 78.1 | 78.3 | 77.6 | 76.0 | 78.0 | 77.7 | 76.9 | 75.7 | **88.0** | 87.5 | 87.1 | 85.6 | 87.9 | 86.7 | 86.8 | 85.0 |
| Affine saturated | 59.9 | 59.6 | 72.3 | 70.9 | 72.6 | 73.3 | 71.7 | 70.6 | **85.0** | 84.8 | 84.1 | 82.8 | 84.9 | 84.0 | 83.7 | 81.9 |
| Spatially-varying Affine | 31.8 | 38.2 | 72.5 | 63.6 | **90.2** | 88.3 | 73.0 | 66.6 | 88.2 | 87.0 | 89.6 | 88.6 | 89.9 | 89.3 | 89.4 | 88.1 |
| Contrast Inversion | 9.6 | 15.4 | 56.5 | 52.9 | 61.3 | 56.2 | 52.9 | 55.0 | 90.2 | **90.3** | 87.6 | 86.3 | 89.2 | 87.8 | 86.5 | 84.6 |
| Noise (low) | 90.8 | 90.6 | 90.1 | 88.5 | 90.0 | 89.8 | 90.0 | 87.8 | 91.6 | **91.7** | 91.0 | 90.0 | 91.5 | 91.2 | 91.3 | 89.8 |
| Noise (high) | 19.0 | 22.4 | 15.2 | 20.3 | 15.5 | 19.8 | 16.0 | 19.2 | 79.6 | **80.0** | 78.7 | 76.6 | 78.3 | 78.3 | 77.9 | 75.2 |
| Gamma (darken) | 74.3 | 76.4 | 79.9 | 76.4 | 82.5 | 79.5 | 79.9 | 76.2 | **89.6** | 88.8 | 88.3 | 86.6 | 89.5 | 87.8 | 88.0 | 86.0 |
| Gamma (lighten) | 85.9 | 85.3 | 90.3 | 88.6 | 90.6 | 89.9 | 90.0 | 88.3 | 91.8 | **91.9** | 91.1 | 90.2 | 91.7 | 91.3 | 91.5 | 90.2 |

Figure 3: **Photometric corruptions used for model robustness evaluation and data augmentation**. The figure displays a reference image (a) and its alteration by various types of corruptions. The distribution plot above each represents the reference pixel distribution (blue) and the perturbed pixel distribution (red). Each sub-figure is labeled for reference: (all transformations, except (h), are global: a unique value is randomly picked in the corresponding distribution) additive shift with $\mu \in [-2.0, 2.0]$ (b) and a saturated version with $\mu \in [-0.7, 0.7]$ (c); scaling with $\lambda \in [0.0, 1.0]$ (d), $\lambda \in [1.0, 4.0]$ (e), and a saturated version with $\lambda \in [1.0, 3.0]$ (f); affine transformation with $\lambda \in [0.0, 4.0]$, $\mu \in [-2.0, 2.0]$ (g); a spatially-varying affine transformation with $\lambda(u, v) \in [0.1, 0.5]$ and $\mu(u, v) \in [-1, 1]$, where u,v are pixel coordinates (h); contrast inversion which is a spatially-varying scale transformation with $\lambda(u, v) \in [-1.0, -0.2]$ (i); additive Gaussian noise $T(x) = x + n$ where $n \sim \mathcal{N}(0, \sigma^2)$ with low intensity ($\sigma \in [0.0, 0.03]$) (j) and high intensity ($\sigma \in [0.15, 0.25]$) (k); non-linear transformations via gamma correction $T(\mathbf{x}) = \mathbf{x}^{\gamma}$, showing image lightening ($\gamma \in [0.2, 1.0]$) (l) and darkening ($\gamma \in [1.0, 5.0]$) (m) ; and affine saturated with $\lambda \in [0.7, 1.3]$, $\mu \in [-0.3, 0.3]$ (n);

test set that change when a global affine intensity transformation is applied (protocol in Supp. C.4). As summarized in Table 2 and detailed in Supp. Table 7, both *SEqSI* and *AffEq* achieve perfect (0% error) affine invariance, empirically confirming their design. In contrast, the *Standard* model is not invariant, and the *SEq* model is only invariant to scaling, highlighting the effectiveness of the architectural constraints.

**Architectural Robustness to Affine and Non-Affine Photometric Corruptions.** To investigate the impact of architectural design, we train all models without photometric augmentation (Table 3, col. Ø) and evaluate their performance against the wide spectrum of corruptions shown in Fig. 3. *SEqSI* and *AffEq* demonstrate their advantage: they maintain high accuracy on all affine corruptions, unlike *Standard* and *SEq* models whose performance degrades significantly. Notably, **this architectural benefit extends to unseen non-affine corruptions, where *SEqSI* outperforms other models on 6 out of 9 transformations.** This suggests that built-in affine invariance fosters a more generalized form of robustness, even for unseen corruptions beyond the theoretical guarantees of the model.

**Interaction with Affine and Non-Affine Data-Augmentation.**    We next investigate the interplay between architectural priors and data-driven robustness, by training all models with three distinct augmentation strategies: **Aff.** (non-saturated affine), **NAff.** (non-affine and saturated affine), and **All** (the complete set of 13 perturbations). To ensure a fair comparison, the transformations used for augmentation share the same parameters as those used for evaluation (Fig. 3). A detailed description of these augmentation strategies is provided in Supp. C.2. The results are presented in Table 3.

The key finding is that architectural priors and data augmentation are complementary. In this controlled setting, where training augmentations perfectly match the evaluation corruptions, the unconstrained *Standard* model is expected to perform better. Nevertheless, the results show that *SEqSI*, despite its architectural constraints, still remains competitive. When trained with the **All** strategy, its accuracy on non-affine corruptions is very close to the *Standard*. This is a significant result because it demonstrates that the *SEqSI* prior is not overly restrictive and does not compromise the capability of the model to learn by using data augmentation. Unlike the strongly constrained *AffEq* model, which systematically underperforms *SEqSI* model, showing a clear advantage of our architecture.

**Scalability to Complex Datasets.**    Beyond CIFAR-10, we conducted further experiments on the more challenging Oxford-IIIT Pets and Stanford Cars datasets. The results, detailed in Supp. C.8, confirm that *SEqSI* maintains its certified invariance and achieves competitive performance, demonstrating its applicability to complex, high-resolution tasks. Moreover, *SEqSI* is compatible with transfer learning from *Standard* architectures, a property not provided by *AffEq* (see Supp. C.10).

**Conclusion**    *SEqSI* provides a practical method for certified affine invariance, overcoming the limitations of prior work. Unlike *AffEq*, it achieves this without the high computational cost or performance degradation from overly restrictive constraints. Our experiments show that **SEqSI architectural prior provides robustness to affine intensity transformations and generalizes to unseen non-affine corruptions, even without specific data augmentation.** Crucially, it complements data augmentation, maintaining certified invariance while demonstrating a learning capability comparable to unconstrained models across a wide range of corruptions.

## 6    EXPERIMENT 2: MACROMOLECULE CLASSIFICATION IN CRYO-ET

To demonstrate the practical benefits of our approach, we evaluate its Out-Of-Distribution (OOD) generalization on a challenging real-world task: macromolecule classification in cryo-electron tomography (cryo-ET), which are 3D grayscale images. We use data from the recent Chan Zuckerberg Initiative (CZI) challenge (Harrington et al., 2024). Each biological sample (tomogram) is processed through four different denoising pipelines (Weighted Back-Projection (WBP), Denoised, IsoNet Corrected and CTF Deconvolved), which act as distinct photometric domains. As shown in Fig. 1, Supp. Fig. 6 and Supp. Table 18, these pipelines induce severe shifts in contrast and brightness, creating a challenging generalization problem and a natural testbed for OOD robustness.

### 6.1    EXPERIMENTAL SETUP

We perform a 6-class macromolecule classification task using a 3D ResNet backbone inspired by ResNet-18 for *Standard*, *SEq* and *AffEq* baselines and our *SEqSI* model. Our experimental design directly test OOD capabilities: we train models on a single domain (WBP) and

Table 4: Performance comparison on test set. Models were trained on *WBP* data. Results are reported as mean accuracy (Acc.) $\pm$ std. Best results for each metric are in **bold**.

| Data Type | Standard Acc. (%) | SEq Acc. (%) | SEqSI Acc. (%) | AffEq Acc. (%) |
|---|---|---|---|---|
| WBP (in-distrib.) | 87.17 $\pm$ 1.50 | **87.69$\pm$0.82** | 85.15$\pm$3.59 | 79.26 $\pm$ 4.04 |
| CTF Deconvolved | 48.91 $\pm$ 11.04 | 65.21$\pm$6.08 | 66.51 $\pm$ 9.75 | **79.07 $\pm$ 5.04** |
| Denoised | 22.36 $\pm$ 6.59 | 28.42$\pm$7.68 | **74.53$\pm$4.25** | 61.79 $\pm$ 5.97 |
| IsoNet Corrected | 15.95 $\pm$ 1.44 | 16.67$\pm$0.00 | **73.21$\pm$4.18** | 46.37 $\pm$ 5.98 |

evaluate their performance on the three unseen domains (Denoised, IsoNet Corrected and CTF Deconvolved). For each run, the model with the highest validation accuracy is selected for testing. We use a strict tomogram-level split (5 train, 1 val, 1 test) to prevent data leakage. The complete experimental protocol is detailed in Supp. D.1.

## 6.2 RESULTS AND DISCUSSION

The results in Table 4 and Fig. 1 highlight the critical role of built-in invariance for OOD generalization. On the in-distribution *WBP* data, the *SEq* model achieves the highest accuracy (87.69%), closely followed by the *Standard* model (87.17%) and our *SEqSI* model (85.15%). The *AffEq* model is also competitive. The true advantage of architectural priors becomes evident when generalizing to the unseen domains. On the *Denoised* and *IsoNet Corrected* data, the performance of both the *Standard* and *SEq* models collapses to near-random chance.[5] In contrast, *SEqSI* maintains high accuracy (74.53% and 73.21% respectively), demonstrating remarkable robustness. The *AffEq* model also proves robust, significantly outperforming the non-invariant baselines, though it does not reach the performance of *SEqSI* on these two domains. Interestingly, on the *CTF Deconvolved* domain, *AffEq* is the top performer (79.07%), with *SEqSI* (66.51%) and *SEq* (65.21%) also showing strong, competitive performance, far surpassing the *Standard* model (48.91%).**This provides strong evidence that architectural invariance is a key mechanism for generalizing across real-world photometric variations, a crucial feature for applications like cryo-ET analysis.** While no single invariant model dominates across all domains, *SEqSI* presents the most compelling trade-off, delivering excellent generalization to most OOD conditions while remaining competitive on in-distribution data.

# 7 EXPERIMENT 3: OBJECT LOCALIZATION

The task of object localization consists in estimating the position of gravity center of each object of interest in an image to either count them or study spatial distribution. It is a common preliminary step in instance segmentation methods: each location can be used to position a bounding-box (Zhou et al., 2019; Duan et al., 2019). For biological images, object localization has been largely studied with approaches locating object in an unsupervised manner in 2D (Midtvedt et al., 2022), improving localization performances in case of very dense object distributions (Van De Looverbosch et al., 2025). A lot of segmentation methods designed for fluorescence microscopy (Schmidt et al., 2018; Weigert et al., 2020; Mandal & Uhlmann, 2021; Rapilly et al., 2025) first locate objects before initializing shape models. In microscopy imaging, photometric conditions can vary a lot, depending on microscope settings. Robust methods to photometric corruptions are of particular interest to process image with unseen conditions. While some solution rely on training models on huge datasets (Pachitariu et al., 2025), frugal approaches based on intrinsic robustness need to be investigated, particularly in the case of 3D, where annotations are rare and costly.

## 7.1 EXPERIMENTAL SETUP

**Architectures.** We compare 4 U-Net architectures (Ronneberger et al., 2015): *Standard*, *SEq*, *AffEq* and our *SEqSI*. The baseline networks (*Standard* and *SEq*) are trained using the approach based on sigmoid and BCE loss. Both *AffEq* and *SEqSI* are trained using the ZMSE and the standardized score maps (Eq. equation 2) ensuring theoretical invariance (see Supp. E.2.1 for details).

**Datasets.** We conduct experiments on two datasets: i) Data Science Bowl 2018 (DSB) (Goodman et al., 2018), a dataset of 2D real microscopy images from a Kaggle challenge, and focused on the 497 fluorescence images, each containing dozens of objects; ii) a set of 3D synthetic images representing nuclei in fluorescence microscopy (see details in Supp. E.2.2).

**Metric.** Each method is evaluated via a score integrating between 0 and a maximum distance $D$, an accuracy curve ($d \mapsto acc(d)$). The accuracy for a given $d$ (in pixels) quantifies the amount of objects correctly predicted, meaning located at a distance lower or equal than $d$ to their corresponding GT (see details in Supp. E.2.3, along with choice of $D$ for each set).

**Choice of the threshold value.** For each method, the thresholding value used during the local maxima extraction, either after the sigmoid for classical approach or after standardization for our approach, is obtained by maximizing the score over the validation set.

---

[5]We further show in App. D.2 that even with a min-max normalization pre-processing, the *Standard* model fails to generalize across domains, unlike our architecturally SEqSI models.

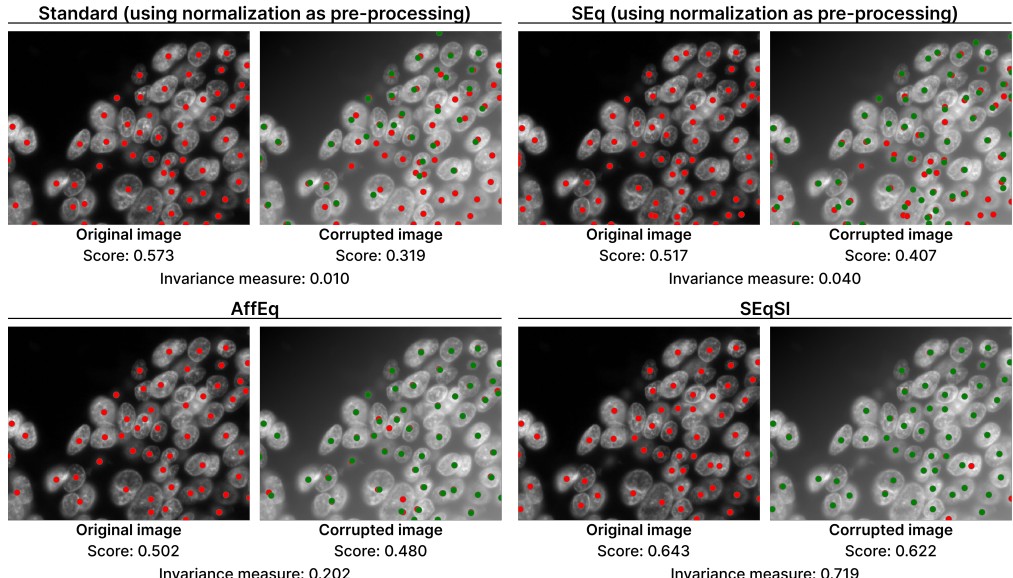

Figure 4: Comparison of behavior for **object localization between a [0,1] normalization pre-processing and invariant approaches on a spatially-varying shift**. The image comes from DSB test set. The positions estimated on the clean (corrupted resp.) image are in red (green, resp.). The shift here increases linearly with the pixel position (from top left to bottom right). The score compares the predictions with the GT positions. The invariance measure compares the predictions on the clean and corrupted image following the method of Supp. E.3.1. Non-visible red points correspond to perfectly overlapping green ones (invariance).

## 7.2 RESULTS AND DISCUSSION

**Certified invariance.** We conduct an empirical validation of the invariance of our approach on the 2D dataset of real images. Methods trained with input images normalized in [0,1] are evaluated for multiple scale, shift or affine transformations. **We obtain similar results than those summarized in Table 2: only *AffEq* and *SEqSI* models with the standardized score-map guarantee affine invariance of locations prediction.** All details on the invariance verification procedure are provided in Supp. E.3.1. Performance results of models trained on the DSB set are illustrated in Fig. 4.

**Normalization pre-processing vs invariant models.** While standard image normalization pre-processing (e.g., normalizing to [0,1] or zero-mean/unit-variance) leads to global photometric affine invariance, it fails for spatially-varying ones. In contrast, our approach maintains robust performance even under spatially-varying corruptions. We compare the *Standard* and *SEq* (with [0,1] normalization pre-processing) and, *AffEq* and *SEqSI* under a spatially-varying photometric shift. This experiment was conducted on the real world DSB set. The result for a shift, varying linearly in space, is illustrated in Fig. 4. Experimental details and theoretical invariance guarantees (focusing on the difference between normalization and the zero-sum layer of *SEqSI*) are provided in App. E.3.2. Though weaker than for global case, the guarantees remain significant.

**Binary segmentation.** Binary segmentation is a subtask of object localization, as score map thresholding is required for both. As segmentation masks are available for the DSB set, we evaluated our method on this set for binary segmentation. Theoretical guarantees hold (see Supp. F).

**Robustness to Affine and Non-Affine Photometric Corruptions.** Similarly to classification, we trained the models using different data-augmentation strategy and evaluated the location estimation on corrupted images (see experimental details in Supp. E.3.3). We see in Table 5 that *SEqSI* is the best model for every augmentation strategy on uncorrupted images and all affine corruptions. *Stand.* trained with Aff. (or All.) get close, but **architectural design performs better than augmentation**. For NAff. corruptions, *SEqSI* is less dominant, but with no augmentation still perform very well on a few corruptions while others' performance drops. With augmentation it is **the best or highly competitive**. It is particularly visible for the NAff. augmentation experiment: Aff. corrup-

Table 5: **Robustness to perturbations on the 3D synthetic test set of fluorescence microscopy.** Test score for $D = 6$ of four architectures under photometric corruptions. Results grouped by the training augmentation strategy : none (∅), affine (Aff.), non-affine (NAff.), or all combined (All). Each row corresponds to a different perturbation applied at evaluation. Within each training strategy (column group), the best result is highlighted in gray. The overall best score for each perturbation is in **bold**. Results with std. given in Supp. Table 23.

| Model / Corruption | Train Aug. = ∅ | | | | Train Aug. = Aff | | | | Train Aug. = NAff | | | | Train Aug. = All | | | |
|---|---|---|---|---|---|---|---|---|---|---|---|---|---|---|---|---|
| | Stand. | SEq | SEqSI | AffEq | Stand. | SEq | SEqSI | AffEq | Stand. | SEq | SEqSI | AffEq | Stand. | SEq | SEqSI | AffEq |
| Original | 0.868 | 0.605 | **0.886** | 0.870 | 0.868 | 0.842 | 0.881 | 0.395 | 0.847 | 0.702 | 0.883 | 0.483 | 0.877 | 0.853 | 0.881 | 0.377 |
| **Affine transformations (Aff.)** | | | | | | | | | | | | | | | | |
| Shift | 0.152 | 0.117 | **0.886** | 0.87 | 0.864 | 0.837 | 0.881 | 0.395 | 0.599 | 0.289 | 0.883 | 0.483 | 0.874 | 0.836 | 0.881 | 0.377 |
| Scale (< 1) | 0.612 | 0.558 | **0.886** | 0.870 | 0.866 | 0.718 | 0.881 | 0.395 | 0.796 | 0.590 | 0.883 | 0.483 | 0.874 | 0.789 | 0.881 | 0.377 |
| Scale (> 1) | 0.491 | 0.506 | **0.886** | 0.870 | 0.864 | 0.835 | 0.881 | 0.395 | 0.835 | 0.723 | 0.883 | 0.483 | 0.875 | 0.855 | 0.881 | 0.377 |
| Affine | 0.149 | 0.093 | **0.886** | 0.870 | 0.843 | 0.799 | 0.881 | 0.395 | 0.631 | 0.240 | 0.883 | 0.483 | 0.837 | 0.803 | 0.881 | 0.377 |
| **Non-Affine transformations (NAff.)** | | | | | | | | | | | | | | | | |
| Shift saturated | 0.074 | 0.066 | 0.069 | 0.359 | 0.457 | 0.298 | 0.322 | 0.264 | 0.508 | 0.385 | 0.564 | 0.234 | **0.567** | 0.405 | 0.555 | 0.244 |
| Scale (> 1) saturated | 0.510 | 0.444 | 0.876 | 0.842 | 0.844 | 0.792 | 0.846 | 0.299 | 0.830 | 0.716 | **0.881** | 0.368 | 0.870 | 0.823 | 0.878 | 0.284 |
| Affine saturated | 0.124 | 0.094 | 0.113 | 0.456 | 0.558 | 0.374 | 0.398 | 0.297 | 0.596 | 0.423 | 0.607 | 0.303 | **0.639** | 0.440 | 0.610 | 0.287 |
| Noise low | 0.633 | 0.505 | **0.885** | 0.736 | 0.868 | 0.841 | 0.882 | 0.385 | 0.846 | 0.703 | 0.884 | 0.471 | 0.878 | 0.853 | 0.881 | 0.372 |
| Noise high | 0.233 | 0.045 | 0.048 | 0.238 | 0.603 | 0.420 | 0.080 | 0.102 | 0.790 | 0.620 | 0.735 | 0.074 | **0.844** | 0.763 | 0.780 | 0.080 |
| Gamma (darken) | 0.535 | 0.462 | 0.424 | 0.704 | 0.838 | 0.676 | 0.669 | 0.388 | 0.798 | 0.581 | 0.864 | 0.438 | **0.873** | 0.752 | 0.777 | 0.376 |
| Gamma (lighten) | 0.512 | 0.390 | 0.883 | 0.805 | 0.865 | 0.835 | 0.876 | 0.382 | 0.843 | 0.718 | **0.884** | 0.471 | 0.876 | 0.850 | 0.879 | 0.364 |
| **Additional experiment on artifacts** | | | | | | | | | | | | | | | | |
| Arti. low | 0.561 | 0.571 | 0.842 | 0.858 | 0.857 | 0.714 | 0.870 | 0.387 | 0.816 | 0.569 | 0.860 | 0.477 | 0.862 | 0.819 | **0.872** | 0.374 |
| Arti. medium | 0.274 | 0.530 | 0.814 | 0.845 | 0.852 | 0.480 | **0.869** | 0.376 | 0.777 | 0.521 | 0.787 | 0.472 | 0.859 | 0.789 | 0.868 | 0.364 |
| Arti. high | 0.064 | 0.0810 | 0.693 | 0.733 | 0.846 | 0.293 | **0.863** | 0.291 | 0.672 | 0.343 | 0.737 | 0.426 | 0.855 | 0.618 | 0.823 | 0.273 |

tions are addressed by design while NAff. corruptions were correctly learned using augmentation, making *SEqSI* **the best method on each corruption but one**. *AffEq*, even though it provides identical invariance guarantee as *SEqSI*, seems to be negatively affected by data-augmentation, reducing its overall performances. Our model, on the other hand, continues to provide excellent results even with augmentation, which allows it to perform very well on non-affine corruptions too.

**Robustness to bright artifacts.** Artifacts are frequent perturbations in fluorescent microscopy, induced by sensor saturation (see supp. Fig. 15). Those perturbations, which **are neither global, nor affine, are nevertheless handled very well by our strategy, even without data-augmentation** (see last rows of Tab. 5). We attribute this to the fact that artifacts, together with the rescaling to the range $[0, 1]$, locally act as a scaling effect away from the artifacts (details in Supp. E.3.4).

**Conclusion.** We provide a method dedicated to object localization, applicable to 2D and 3D images, that guarantees invariance to affine transformation. It performs very well on affine corruptions and, combined with data-augmentation, addresses a wide spectrum of non-affine corruptions, making it a method of choice to perform object location estimation on non-standardized data.

# 8    CONCLUSION

In this work, we introduced the *Scale-Equivariant Shift-Invariant* (*SEqSI*) model, a novel architecture that provides a practical path to photometric robustness. By prepending a single shift-invariant layer to a scale-equivariant backbone with appropriate post-processing, *SEqSI* achieves robustness to affine transformations by design, while remaining compatible with standard components like ReLU and incurring negligible computational cost. Our extensive benchmarks demonstrate that this architectural prior delivers significant practical benefits: *SEqSI* provides certified robustness to affine corruptions and enhances generalization to non-affine transformations. This robustness extends to challenging real-world domain shifts: *SEqSI* maintains high accuracy on OOD cryo-ET data where standard models collapse, and provides robustness to bright artifacts in microscopy object localization. This work establishes that architectural invariance, as implemented in *SEqSI*, is a powerful tool that complements data-driven techniques to build robust and high-performing models.

**Future Work.** Future work could explore SEqSI robustness to composite transformation groups like (e.g., MixUp (Zhang et al., 2018)) or data modalities like videos. Additionally, developing compatible normalization layers to further improve training stability and performance could be a valuable direction (a preliminary work is available in Supp. C.7). Finally, we aim at extending the potential of our object-localization work to segmentation.

CODE AND DATA AVAILABILITY

All our implementations are written in Python and are based on the PyTorch library. The code is available on `https://github.com/MounirMessaoudi/SEqSi`. The datasets CIFAR-10 (Krizhevsky et al., 2009) (Experiment 1), CZI cryo-ET challenge (Harrington et al., 2024) (Experiment 2) and Data Science Bowl 2018 (Goodman et al., 2018) (Experiment 3) are publicly available. Access and processing details are given in the README of the code and Appendix (see Reproducibility Statement 8). For the object localization experiment (Experiment 3), the 3D synthetic dataset will be made available upon publication. A sample of the 3D synthetic dataset is given in Supplementary Materials.

REPRODUCIBILITY STATEMENT

We are committed to ensuring the reproducibility of our work. All implementation details for our models, including architectures, training procedures, and hyperparameter settings, are described throughout the paper and detailed in the Appendix. The full code and instructions to run all experiments are provided in the supplementary material. For clarity, we list the relevant appendix sections for each experiment below.

**Experiment 1 (CIFAR10 Classification):**  Details on network architectures, training hyperparameters, and the CIFAR-10 dataset are in Supp. C.1. The photometric corruptions used for robustness evaluation are described in Supp. C.2. A detailed protocol for invariance verification is available in Supp. C.4.

**Experiment 2 (Macromolecule Classification):**  Details on model architectures, training procedure, and data handling for the CZI Cryo-ET dataset (Harrington et al., 2024) are detailed in Supp. D.1.

**Experiment 3 (Object Localization):**  Architectural details for the 2D and 3D U-Nets are in Supp. E.2.1. Specifics of the synthetic and DSB 2018 datasets are in Supp. E.2.2. The accuracy metric is detailed in Supp. E.2.3. Settings for data augmentation and robustness tests are in Supp. C.2 and E.3.3, with further details on specific perturbations like bright artifacts in Supp. E.3.4.

ACKNOWLEDGEMENTS

This work was support by Agence Nationale de la Recherche (ANR-23-CE45-0012-02). Access to the HPC resources of IDRIS was granted under the allocation 2025-AD011015932R1 made by GENCI.

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

# Appendix

## Table of Contents

## A LARGE LANGUAGE MODELS (LLMS) USAGE

LLMs were used mainly to polish writing.

## B METHODS - APPENDIX

### B.1 MATHEMATICAL FOUNDATIONS

In this section, we denote $\mathcal{T}$ any group of transformation.

**Proposition 1 (Composition of Equivariant Functions)** *The composition of $\mathcal{T}$-equivariant functions is $\mathcal{T}$-equivariant.*

**Proof of Proposition 1** Let $f_1$ and $f_2$ be two functions, equivariant to a class of transformation $\mathcal{T}$. For all $T \in \mathcal{T}$, $f_1 \circ f_2 \circ T = f_1 \circ T \circ f_2 = T \circ f_1 \circ f_2$ as both $f_1$ and $f_2$ commute with any $T$. Therefore, $f_1 \circ f_2$ is also $\mathcal{T}$-equivariant. The extension to the composition of any finite number of equivariant functions is obtained by mathematical induction, proving the proposition.

**Proposition 2 (Composition for Invariance)** *The composition of a $\mathcal{T}$-equivariant function followed by a $\mathcal{T}$-invariant function and then any function (with or without invariance or equivariance properties) is $\mathcal{T}$-invariant.*

**Proof of Proposition 2** Let $f_1$ be equivariant to $\mathcal{T}$ and $f_2$ invariant to $\mathcal{T}$. Let $f_3$ be a function with no particular invariant or equivariant properties.

$$\forall T \in \mathcal{T}, \ f_3 \circ f_2 \circ f_1 \circ T = f_3 \circ f_2 \circ T \circ f_1 \text{ by equivariance of } f_1,$$
$$= f_3 \circ f_2 \circ f_1 \text{ by invariance of } f_2.$$

Therefore, $f_3 \circ f_2 \circ f_1$ is invariant to $\mathcal{T}$.

In our case, we built the SEqSI network by positionning the *shift-invariant* layer at the very beginning (i.e. $f_1 = I_d$ the identity function) without the use of any equivariant layer before.

### B.2 PROOF OF CONVOLUTION AND LINEAR LAYERS EQUIVARIANCE AND INVARIANCE PROPERTIES

According to Herbreteau et al. (2023), let $f$ be a linear or convolutional layer defined by its weights $\boldsymbol{W}$ and bias $\boldsymbol{b}$. The action of the layer on an input vector $\boldsymbol{x}$ is $f(\boldsymbol{x}) = \boldsymbol{W}\boldsymbol{x} + \boldsymbol{b}$. For a convolutional layer, the operation is a convolution $f(\boldsymbol{x}) = \boldsymbol{W} * \boldsymbol{x} + \boldsymbol{b}$, but the reasoning is still the same. An affine transformation is defined as $T_{\lambda,\mu}(\boldsymbol{x}) = \lambda\boldsymbol{x} + \mu\boldsymbol{1}$, where $\boldsymbol{1}$ is a vector of ones of appropriate dimension.

**1. Scale-Equivariance ($T_{\lambda,0}$)** A layer $f$ is scale-equivariant if $f(T_{\lambda,0}(\boldsymbol{x})) = T_{\lambda,0}(f(\boldsymbol{x}))$ for any $\lambda \in \mathbb{R}_+^*$.

- The left-hand side is: $f(T_{\lambda,0}(\boldsymbol{x})) = f(\lambda\boldsymbol{x}) = \boldsymbol{W}(\lambda\boldsymbol{x}) + \boldsymbol{b} = \lambda(\boldsymbol{W}\boldsymbol{x}) + \boldsymbol{b}$.
- The right-hand side is: $T_{\lambda,0}(f(\boldsymbol{x})) = \lambda f(\boldsymbol{x}) = \lambda(\boldsymbol{W}\boldsymbol{x} + \boldsymbol{b}) = \lambda(\boldsymbol{W}\boldsymbol{x}) + \lambda\boldsymbol{b}$.

For the equality to hold for any $\lambda$, we must have $\boldsymbol{b} = \lambda\boldsymbol{b}$. Since this must be true for any $\lambda \neq 1$, the only solution is $\boldsymbol{b} = \boldsymbol{0}$. Thus, a layer is scale-equivariant if and only if it has no bias.

**2. Shift-Equivariance ($T_{1,\mu}$)** A layer $f$ is shift-equivariant if $f(T_{1,\mu}(\boldsymbol{x})) = T_{1,\mu}(f(\boldsymbol{x}))$ for any $\mu \in \mathbb{R}$. Note that the shift value $\mu$ is preserved.

- The left-hand side is: $f(T_{1,\mu}(\boldsymbol{x})) = f(\boldsymbol{x} + \mu\boldsymbol{1}) = \boldsymbol{W}(\boldsymbol{x} + \mu\boldsymbol{1}) + \boldsymbol{b} = \boldsymbol{W}\boldsymbol{x} + \mu(\boldsymbol{W}\boldsymbol{1}) + \boldsymbol{b}$.
- The right-hand side is: $T_{1,\mu}(f(\boldsymbol{x})) = f(\boldsymbol{x}) + \mu\boldsymbol{1} = (\boldsymbol{W}\boldsymbol{x} + \boldsymbol{b}) + \mu\boldsymbol{1}$.

By identifying the terms, we see that the equality holds if and only if $\mu(\boldsymbol{W}\boldsymbol{1}) = \mu\boldsymbol{1}$. Since this must be true for any $\mu \neq 0$, this implies $\boldsymbol{W}\boldsymbol{1} = \boldsymbol{1}$. This condition means that for each output neuron, the sum of its weights must be 1. Note that the bias term $\boldsymbol{b}$ does not affect this property.

**3. Affine-Equivariance** ($T_{\lambda,\mu}$)   A layer $f$ is affine-equivariant if $f(T_{\lambda,\mu}(\boldsymbol{x})) = T_{\lambda,\mu}(f(\boldsymbol{x}))$ for any $(\lambda, \mu) \in \mathbb{R}_+^* \times \mathbb{R}$. This requires combining the two previous properties. We set $\boldsymbol{b} = \boldsymbol{0}$.

- The left-hand side is: $f(T_{\lambda,\mu}(\boldsymbol{x})) = f(\lambda \boldsymbol{x} + \mu \mathbf{1}) = \boldsymbol{W}(\lambda \boldsymbol{x} + \mu \mathbf{1}) = \lambda(\boldsymbol{W}\boldsymbol{x}) + \mu(\boldsymbol{W}\mathbf{1})$.
- The right-hand side is: $T_{\lambda,\mu}(f(\boldsymbol{x})) = \lambda f(\boldsymbol{x}) + \mu \mathbf{1} = \lambda(\boldsymbol{W}\boldsymbol{x}) + \mu \mathbf{1}$.

The equality holds if and only if $\mu(\boldsymbol{W}\mathbf{1}) = \mu\mathbf{1}$, which implies $\boldsymbol{W}\mathbf{1} = \mathbf{1}$. Therefore, a layer is affine-equivariant if and only if its bias is null and its weights sum to 1 for each output neuron.

**4. Shift-Invariance** ($T_{1,\mu}$)   A layer $f$ is shift-invariant if $f(T_{1,\mu}(\boldsymbol{x})) = f(\boldsymbol{x})$ for any $\mu \in \mathbb{R}$.

- The left-hand side is: $f(T_{1,\mu}(\boldsymbol{x})) = f(\boldsymbol{x} + \mu \mathbf{1}) = \boldsymbol{W}(\boldsymbol{x} + \mu \mathbf{1}) + \boldsymbol{b} = \boldsymbol{W}\boldsymbol{x} + \mu(\boldsymbol{W}\mathbf{1}) + \boldsymbol{b}$.
- The right-hand side is: $f(\boldsymbol{x}) = \boldsymbol{W}\boldsymbol{x} + \boldsymbol{b}$.

The equality holds if and only if $\mu(\boldsymbol{W}\mathbf{1}) = \boldsymbol{0}$. Since this must be true for any $\mu \neq 0$, this implies $\boldsymbol{W}\mathbf{1} = \boldsymbol{0}$. This condition means that for each output neuron, the sum of its weights must be 0. As with shift-equivariance, the bias term $\boldsymbol{b}$ does not affect this property. However, in architectures that are also scale-equivariant (like *SEqSI*), the bias must be null.

**5. Scale Invariance**   We did not identify any way to make a convolution *scale-invariant* while maintaining *shift-invariance*. Applying a log function could transform a scale in a shift: $\log(T_{\lambda,0}(\boldsymbol{x})) = T_{1,\lambda}(\log(\boldsymbol{x}))$ and could be used with a *shift-invariant* convolution. Unfortunately the $\log$ function is not compatible with shift as a negative shift would lead to intput outside log definition domain. Considering first removing the shift using a *shift-invariant* convolution before the log could be imagined but is not compatible with the required positivity of $\log$ input.

**Padding Strategy.**   To preserve invariance/equivariance properties across the entire feature map, the padding strategy must be consistent with the corruption of the input image. Standard zero-padding breaks both shift-equivariance and shift-invariance, as it introduces a fixed value (0) at the borders that does not match according to the input signal. Instead, reflection padding must be used, as it fills the boundaries with transformed versions of the input data, maintaining the integrity of the equivariance/invariance property. Other alternatives such as padding the image with a fixed value that depend of the input image can be designed (e.g., padding with the min of the channel, or its mean).

### B.3 PROOF OF POOLING LAYERS EQUIVARIANCE

Let $f_{pool}$ be a pooling operation (max or average) applied over a set of values $\boldsymbol{X} = \{\boldsymbol{x}_1, \ldots, \boldsymbol{x}_N\}$ within a local window. Let $T_{\lambda,\mu}(\boldsymbol{X}) = \{\lambda\boldsymbol{x}_i + \mu \mid \boldsymbol{x}_i \in \boldsymbol{X}\}$ be the application of an affine transformation with $\lambda \in \mathbb{R}_+^*$ and $\mu \in \mathbb{R}$. The operation is $\mathcal{T}_{\text{aff}}$-equivariant if $f_{pool}(T_{\lambda,\mu}(\boldsymbol{X})) = T_{\lambda,\mu}(f_{pool}(\boldsymbol{X}))$.

**1. Max-Pooling.** The max-pooling operation is $f_{max}(\boldsymbol{X}) = \max\{\boldsymbol{x}_1, \ldots, \boldsymbol{x}_N\}$.

- The left-hand side is: $f_{max}(T_{\lambda,\mu}(\boldsymbol{X})) = \max\{\lambda\boldsymbol{x}_1 + \mu, \ldots, \lambda\boldsymbol{x}_N + \mu\}$.
- Since $\lambda > 0$, the scaling is monotonic, and the addition of $\mu$ is a simple shift. Therefore, the maximum of the transformed values is the transformed maximum of the original values:

$$\max\{\lambda\boldsymbol{x}_i + \mu\}_{i=1}^N = \lambda(\max\{\boldsymbol{x}_i\}_{i=1}^N) + \mu. \tag{4}$$

- The right-hand side is: $T_{\lambda,\mu}(f_{max}(\boldsymbol{X})) = \lambda(\max\{\boldsymbol{x}_i\}_{i=1}^N) + \mu$.

Since both sides are equal, max-pooling is $\mathcal{T}_{\text{aff}}$-equivariant.

**2. Average-Pooling.** The average-pooling operation is $f_{avg}(\boldsymbol{X}) = \frac{1}{N} \sum_{i=1}^N \boldsymbol{x}_i$.

- The left-hand side is: $f_{avg}(T_{\lambda,\mu}(\boldsymbol{X})) = \frac{1}{N} \sum_{i=1}^N (\lambda\boldsymbol{x}_i + \mu)$.
- By the linearity of the summation operator:

$$\frac{1}{N} \sum_{i=1}^N (\lambda\boldsymbol{x}_i + \mu) = \frac{1}{N} \left( \sum_{i=1}^N \lambda\boldsymbol{x}_i + \sum_{i=1}^N \mu \right) = \frac{1}{N} \left( \lambda \sum_{i=1}^N \boldsymbol{x}_i + N\mu \right) = \lambda \left( \frac{1}{N} \sum_{i=1}^N \boldsymbol{x}_i \right) + \mu. \tag{5}$$

- The right-hand side is: $T_{\lambda,\mu}(f_{avg}(\boldsymbol{X})) = \lambda \left( \frac{1}{N} \sum_{i=1}^N \boldsymbol{x}_i \right) + \mu$.

Since both sides are equal, average-pooling is also $\mathcal{T}_{\text{aff}}$-equivariant.

### B.4 PROOF OF MINMAX-NORMALIZATION INVARIANCE TO AFFINE INTENSITY TRANSFORMATION

We provide a formal proof that normalization, such as min-max scaling to $[0, 1]$, is invariant to global affine intensity transformations.

Let $\boldsymbol{x}$ be an input image. An affine transformation $T_{\lambda,\mu} \in \mathcal{T}_{\text{aff}}$ is applied, resulting in a perturbed image $\boldsymbol{x}' = \lambda \boldsymbol{x} + \mu$, with $\lambda \in \mathbb{R}_+^*$ and $\mu \in \mathbb{R}$. The min-max normalization function is defined as:

$$\text{Norm}(\boldsymbol{x}) = \frac{\boldsymbol{x} - \min(\boldsymbol{x})}{\max(\boldsymbol{x}) - \min(\boldsymbol{x})} \tag{6}$$

We aim to prove that applying this normalization to the perturbed image $\boldsymbol{x}'$ yields the same result as applying it to the original image $\boldsymbol{x}$, i.e., $\text{Norm}(\boldsymbol{x}') = \text{Norm}(\boldsymbol{x})$.

First, we determine the min and max of the perturbed image $\boldsymbol{x}'$. Due to the linearity of the transformation and since $\lambda > 0$:

$$\min(\boldsymbol{x}') = \min(\lambda \boldsymbol{x} + \mu) = \lambda \min(\boldsymbol{x}) + \mu \tag{7}$$

$$\max(\boldsymbol{x}') = \max(\lambda \boldsymbol{x} + \mu) = \lambda \max(\boldsymbol{x}) + \mu \tag{8}$$

Now, we apply the normalization to $\boldsymbol{x}'$:

$$\text{Norm}(\boldsymbol{x}') = \frac{\boldsymbol{x}' - \min(\boldsymbol{x}')}{\max(\boldsymbol{x}') - \min(\boldsymbol{x}')} = \frac{(\lambda \boldsymbol{x} + \mu) - (\lambda \min(\boldsymbol{x}) + \mu)}{(\lambda \max(\boldsymbol{x}) + \mu) - (\lambda \min(\boldsymbol{x}) + \mu)} \tag{9}$$

Simplifying the expression by canceling out $\mu$ in both the numerator and the denominator:

$$\text{Norm}(\boldsymbol{x}') = \frac{\lambda \boldsymbol{x} - \lambda \min(\boldsymbol{x})}{\lambda \max(\boldsymbol{x}) - \lambda \min(\boldsymbol{x})} = \frac{\lambda(\boldsymbol{x} - \min(\boldsymbol{x}))}{\lambda(\max(\boldsymbol{x}) - \min(\boldsymbol{x}))} \tag{10}$$

Finally, canceling the scaling factor $\lambda$:

$$\text{Norm}(\boldsymbol{x}') = \frac{\boldsymbol{x} - \min(\boldsymbol{x})}{\max(\boldsymbol{x}) - \min(\boldsymbol{x})} = \text{Norm}(\boldsymbol{x}) \tag{11}$$

This proves that applying min-max normalization as a pre-processing step after a global affine intensity perturbation effectively cancels out the perturbation, making the input to the network invariant. A similar proof holds for z-score normalization.

# C EXPERIMENTS IN CLASSIFICATION : CIFAR-10, OXFORD-IIIT PET AND STANFORD CARS - APPENDIX

## C.1 DETAILS ON MODELS IMPLEMENTATION AND DATA PROCESSING

All classification models are based on a ResNet-20 architecture He et al. (2015), adapted for the 32×32 images of the CIFAR-10 dataset. The modular design allows us to instantiate the four model families—*Standard*, *SEq*, *SEqSI*, and *AffEq*—by applying the specific constraints summarized in Table 1 of the main paper. A key difference lies in the padding strategy as described in Appendix B.2: *Standard* and *SEq* models use zero-padding, whereas ***AffEq* and *SEqSI* models use reflection padding** to preserve their respective equivariance and invariance properties.

**Overall Architecture.** The network follows a ResNet-20 structure adapted for 32×32 images, comprising an initial convolutional layer, three residual stages, and a classification head.

The **initial layer** is a 3×3 convolution mapping the 3-channel input to 64 feature maps while preserving the 32×32 resolution. Its properties are configured for each model family as per Table 1: for instance, it is shift-invariant (weights sum to 0) for *SEqSI* and affine-equivariant (weights sum to 1, bias-free) for *AffEq*.

The network then proceeds through three **residual stages**, each with 3 residual blocks. Stage 1 operates on 64-channel feature maps. Stages 2 and 3 double the channel count to 128 and 256, respectively, while halving the spatial resolution to 16×16 and 8×8 pixels. This downsampling is performed by the first block of each of these stages, which uses a stride of 2 in its first convolutional layer.

The **classification head** processes the final 8×8×256 feature map. A global average pooling layer reduces it to a 256-dimensional vector, which is then projected onto 10 output logits by a fully connected layer. A final 'softmax' function converts logits to class probabilities.

**Residual Block Structure.** Each of the 9 residual blocks consists of a main path and a shortcut connection.

The **main path** contains two 3×3 convolutional layers. Based on the implementation, the data flow is $Conv \rightarrow Norm \rightarrow Activation \rightarrow Dropout \rightarrow Conv \rightarrow Norm \rightarrow Dropout$. The output of this path is then added to the output of th shortcut, and a final activation is applied to the sum. The specific activation ('ReLU' or 'SortPool') depends on the model family (Table 1).

The **shortcut connection** is an identity mapping if input and output dimensions are identical. In downsampling blocks, it employs a 1×1 convolution with a stride of 2 to match the output dimensions of the main path before the element-wise addition.

**Optional Layers for Compatibility Study.** By default, none of the model families include normalization or dropout layers, ensuring a baseline comparison focused purely on the architectural constraints. For the compatibility study presented in Section C.7, these layers are optionally inserted into the residual blocks. Normalization layers (InstanceNorm, BatchNorm) are positioned after each convolutional layer but before the activation function. The Dropout layer is inserted after the activation function.

**Data Preprocessing and Data Augmentation.** To ensure a fair comparison and enlighten the benefits of architectural invariance, all models are trained with a carefully designed data pipeline. A key aspect of this pipeline is the **exclusive use of geometric data augmentation**. We decided to avoid photometric augmentations (e.g., brightness or contrast changes) to prevent models from learning invariant properties. This setup guarantees that the observed robustness to photometric corruptions is a direct result of architectural design, not from the training dataset.

The pipeline begins by scaling input images values from the original [0, 255] range to [0, 1] (a simple division by 255 is applied). During training, we then apply the following geometric augmentations to enhance generalization:

**Random Horizontal Flips:** Each image is flipped horizontally with a probability of 50%.

**Random Cropping:** The 32×32 input images are first padded with 4 pixels on each side to create a 40×40 image. A random 32×32 crop is then extracted from this padded image.

For model selection, we evaluated performance on the validation set without applying any data augmentation; only the scaling to [0, 1] range is applied. This standard procedure ensures a fair comparison, as the best checkpoint for each model is selected based on its performance on the clean, canonical data distribution, prior to the final robustness evaluation on the test set.

**Training Details.** To ensure a fair comparison, all models were trained under identical conditions. We train each model for 500 epochs with a batch size of 128, using the standard Cross-Entropy loss. Optimization was performed with SGD, with a momentum of 0.9 and a weight decay of 5e-4. The learning rate was initialized to 0.1 and followed a cosine annealing schedule. We applied gradient clipping with a maximum norm of 1.0. To ensure robust results, we repeated each experiment with 5 distinct random seeds (5, 7, 42, 137, 181). For each run, we selected the checkpoint with the highest validation accuracy and reported the mean and standard deviation of the test performance between seeds. We use the standard CIFAR-10 splits, reserving 10% (5000 images) of the training set for validation. All experiments were performed on a single NVIDIA A40 GPU. The training and validation curves in Supp. Figure 5, shows that *Standard*, *SEq* and *SEqSI* converge similarly, while *AffEq* takes more time to converge.

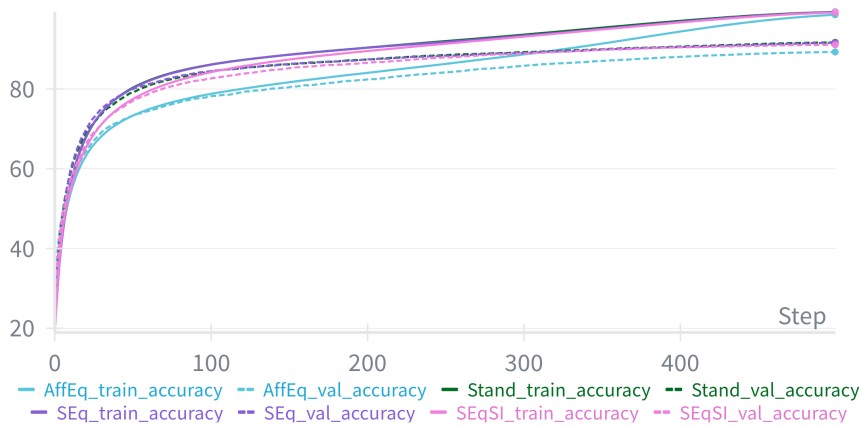

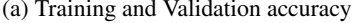

(a) Training and Validation accuracy

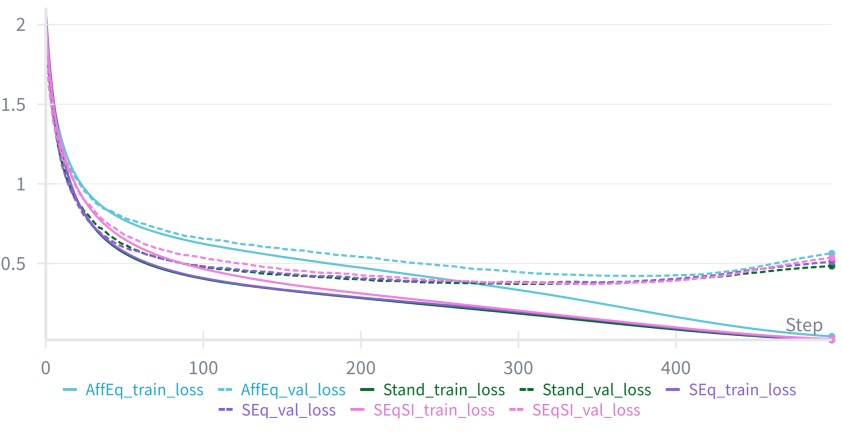

(b) Training and Validation loss

Figure 5: **Training dynamics on CIFAR-10.** Validation accuracy (a) and loss (b) curves for all ResNet-20 variants.

Table 6: Computational costs for each model, measured on an NVIDIA A40 GPU. Metrics are averaged over 495 epochs after a 5-epoch warmup. **Training (s/epoch)**: Time per training epoch on 45,000 images. **Inference (s)**: Total time for the 5,000-image validation set. **Peak Mem. (GB)**: Maximum GPU memory allocated during training. **# Parameters**: Number of trainable parameters.

| Model | Training (s/epoch) | Inference (s) | Peak Mem. (GB) | # Parameters |
|---|---|---|---|---|
| Standard | 11.87 | 0.66 | 0.44 | 4 324 618 |
| SEq | 10.79 | 0.61 | 0.44 | 4 321 472 |
| SEqSI | 11.23 | 0.61 | 0.44 | 4 321 472 |
| AffEq | 18.61 | 0.86 | 1.26 | 4 321 481 |

**Computational Costs.** Supp. Table 6 details the computational costs for each tested model. The results highlight the efficiency of the *SEqSI* model, which operates with a computational footprint nearly identical to the *SEq* and even better than *Standard*. In stark contrast, the *AffEq* model is substantially more resource-intensive, with a training time over 50% longer, an inference time over 30% longer, and a peak memory usage nearly three times higher.

C.2   DETAILS ON PHOTOMETRIC AUGMENTATION AND EVALUATION PERTURBATION

The photometric transformations used for both data augmentation and evaluation are detailed below. Note that the parameters are the same for transformation and augmentation, respectively, at inference and training time. The parameters correspond to those described in Figure 3. To ensure experimental consistency, the same parameters are used for Experiment 3: Object Localization (Section 7).

**Augmentation Pipeline.**   During training, for the 'Aff', 'NAff', and 'All' strategies, we applied photometric augmentation using a 'transforms.RandomChoice' mechanism. For each image, one transformation was randomly selected from the corresponding set and applied. This set also includes an identity transformation, meaning there is a chance an image is not photometrically augmented. The parameters for each transformation (e.g., the specific shift value) are sampled uniformly from the specified range for each image independently. All photometric augmentations are applied after geometric augmentations (random crops and flips).

**Evaluation Protocol.**   For the robustness evaluation (Table 3), each transformation type was tested independently. To measure robustness to a given corruption (e.g., 'Shift'), we applied the corresponding transformation to each image in the test set. Crucially, and similarly to the augmentation pipeline, the specific parameter for the transformation (e.g., the value of $\mu$ for a shift) was sampled independently and uniformly from the specified range for each image. This ensures that the evaluation assesses performance across the entire distribution of the corruption, rather than just a single fixed value.

**Transformation Groups.**   The transformations are categorized into two main groups:

**Affine Transformations (Aff):** This group consists of 4 non-saturated global affine transformations. They form the 'Aff' augmentation set.

- **Global Additive Shift:** $T(\boldsymbol{x}) = \boldsymbol{x} + \mu$, with $\mu \sim \mathcal{U}[-2.0, 2.0]$.
- **Global Scaling (Compression):** $T(\boldsymbol{x}) = \lambda\boldsymbol{x}$, with $\lambda \sim \mathcal{U}[0.0, 1.0]$.
- **Global Scaling (Dilation):** $T(\boldsymbol{x}) = \lambda\boldsymbol{x}$, with $\lambda \sim \mathcal{U}[1.0, 4.0]$.
- **Global Affine:** $T(\boldsymbol{x}) = \lambda\boldsymbol{x} + \mu$, with $\lambda \sim \mathcal{U}[0.0, 4.0]$ and $\mu \sim \mathcal{U}[-2.0, 2.0]$.

**Non-Affine and Saturated Transformations (NAff):** This group includes 9 transformations that are either non-affine, spatially-varying, or saturated. A *saturated* transformation means the output pixel values are clipped to the $[0, 1]$ range. This group forms the 'NAff' augmentation set.

- **Saturated Additive Shift:** $T(\boldsymbol{x}) = \mathrm{clip}(\boldsymbol{x} + \mu, 0, 1)$, with $\mu \sim \mathcal{U}[-0.7, 0.7]$.
- **Saturated Scaling:** $T(\boldsymbol{x}) = \mathrm{clip}(\lambda\boldsymbol{x}, 0, 1)$, with $\lambda \sim \mathcal{U}[1.0, 3.0]$.
- **Saturated Affine:** $T(\boldsymbol{x}) = \mathrm{clip}(\lambda\boldsymbol{x} + \mu, 0, 1)$, with $\lambda \sim \mathcal{U}[0.7, 1.3]$ and $\mu \sim \mathcal{U}[-0.3, 0.3]$.
- **Spatially-Varying Affine:** $T(\boldsymbol{x})(u, v) = \lambda(u, v)\boldsymbol{x}(u, v) + \mu(u, v)$, with $\lambda(u, v) \sim \mathcal{U}[0.1, 0.5]$ and $\mu(u, v) \sim \mathcal{U}[-1, 1]$ for each pixel $(u, v)$.
- **Contrast Inversion:** A spatially-varying scaling $T(\boldsymbol{x})(u, v) = \lambda(u, v)\boldsymbol{x}(u, v)$, with $\lambda(u, v) \sim \mathcal{U}[-1.0, -0.2]$ for each pixel $(u, v)$.
- **Additive Gaussian Noise (Low):** $T(\boldsymbol{x}) = \mathrm{clip}(\boldsymbol{x} + n, 0, 1)$, where $n \sim \mathcal{N}(0, \sigma^2)$ and $\sigma \sim \mathcal{U}[0.0, 0.03]$.
- **Additive Gaussian Noise (High):** $T(\boldsymbol{x}) = \mathrm{clip}(\boldsymbol{x} + n, 0, 1)$, where $n \sim \mathcal{N}(0, \sigma^2)$ and $\sigma \sim \mathcal{U}[0.15, 0.25]$.
- **Gamma Correction (Lighten):** $T(\boldsymbol{x}) = \boldsymbol{x}^\gamma$, with $\gamma \sim \mathcal{U}[0.2, 1.0]$.
- **Gamma Correction (Darken):** $T(\boldsymbol{x}) = \boldsymbol{x}^\gamma$, with $\gamma \sim \mathcal{U}[1.0, 5.0]$.

The 'All' augmentation strategy combines both 'Aff' and 'NAff' sets, for a total of 13 transformations.

## C.3 INVARIANCE BY POST-PROCESSING FOR ARGMAX-BASED TASKS

This section proves that for any task where the final prediction is derived from an 'argmax' operation (e.g., classification, semantic segmentation), the prediction is invariant to input affine transformations when using either an *AffEq* or a *SEqSI* model.

Let $z = f(x)$ be the logits produced by a network for an input $x$. When the input is transformed by $T_{\lambda,\mu}(x) = \lambda x + \mu \mathbf{1}$, the logits $z'$ transform differently depending on the architecture:

- For an ***Affine-Equivariant (AffEq)*** network: $z' = f(T_{\lambda,\mu}(x)) = \lambda z + \mu \mathbf{1}$.
- For a ***Scale-Equivariant, shift-invariant (SEqSI)*** network: $z' = f(T_{\lambda,\mu}(x)) = \lambda z$.

Both cases can be unified by considering the general transformation $z' = \lambda z + \mu \mathbf{1}$, where $\mu = 0$ for the *SEqSI* model. The core of the proof is that the 'argmax' operation is insensitive to any strictly monotonic increasing transformation of its input vector.

**1. Invariance of Argmax.** The transformation $z \mapsto z' = \lambda z + \mu \mathbf{1}$ is strictly order-preserving for any $\lambda > 0$. That is, if $z_i > z_j$ for any two elements of $z$, then their transformed values also satisfy $\lambda z_i + \mu > \lambda z_j + \mu$. Consequently, the transformation does not change the index of the maximum element:

$$\arg\max(z') = \arg\max(\lambda z + \mu \mathbf{1}) = \arg\max(z). \tag{12}$$

This shows that applying 'argmax' directly to the logits of either an *AffEq* or *SEqSI* network yields a provably invariant prediction.

**2. Invariance with a Strictly Monotonic Increasing Activation.** Now, consider applying a strictly monotonic increasing function $h$ to the logits before the 'argmax', like 'softmax'. Since $h$ is strictly monotonic increasing, it also preserves the order of its inputs. Therefore, the order of elements in $h(z)$ is the same as in $z$, and the order in $h(z')$ is the same as in $z'$.

$$\arg\max(h(z')) = \arg\max(z') = \arg\max(z) = \arg\max(h(z)). \tag{13}$$

This demonstrates that the composition 'argmax' ∘ 'h' is invariant to affine transformations of the logits, provided $h$ is strictly monotonic increasing. This justifies its use as a final post-processing step for any argmax-based task, guaranteeing that the final prediction is robust to affine photometric transformations of the input image.

**Practical Consideration for Numerical Stability.** In practice, when dealing with very large input transformations, the logits $z$ can reach extreme values, potentially causing numerical overflow in activation functions like 'softmax'. A standard technique to ensure stability is to shift the logits by subtracting their maximum value before applying the activation, i.e., computing $h(z - \max(z))$. This operation does not affect the final prediction, as 'argmax' is invariant to this shift: $\arg\max(h(z - \max(z))) = \arg\max(z - \max(z)) = \arg\max(z)$. This makes the computation robust to large shifts in logit values.

Table 7: **Mean and standard deviation of prediction invariance error** (%, lower is better) over multiple seeds, and different degrees of affine transformation. Format is mean ± std. Best mean result per row in extbfbold. Note that, 'dil.' and 'comp.' stand for dilation and compression, respectively.

| Transf. | Setting | Standard | SEq | SEqSI | AffEq |
|---------|---------|----------|-----|-------|-------|
| **Shift** | $\mu = -2$ (left shift, moderate) | 90.0 ±0.37 | 89.4 ±0.59 | **0** ±0.00 | **0** ±0.00 |
| | $\mu = 0.5$ (right shift, small) | 12.2 ±0.78 | 11.4 ±1.21 | **0** ±0.00 | **0** ±0.00 |
| | $\mu = 2$ (right shift, moderate) | 58.1 ±2.71 | 58.6 ±3.25 | **0** ±0.00 | **0** ±0.00 |
| | $\mu = 10$ (right shift, extreme) | 88.0 ±2.39 | 87.4 ±2.58 | **0** ±0.00 | **0** ±0.00 |
| **Scale** | $\lambda = 0.5$ (comp.) | 8.48 ±0.64 | **0** ±0.00 | **0** ±0.00 | **0** ±0.00 |
| | $\lambda = 3.0$ (dil., moderate) | 4.48 ±0.47 | **0** ±0.00 | **0** ±0.00 | **0** ±0.00 |
| | $\lambda = 255$ (dil., extreme) | 5.93 ±0.70 | **0** ±0.00 | **0** ±0.00 | **0** ±0.00 |
| **Affine** | $\mu = -2, \lambda = 10$ (left shift + strong dil.) | 8.05 ±0.35 | 6.78 ±0.20 | **0** ±0.00 | **0** ±0.00 |
| | $\mu = 5, \lambda = 0.1$ (shift + strong comp.) | 90.2 ±0.26 | 89.5 ±0.59 | **0** ±0.00 | **0** ±0.00 |
| | $\mu = 5, \lambda = 3$ (shift + dilation, moderate) | 49.3 ±2.39 | 49.5 ±3.93 | **0** ±0.00 | **0** ±0.00 |

## C.4 PROTOCOL FOR INVARIANCE VERIFICATION

To empirically verify the theoretical invariance properties of our proposed architectures, we conducted an evaluation on the best performing checkpoints, selected based on validation accuracy after training. This step is crucial to ensure that the assessment of invariance is not confounded by the arbitrary behavior of an untrained model, where initial weight values could themselves influence predictions. For each model, trained with one of our 5 distinct seeds, we applied a set of affine transformations (shift, scale, or both) to the **entire** CIFAR-10 test set. While this evaluation is deterministic, we set the random seed to match the one used for training each model to ensure full reproducibility.

$$\text{Pred. Inv. Err.} = \frac{1}{N} \sum_{i=1}^{N} \mathbb{1} \left( \operatorname{argmax}(f(\mathcal{T}_{\text{aff}}(\mathbf{x}_i))) \neq \operatorname{argmax}(f(\mathbf{x}_i)) \right) \times 100 \qquad (14)$$

where $\mathbb{1}$ is the indicator function and N is the number of images.

We then measured the **prediction invariance error** (see Supp. Equation 14), defined as the percentage of images for which the predicted class changes once the transformation has been applied. To accurately measure perfect invariance (0% error), this test is run in float64 precision, as standard float32 can introduce numerical errors that break theoretical guarantees (see Appendix C.5 for more details). This protocol allowed to enlighten us to isolate and quantify the inherent robustness provided by each architectural design.

In Supp. Table 7, we reported the prediction invariance error for each model family. To test these properties, we applied a range of affine transformations (shift, scale, and their combination) for varying intensities, from moderate to extreme. This approach enabled to quantify how the invariance error changes with the magnitude of the perturbation, highlighting the stable, perfect invariance of the certified models in all conditions.

Table 8: **Prediction invariance error (% as mean $\pm$ std) in float32 precision.** Comparison of "Telescopic" and "Mean Subtraction" implementations. Values of 0.00 are highlighted in gray.

| Transform. | Setting | SEqSI | | AffEq | |
|---|---|---|---|---|---|
| | | Telescopic | Mean Subtr. | Telescopic | Mean Subtr. |
| **Shift** | $\mu = -2$ | 0.01 $\pm$0.00 | 0.00 $\pm$0.00 | 0.04 $\pm$0.01 | 0.05 $\pm$0.02 |
| | $\mu = 0.5$ | 0.00 $\pm$0.00 | 0.00 $\pm$0.00 | 0.03 $\pm$0.01 | 0.02 $\pm$0.01 |
| | $\mu = 2$ | 0.01 $\pm$0.01 | 0.00 $\pm$0.00 | 0.04 $\pm$0.02 | 0.06 $\pm$0.02 |
| | $\mu = 10$ | 0.00 $\pm$0.01 | 0.00 $\pm$0.00 | 0.22 $\pm$0.03 | 0.26 $\pm$0.05 |
| **Scale** | $\lambda = 0.5$ | 0.00 $\pm$0.00 | 0.00 $\pm$0.00 | 0.00 $\pm$0.00 | 0.00 $\pm$0.00 |
| | $\lambda = 3$ | 0.01 $\pm$0.00 | 0.01 $\pm$0.01 | 0.02 $\pm$0.01 | 0.02 $\pm$0.01 |
| | $\lambda = 255$ | 0.01 $\pm$0.01 | 0.00 $\pm$0.01 | 0.02 $\pm$0.01 | 0.02 $\pm$0.01 |
| **Affine** | $\mu = -2, \lambda = 10$ | 0.01 $\pm$0.01 | 0.00 $\pm$0.00 | 0.02 $\pm$0.01 | 0.02 $\pm$0.00 |
| | $\mu = 5, \lambda = 0.1$ | 0.01 $\pm$0.01 | 0.01 $\pm$0.00 | 1.16 $\pm$0.12 | 1.41 $\pm$0.10 |
| | $\mu = 5, \lambda = 3$ | 0.01 $\pm$0.01 | 0.01 $\pm$0.01 | 0.04 $\pm$0.01 | 0.03 $\pm$0.01 |

Table 9: **Prediction invariance error (% as mean $\pm$ std) in float64 precision.** Comparison of "Telescopic" and "Mean Subtraction" implementations. Values of 0.00 are highlighted in gray.

| Transform. | Setting | SEqSI | | AffEq | |
|---|---|---|---|---|---|
| | | Telescopic | Mean Subtr. | Telescopic | Mean Subtr. |
| **Shift** | $\mu = -2$ | 0.00 $\pm$0.00 | 0.00 $\pm$0.00 | 0.00 $\pm$0.00 | 0.00 $\pm$0.00 |
| | $\mu = 0.5$ | 0.00 $\pm$0.00 | 0.00 $\pm$0.00 | 0.00 $\pm$0.00 | 0.00 $\pm$0.00 |
| | $\mu = 2$ | 0.00 $\pm$0.00 | 0.00 $\pm$0.00 | 0.00 $\pm$0.00 | 0.00 $\pm$0.00 |
| | $\mu = 10$ | 0.00 $\pm$0.00 | 0.00 $\pm$0.00 | 0.00 $\pm$0.00 | 0.00 $\pm$0.00 |
| **Scale** | $\lambda = 0.5$ | 0.00 $\pm$0.00 | 0.00 $\pm$0.00 | 0.00 $\pm$0.00 | 0.00 $\pm$0.00 |
| | $\lambda = 3$ | 0.00 $\pm$0.00 | 0.00 $\pm$0.00 | 0.00 $\pm$0.00 | 0.00 $\pm$0.00 |
| | $\lambda = 255$ | 0.00 $\pm$0.00 | 0.00 $\pm$0.00 | 0.00 $\pm$0.00 | 0.00 $\pm$0.00 |
| **Affine** | $\mu = -2, \lambda = 10$ | 0.00 $\pm$0.00 | 0.00 $\pm$0.00 | 0.00 $\pm$0.00 | 0.00 $\pm$0.00 |
| | $\mu = 5, \lambda = 0.1$ | 0.00 $\pm$0.00 | 0.00 $\pm$0.00 | 0.00 $\pm$0.00 | 0.00 $\pm$0.00 |
| | $\mu = 5, \lambda = 3$ | 0.00 $\pm$0.00 | 0.00 $\pm$0.00 | 0.00 $\pm$0.00 | 0.00 $\pm$0.00 |

## C.5 NUMERICAL STABILITY OF WEIGHT CONSTRAINT IMPLEMENTATIONS

The weight constraints for the *AffEq* and *SEqSI* models (respectively, $\sum \boldsymbol{w} = 1$ and $\sum \boldsymbol{w} = 0$) can be enforced in several ways. We compare two common parameterization methods: "Telescopic" and "Mean Subtraction".

$$
\begin{aligned}
&\textit{SEqSI (shift-inv., } \sum \boldsymbol{w}' = 0): &&\begin{cases} \boldsymbol{w}' = \text{roll}(\boldsymbol{w}) - \boldsymbol{w} & \text{(Telescopic)} \\ \boldsymbol{w}' = \boldsymbol{w} - \frac{1}{N}\sum_{i=1}^{N} w_i & \text{(Mean Subtraction)} \end{cases} \\
&\textit{AffEq (shift-equiv., } \sum \boldsymbol{w}' = 1): &&\begin{cases} \boldsymbol{w}' = \text{roll}(\boldsymbol{w}) - \boldsymbol{w} + \frac{1}{N} & \text{(Telescopic)} \\ \boldsymbol{w}' = \boldsymbol{w} - \frac{1}{N}\sum_{i=1}^{N} w_i + \frac{1}{N} & \text{(Mean Subtraction)} \end{cases}
\end{aligned}
\tag{15}
$$

where roll($\boldsymbol{w}$) denotes a circular shift of the vector $\boldsymbol{w}$ by one position (i.e., roll($\boldsymbol{w}$)$_i = w_{i-1}$).

Both the "Telescopic" and "Mean Subtraction" implementations are theoretically invariant. However, we observed that numerical approximations in float32 precision can break these theoretical guarantees. As shown in Supp. Table 8, the key finding is that numerical stability is mainly driven by the architecture (*SEqSI* vs. *AffEq*) rather than the specific parameterization. While both implementations for *SEqSI* are highly stable with only negligible errors, the *AffEq* model is significantly more prone to instability, exhibiting much larger errors regardless of the parameterization. Within the *SEqSI* model, the "Mean Subtraction" variant is marginally more robust. As detailed in Supp. Table 9, perfect invariance was restored for all models and implementations when performing calculations in float64 precision.

The "Mean Subtraction" method was used for all classification-task experiments. Invariance experiments were performed in float64, while performance experiments were performed in float32 for computational reasons. The object location experiments were conducted with the "Telescopic" implementation in float64 precision. We reported here implementation details for reproducibility of results.

## C.6 FULL ROBUSTNESS RESULTS WITH STANDARD DEVIATIONS

In Supp. Table 10, we present the full robustness results, corresponding to the main results in Table 3. To ensure statistical significance, we reported the mean and standard deviation of accuracy over 5 complete training and evaluation runs, each performed with a distinct random seed. The results are split into two parts for readability. Note that all experiments were conducted in float32 precision.

Table 10: **Robustness to evaluation-time perturbations (mean $\pm$ std over 5 seeds).** Test accuracy (%) of four architectures under various photometric corruptions. Models are grouped by the training augmentation strategy used: none ($\emptyset$), affine (**Aff.**), non-affine (**NAff.**), or all combined (**All**). Each row corresponds to a different perturbation applied at evaluation. Within each strategy group, the best result is highlighted in gray. The overall best result for each perturbation is in **bold**.

| Model / Corruption | Train Aug. = $\emptyset$ | | | | Train Aug. = Aff | | | |
|---|---|---|---|---|---|---|---|---|
| | Stand. | SEq | SEqSI | AffEq | Stand. | SEq | SEqSI | AffEq |
| Original | 91.7 ±0.3 | 91.3 ±0.1 | 91.2 ±0.3 | 89.6 ±0.1 | 91.2 ±0.3 | 90.6 ±0.1 | 91.0 ±0.1 | 88.9 ±0.2 |
| **Affine transformations (Aff.)** | | | | | | | | |
| Shift | 51.1 ±1.0 | 53.4 ±1.2 | 91.2 ±0.3 | 89.6 ±0.1 | 91.2 ±0.4 | 90.3 ±0.2 | 91.0 ±0.1 | 88.9 ±0.2 |
| Scale ($< 1$) | 68.0 ±0.8 | 91.3 ±0.1 | 91.2 ±0.3 | 89.6 ±0.1 | 90.2 ±0.5 | 90.6 ±0.1 | 91.0 ±0.1 | 88.9 ±0.2 |
| Scale ($> 1$) | 91.0 ±0.1 | 91.3 ±0.1 | 91.2 ±0.3 | 89.6 ±0.1 | 91.1 ±0.3 | 90.6 ±0.2 | 91.0 ±0.1 | 88.9 ±0.2 |
| Affine | 64.1 ±0.9 | 65.1 ±0.7 | 91.2 ±0.3 | 89.6 ±0.1 | 90.7 ±0.3 | 89.6 ±0.2 | 91.0 ±0.1 | 88.9 ±0.2 |
| **Non-Affine transformations (NAff.)** | | | | | | | | |
| Shift saturated | 72.0 ±0.6 | 73.4 ±0.4 | 79.0 ±0.6 | 76.7 ±0.4 | 79.3 ±0.6 | 79.0 ±0.8 | 78.4 ±0.6 | 76.1 ±0.3 |
| Scale ($> 1$) saturated | 78.1 ±0.5 | 78.3 ±0.6 | 77.6 ±0.5 | 76.0 ±0.5 | 78.0 ±0.7 | 77.7 ±0.7 | 76.9 ±0.5 | 75.7 ±0.3 |
| Affine saturated | 59.9 ±0.8 | 59.6 ±1.1 | 72.3 ±1.0 | 70.9 ±0.5 | 72.6 ±0.9 | 73.3 ±1.0 | 71.7 ±0.7 | 70.6 ±0.3 |
| Spatially-varying Affine | 31.8 ±1.0 | 38.2 ±2.3 | 72.5 ±2.2 | 63.6 ±1.0 | 90.2 ±0.6 | 88.3 ±0.4 | 73.0 ±1.1 | 66.6 ±2.7 |
| Contrast Inversion | 9.6 ±0.7 | 15.4 ±2.0 | 56.5 ±1.6 | 52.9 ±1.0 | 61.3 ±0.8 | 56.2 ±2.2 | 52.9 ±1.3 | 55.0 ±1.5 |
| Noise (low) | 90.8 ±0.2 | 90.6 ±0.2 | 90.1 ±0.6 | 88.5 ±0.2 | 90.0 ±0.3 | 89.8 ±0.2 | 90.0 ±0.1 | 87.8 ±0.2 |
| Noise (high) | 19.0 ±2.4 | 22.4 ±3.5 | 15.2 ±1.0 | 20.3 ±0.6 | 15.5 ±1.1 | 19.8 ±3.4 | 16.0 ±2.2 | 19.2 ±1.9 |
| Gamma (darken) | 74.3 ±1.0 | 76.4 ±0.7 | 79.9 ±0.4 | 76.4 ±0.5 | 82.5 ±0.3 | 79.5 ±1.1 | 79.9 ±0.4 | 76.2 ±0.2 |
| Gamma (lighten) | 85.9 ±0.6 | 85.3 ±0.7 | 90.3 ±0.2 | 88.6 ±0.1 | 90.6 ±0.4 | 89.9 ±0.1 | 90.0 ±0.2 | 88.3 ±0.2 |

| Model / Corruption | Train Aug. = NAff | | | | Train Aug. = All | | | |
|---|---|---|---|---|---|---|---|---|
| | Stand. | SEq | SEqSI | AffEq | Stand. | SEq | SEqSI | AffEq |
| Original | 92.1 ±0.1 | **92.1 ±0.2** | 91.5 ±0.2 | 90.5 ±0.1 | 92.0 ±0.1 | 91.6 ±0.1 | 91.8 ±0.2 | 90.6 ±0.2 |
| **Affine transformations (Aff.)** | | | | | | | | |
| Shift | 88.5 ±0.6 | 88.7 ±0.3 | 91.5 ±0.2 | 90.5 ±0.1 | 91.4 ±0.2 | 90.7 ±0.2 | **91.8 ±0.2** | 90.6 ±0.2 |
| Scale ($< 1$) | 88.8 ±0.4 | 92.1 ±0.2 | 91.5 ±0.2 | 90.5 ±0.1 | 90.6 ±0.3 | 91.6 ±0.1 | 91.8 ±0.2 | 90.6 ±0.2 |
| Scale ($> 1$) | 91.6 ±0.1 | 92.1 ±0.2 | 91.5 ±0.2 | 90.5 ±0.1 | 92.0 ±0.2 | 91.6 ±0.1 | 91.8 ±0.2 | 90.6 ±0.2 |
| Affine | 87.0 ±0.8 | 88.1 ±0.4 | 91.5 ±0.2 | 90.5 ±0.1 | 90.9 ±0.2 | 90.1 ±0.1 | **91.8 ±0.2** | 90.5 ±0.2 |
| **Non-Affine transformations (NAff.)** | | | | | | | | |
| Shift saturated | **87.3 ±0.2** | 86.9 ±0.2 | 86.4 ±0.3 | 85.0 ±0.3 | 87.1 ±0.5 | 86.1 ±0.4 | 86.4 ±0.1 | 84.4 ±0.5 |
| Scale ($> 1$) saturated | **88.0 ±0.1** | 87.5 ±0.3 | 87.1 ±0.3 | 85.6 ±0.3 | 87.9 ±0.2 | 86.7 ±0.1 | 86.8 ±0.2 | 85.0 ±0.3 |
| Affine saturated | **85.0 ±0.2** | 84.8 ±0.3 | 84.1 ±0.3 | 82.8 ±0.3 | 84.9 ±0.3 | 84.0 ±0.3 | 83.7 ±0.3 | 81.9 ±0.5 |
| Spatially-varying Affine | 88.2 ±0.6 | 87.0 ±0.2 | 89.6 ±0.2 | 88.6 ±0.2 | 89.9 ±0.2 | 89.3 ±0.2 | 89.4 ±0.1 | 88.1 ±0.3 |
| Contrast Inversion | 90.2 ±0.2 | **90.3 ±0.2** | 87.6 ±0.2 | 86.3 ±0.4 | 89.2 ±0.4 | 87.8 ±0.2 | 86.5 ±0.1 | 84.6 ±0.4 |
| Noise (low) | 91.6 ±0.1 | **91.7 ±0.2** | 91.0 ±0.2 | 90.0 ±0.2 | 91.5 ±0.2 | 91.2 ±0.1 | 91.3 ±0.4 | 89.8 ±0.2 |
| Noise (high) | 79.6 ±0.3 | **80.0 ±0.6** | 78.7 ±0.7 | 76.6 ±0.5 | 78.3 ±0.4 | 78.3 ±0.4 | 77.9 ±0.5 | 75.2 ±0.9 |
| Gamma (darken) | **89.6 ±0.2** | 88.8 ±0.4 | 88.3 ±0.2 | 86.6 ±0.3 | 89.5 ±0.4 | 87.8 ±0.1 | 88.0 ±0.2 | 86.0 ±0.4 |
| Gamma (lighten) | 91.8 ±0.2 | **91.9 ±0.2** | 91.1 ±0.2 | 90.2 ±0.1 | 91.7 ±0.2 | 91.3 ±0.1 | 91.5 ±0.3 | 90.2 ±0.2 |

## C.7 Compatibility Study with Normalization and Dropout Layers

**Analysis of Compatibility**   We conducted a study to assess the compatibility of our proposed architectures with standard deep learning components that are known to improve performance and training stability: normalization layers and dropout. For this study, we integrated Batch Normalization (BN), Layer Normalization (LN), Instance Normalization (IN), and Dropout (DP) into each of the four model families. All layers were used with their default PyTorch parameters to reflect common practices. We also evaluated a "Mix" strategy, which employs Instance Normalization after the first convolutional block and Batch Normalization in all subsequent blocks, a technique intended to combine the benefits of both. The results, presented in Supp. Tables 11 and 12, are analyzed in the following from the perspectives of both theoretical invariance and empirical performance.

**Impact on Invariance Guarantees.**   The invariance error measurements (Supp. Table 11) reveal how these layers interact with the architectural constraints.

- **SEqSI Model:** A key finding is the robustness of shift-invariance. This property, enforced by the first layer, is perfectly preserved (0% error) across all configurations, including all normalization types and dropout. However, its scale-equivariance is broken by all normalization layers (BN, LN, IN, and Mix).

- **AffEq Model:** The more restrictive *AffEq* model is less compatible. Its affine-equivariance is completely broken by Batch Normalization and partially by Layer Normalization. It only maintains its guarantees with Instance Normalization, which is itself $\epsilon$-affine-invariant (where $\epsilon$ is a value added to the denominator for numerical stability. Default: 1e-5.)

- **Standard Components:** Dropout, being inactive at inference, does not affect the invariance of any model. Instance Normalization, because of its own invariance properties, can provide shift-invariance to the *Standard* and *SEq* models, but at the cost of breaking the scale-equivariance needed for a principled design like *SEqSI*.

This analysis confirms that integrating standard normalization layers breaks the theoretical scale-equivariance of the *SEqSI*. The magnitude of this impact on scale-invariance varies significantly: Batch Normalization severely degrades it (e.g., up to 86.8% error), whereas Layer and Instance Normalization have a much smaller impact, with an error typically below 1% but reaching up to 4.93% in some cases. Crucially, the shift-invariance guarantee remains perfectly intact across all normalization layers. This establishes a practical trade-off, which we now evaluate by analyzing the impact on performance.

**Impact on Performance.**   While normalization layers can break theoretical invariance properties, the performance is still preserved in practical experiments (see Supp. Table 12).

- **Synergy with SEqSI:** Without normalization, the architectural constraints of *SEqSI* result in a clean accuracy of 91.2%, slightly below the *Standard* model accuracy (91.7%). However, this gap is reduced and often reversed when normalization is introduced. *SEqSI* benefits immensely from normalization, with its performance increasing by over 4 percentage points to 95.7% with BN and 95.5% with the Mix strategy. This synergy elevates *SEqSI* to match or exceed the performance of the *Standard* model on clean data, demonstrating its compatibility with modern training recipes.

- **Enhanced Robustness:** The combination of the *SEqSI* prior and normalization layers leads to remarkable empirical robustness, even when theoretical scale-equivariance is partially broken. When paired with normalization, *SEqSI* consistently outperforms its *Standard* counterpart. For instance, the *SEqSI+Mix* variant is superior to *Standard+Mix* on 8 out of 13 corruptions, with particularly large gains on challenging non-affine transformations like "Spatially-varying Affine" (94.9% vs. 89.4%) and "Contrast Inversion" (71.7% vs. 66.7%). This demonstrates that the architectural properties provides a powerful inductive bias for learning robust features, which is further amplified by the stabilized training dynamics from normalization.

- **Practicality over AffEq:** In contrast, the *AffEq* model, while theoretically well-grounded, proves less in practical imaging. Its performance on clean data drops significantly with Dropout (from 89.6% to 81.2%), and it is consistently outperformed by *SEqSI* across all

normalization schemes. This underscores *SEqSI* superior balance of theoretical principles and practical utility.

**Conclusion:** This study demonstrates that while *SEqSI* is compatible with standard normalization layers at the cost of its theoretical scale-equivariance, the practical benefits are substantial. The combination of *SEqSI* architectural properties with normalization layers results in models that not only match the performance of standard networks on clean data, but consistently outperform them in terms of robustness to a wide range of photometric corruptions. This synergy establishes *SEqSI* as a practical, effective, and principled choice for building high-performing and robust networks, making it a valuable tool for real-world applications where data variability is a key challenge.

Table 11: Detailed invariance error (% as mean $\pm$ std over seeds) for different normalization layers. The best result (lowest error) per row is in **bold**. A value of 0 indicates perfect invariance and is highlighted in gray. Ø: no Norm, BN: Batch Norm, LN: Layer Norm, IN: Instance Norm, Mix: Mixed Norms, DP: Dropout.

| Transform. | Setting | Ø | | | | DP | | | |
|---|---|---|---|---|---|---|---|---|---|
| | | Stand. | SEq | SEqSI | AffEq | Stand. | SEq | SEqSI | AffEq |
| **Shift** | $\mu = -2$ (left shift, moderate) | 90.0 ±0.37 | 89.4 ±0.59 | **0.00** ±0.00 | **0.00** ±0.00 | 89.3 ±0.57 | 89.1 ±1.70 | **0.00** ±0.00 | **0.00** ±0.00 |
| | $\mu = 0.5$ (right shift, small) | 12.2 ±0.78 | 11.4 ±1.21 | **0.00** ±0.00 | **0.00** ±0.00 | 12.3 ±0.59 | 8.27 ±0.38 | **0.00** ±0.00 | **0.00** ±0.00 |
| | $\mu = 2$ (right shift, moderate) | 58.1 ±2.71 | 58.6 ±3.25 | **0.00** ±0.00 | **0.00** ±0.00 | 55.7 ±2.86 | 51.3 ±1.60 | **0.00** ±0.00 | **0.00** ±0.00 |
| | $\mu = 10$ (right shift, extreme) | 88.0 ±2.39 | 87.4 ±2.58 | **0.00** ±0.00 | **0.00** ±0.00 | 88.9 ±1.29 | 88.8 ±0.94 | **0.00** ±0.00 | **0.00** ±0.00 |
| **Scale** | $\lambda = 0.5$ (compression) | 8.48 ±0.64 | **0.00** ±0.00 | **0.00** ±0.00 | **0.00** ±0.00 | 7.31 ±1.06 | **0.00** ±0.00 | **0.00** ±0.00 | **0.00** ±0.00 |
| | $\lambda = 3.0$ (dilation, moderate) | 4.48 ±0.47 | **0.00** ±0.00 | **0.00** ±0.00 | **0.00** ±0.00 | 10.6 ±0.99 | **0.00** ±0.00 | **0.00** ±0.00 | **0.00** ±0.00 |
| | $\lambda = 255$ (dilation, extreme) | 5.93 ±0.70 | **0.00** ±0.00 | **0.00** ±0.00 | **0.00** ±0.00 | 30.8 ±4.05 | **0.00** ±0.00 | **0.00** ±0.00 | **0.00** ±0.00 |
| **Affine** | $\mu = -2, \lambda = 10$ (left shift + strong dilation) | 8.05 ±0.35 | 6.78 ±0.20 | **0.00** ±0.00 | **0.00** ±0.00 | 19.0 ±1.71 | **0.00** ±0.00 | **0.00** ±0.00 | **0.00** ±0.00 |
| | $\mu = 5, \lambda = 0.1$ (shift + strong compression) | 90.2 ±0.26 | 89.5 ±0.59 | **0.00** ±0.00 | **0.00** ±0.00 | 89.5 ±0.18 | 89.7 ±0.13 | **0.00** ±0.00 | **0.00** ±0.00 |
| | $\mu = 5, \lambda = 3$ (shift + dilation, moderate) | 49.3 ±2.39 | 49.5 ±3.93 | **0.00** ±0.00 | **0.00** ±0.00 | 61.7 ±6.60 | 41.4 ±1.52 | **0.00** ±0.00 | **0.00** ±0.00 |

| Transform. | Setting | BN | | | | LN | | | |
|---|---|---|---|---|---|---|---|---|---|
| | | Stand. | SEq | SEqSI | AffEq | Stand. | SEq | SEqSI | AffEq |
| **Shift** | $\mu = -2$ (left shift, moderate) | 89.7 ±0.83 | 90.2 ±0.71 | **0.00** ±0.00 | 90.0 ±0.06 | 90.9 ±1.42 | 90.0 ±0.22 | **0.00** ±0.00 | **0.00** ±0.00 |
| | $\mu = 0.5$ (right shift, small) | 10.6 ±0.62 | 11.7 ±0.60 | **0.00** ±0.00 | 17.2 ±1.70 | 7.65 ±0.39 | 5.73 ±0.48 | **0.00** ±0.00 | **0.00** ±0.00 |
| | $\mu = 2$ (right shift, moderate) | 89.5 ±1.01 | 90.0 ±0.61 | **0.00** ±0.00 | 90.0 ±0.91 | 27.4 ±3.97 | 17.1 ±1.68 | **0.00** ±0.00 | **0.00** ±0.00 |
| | $\mu = 10$ (right shift, extreme) | 90.4 ±0.62 | 89.9 ±0.05 | **0.00** ±0.00 | 90.0 ±0.05 | 82.2 ±5.81 | 60.8 ±3.56 | **0.00** ±0.00 | **0.00** ±0.00 |
| **Scale** | $\lambda = 0.5$ (compression) | 4.70 ±0.32 | 4.56 ±0.15 | 4.06 ±0.10 | 7.01 ±0.34 | 5.91 ±0.25 | 0.04 ±0.03 | 0.30 ±0.04 | **0.01** ±0.00 |
| | $\lambda = 3.0$ (dilation, moderate) | 52.4 ±7.43 | 56.7 ±6.68 | 16.2 ±1.78 | 75.5 ±3.59 | 4.17 ±0.13 | 0.01 ±0.01 | 0.10 ±0.02 | **0.00** ±0.01 |
| | $\lambda = 255$ (dilation, extreme) | 89.9 ±1.30 | 90.2 ±1.81 | 86.8 ±2.79 | 90.1 ±0.41 | 7.24 ±1.19 | 0.01 ±0.01 | 0.10 ±0.02 | **0.00** ±0.01 |
| **Affine** | $\mu = -2, \lambda = 10$ (left shift + strong dilation) | 89.5 ±1.62 | 88.2 ±2.34 | 83.2 ±1.43 | 90.2 ±0.82 | 6.63 ±1.04 | 8.88 ±0.62 | 0.10 ±0.02 | **0.00** ±0.01 |
| | $\mu = 5, \lambda = 0.1$ (shift + strong compression) | 90.0 ±0.17 | 90.0 ±0.10 | 72.1 ±3.27 | 90.5 ±0.84 | 90.0 ±0.07 | 89.5 ±0.41 | 4.93 ±0.59 | **0.25** ±0.05 |
| | $\mu = 5, \lambda = 3$ (shift + dilation, moderate) | 90.0 ±0.18 | 90.2 ±0.54 | 16.2 ±1.78 | 90.3 ±0.50 | 28.3 ±5.80 | 14.7 ±1.59 | 0.10 ±0.02 | **0.00** ±0.01 |

| Transform. | Setting | IN | | | | Mix | | | |
|---|---|---|---|---|---|---|---|---|---|
| | | Stand. | SEq | SEqSI | AffEq | Stand. | SEq | SEqSI | AffEq |
| **Shift** | $\mu = -2$ (left shift, moderate) | 6.66 ±0.53 | 5.89 ±0.31 | **0.00** ±0.00 | **0.00** ±0.00 | 3.48 ±0.25 | 3.22 ±0.09 | **0.00** ±0.00 | **0.00** ±0.00 |
| | $\mu = 0.5$ (right shift, small) | 3.10 ±0.20 | 3.26 ±0.05 | **0.00** ±0.00 | **0.00** ±0.00 | 1.81 ±0.11 | 1.83 ±0.11 | **0.00** ±0.00 | **0.00** ±0.00 |
| | $\mu = 2$ (right shift, moderate) | 9.03 ±0.37 | 8.61 ±0.40 | **0.00** ±0.00 | **0.00** ±0.00 | 5.10 ±0.25 | 4.91 ±0.56 | **0.00** ±0.00 | **0.00** ±0.00 |
| | $\mu = 10$ (right shift, extreme) | 29.4 ±1.62 | 25.9 ±3.02 | **0.00** ±0.00 | **0.00** ±0.00 | 19.2 ±3.62 | 18.8 ±1.60 | **0.00** ±0.00 | **0.00** ±0.00 |
| **Scale** | $\lambda = 0.5$ (compression) | 0.09 ±0.03 | 0.12 ±0.03 | 0.52 ±0.06 | **0.01** ±0.00 | 0.20 ±0.03 | 0.18 ±0.04 | 0.71 ±0.09 | 0.01 ±0.01 |
| | $\lambda = 3.0$ (dilation, moderate) | 0.04 ±0.02 | 0.04 ±0.01 | 0.21 ±0.03 | **0.00** ±0.00 | 0.06 ±0.01 | 0.05 ±0.01 | 0.34 ±0.07 | **0.00** ±0.00 |
| | $\lambda = 255$ (dilation, extreme) | 0.05 ±0.02 | 0.04 ±0.01 | 0.24 ±0.04 | **0.00** ±0.00 | 0.06 ±0.01 | 0.05 ±0.01 | 0.38 ±0.07 | **0.00** ±0.00 |
| **Affine** | $\mu = -2, \lambda = 10$ (left shift + strong dilation) | 2.18 ±0.12 | 2.21 ±0.11 | 0.23 ±0.03 | **0.00** ±0.00 | 1.24 ±0.18 | 1.05 ±0.17 | 0.38 ±0.07 | **0.00** ±0.00 |
| | $\mu = 5, \lambda = 0.1$ (shift + strong compression) | 74.8 ±3.04 | 67.9 ±5.02 | 4.00 ±0.15 | 0.24 ±0.03 | 77.5 ±7.55 | 73.5 ±6.92 | 4.97 ±0.31 | **0.18** ±0.04 |
| | $\mu = 5, \lambda = 3$ (shift + dilation, moderate) | 7.86 ±0.31 | 7.69 ±0.28 | 0.21 ±0.03 | **0.00** ±0.00 | 4.42 ±0.18 | 4.28 ±0.52 | 0.34 ±0.07 | **0.00** ±0.00 |

Table 12: Detailed model performance (Accuracy in % as mean ± std over 5 seeds) for different normalization strategies. For each row (perturbation), the best result within each normalization group is highlighted in gray. The overall best result is in **bold**.

| Model / Corruption | Ø | | | | Dropout | | | |
|---|---|---|---|---|---|---|---|---|
| | Stand. | SE | SEqSI | AffEq | Stand. | SE | SEqSI | AffEq |
| Original | 91.7 ±0.3 | 91.3 ±0.1 | 91.2 ±0.3 | 89.6 ±0.1 | 91.3 ±0.2 | 91.6 ±0.2 | 91.7 ±0.3 | 81.2 ±0.6 |
| **Affine transformations (Aff.)** | | | | | | | | |
| Shift | 51.1 ±1.0 | 53.4 ±1.2 | 91.2 ±0.3 | 89.6 ±0.1 | 54.5 ±0.8 | 58.6 ±1.2 | 91.7 ±0.3 | 81.2 ±0.6 |
| Scale (< 1) | 68.0 ±0.8 | 91.3 ±0.1 | 91.2 ±0.3 | 89.6 ±0.1 | 71.1 ±1.0 | 91.6 ±0.1 | 91.7 ±0.3 | 81.2 ±0.6 |
| Scale (> 1) | 91.0 ±0.1 | 91.3 ±0.1 | 91.2 ±0.3 | 89.6 ±0.1 | 87.4 ±0.7 | 91.6 ±0.2 | 91.7 ±0.3 | 81.2 ±0.6 |
| Affine | 64.1 ±0.9 | 65.1 ±0.7 | 91.2 ±0.3 | 89.6 ±0.1 | 63.5 ±0.8 | 68.2 ±0.5 | 91.7 ±0.3 | 81.2 ±0.6 |
| **Non-Affine transformations (NAff.)** | | | | | | | | |
| Shift saturated | 72.0 ±0.6 | 73.4 ±0.4 | 79.0 ±0.6 | 76.7 ±0.4 | 73.2 ±0.4 | 76.0 ±0.2 | 79.6 ±0.3 | 71.4 ±0.7 |
| Scale (> 1) saturated | 78.1 ±0.5 | 78.3 ±0.6 | 77.6 ±0.5 | 76.0 ±0.5 | 77.9 ±0.4 | 79.0 ±0.4 | 79.4 ±0.2 | 72.1 ±0.5 |
| Affine saturated | 59.9 ±0.8 | 59.6 ±1.1 | 72.3 ±1.0 | 70.9 ±0.5 | 61.2 ±0.9 | 62.5 ±0.6 | 74.1 ±0.6 | 68.3 ±0.8 |
| Spatially-varying Affine | 31.8 ±1.0 | 38.2 ±2.3 | 72.5 ±2.2 | 63.6 ±1.0 | 39.9 ±1.3 | 44.4 ±1.1 | 73.8 ±0.9 | 42.6 ±0.8 |
| Contrast Inversion | 9.6 ±0.7 | 15.4 ±2.0 | 56.5 ±1.6 | 52.9 ±1.4 | 13.9 ±0.4 | 41.7 ±3.7 | 58.7 ±0.9 | 37.1 ±1.2 |
| Noise (low) | 90.8 ±0.2 | 90.6 ±0.2 | 90.1 ±0.6 | 88.5 ±0.2 | 90.7 ±0.1 | 90.8 ±0.2 | 90.6 ±0.2 | 80.0 ±0.7 |
| Noise (high) | 19.0 ±2.4 | 22.4 ±3.5 | 15.2 ±1.0 | 20.3 ±0.6 | 22.1 ±0.4 | 18.4 ±1.0 | 13.6 ±0.4 | 17.3 ±2.5 |
| Gamma (darken) | 74.3 ±1.0 | 76.4 ±0.7 | 79.9 ±0.4 | 76.4 ±0.5 | 77.1 ±0.6 | 79.8 ±0.4 | 80.7 ±0.2 | 70.5 ±0.9 |
| Gamma (lighten) | 85.9 ±0.6 | 85.3 ±0.7 | 90.3 ±0.2 | 88.6 ±0.1 | 87.0 ±0.3 | 87.3 ±0.3 | 90.9 ±0.2 | 80.4 ±0.6 |

| Model / Corruption | IN | | | | BN | | | |
|---|---|---|---|---|---|---|---|---|
| | Stand. | SE | SEqSI | AffEq | Stand. | SE | SEqSI | AffEq |
| Original | 94.2 ±0.1 | 94.0 ±0.2 | 93.9 ±0.3 | 91.6 ±0.2 | **95.7** ±0.1 | **95.7** ±0.2 | **95.7** ±0.2 | 94.3 ±0.1 |
| **Affine transformations (Aff.)** | | | | | | | | |
| Shift | 91.6 ±0.2 | 91.7 ±0.2 | 93.9 ±0.3 | 91.6 ±0.2 | 40.3 ±1.7 | 39.1 ±1.0 | **95.7** ±0.2 | 34.1 ±0.5 |
| Scale (< 1) | 94.1 ±0.1 | 93.9 ±0.2 | 93.5 ±0.3 | 90.8 ±0.3 | 79.1 ±1.2 | 79.2 ±0.5 | 80.8 ±0.5 | 74.3 ±1.4 |
| Scale (> 1) | 94.2 ±0.1 | 94.0 ±0.2 | 93.9 ±0.3 | 91.6 ±0.2 | 64.5 ±3.7 | 62.0 ±3.9 | 86.0 ±1.2 | 50.2 ±2.6 |
| Affine | 90.9 ±0.1 | 91.2 ±0.2 | 93.8 ±0.3 | 91.4 ±0.2 | 34.2 ±2.1 | 33.4 ±1.8 | 84.4 ±1.1 | 25.1 ±0.9 |
| **Non-Affine transformations (NAff.)** | | | | | | | | |
| Shift saturated | 79.9 ±0.6 | 80.3 ±0.3 | 80.1 ±0.4 | 78.2 ±0.5 | 77.9 ±0.7 | 77.9 ±0.4 | **81.9** ±0.5 | 74.8 ±0.6 |
| Scale (> 1) saturated | 79.7 ±0.6 | 79.7 ±0.3 | 79.0 ±0.2 | 76.5 ±0.6 | 79.2 ±0.9 | 79.6 ±0.2 | 78.8 ±0.5 | 76.8 ±0.8 |
| Affine saturated | 72.3 ±0.8 | 72.8 ±0.5 | 73.3 ±0.4 | 71.1 ±0.4 | 67.1 ±0.6 | 67.3 ±0.5 | **74.3** ±0.5 | 65.1 ±0.6 |
| Spatially-varying Affine | 86.5 ±0.6 | 86.8 ±0.3 | 93.4 ±0.2 | 40.7 ±1.1 | 47.5 ±1.1 | 47.4 ±1.2 | 71.9 ±1.0 | 38.6 ±0.9 |
| Contrast Inversion | 58.4 ±0.8 | 58.3 ±0.9 | 67.6 ±0.6 | 61.6 ±0.7 | 10.8 ±1.1 | 10.8 ±0.4 | 67.9 ±0.5 | 10.2 ±1.0 |
| Noise (low) | 92.4 ±0.1 | 92.2 ±0.3 | 91.5 ±0.2 | 89.9 ±0.2 | 92.8 ±0.3 | 92.8 ±0.1 | 92.2 ±0.4 | 91.2 ±0.2 |
| Noise (high) | 28.6 ±1.7 | 27.8 ±1.7 | 19.3 ±1.3 | **32.2** ±1.0 | 13.3 ±1.8 | 12.7 ±0.7 | 12.8 ±0.9 | 13.3 ±2.5 |
| Gamma (darken) | 85.0 ±0.5 | 85.2 ±0.3 | 84.7 ±0.5 | 80.2 ±0.6 | 82.3 ±0.5 | 82.5 ±0.4 | 85.9 ±0.1 | 77.2 ±0.5 |
| Gamma (lighten) | 93.2 ±0.1 | 93.1 ±0.3 | 93.2 ±0.3 | 90.7 ±0.1 | 92.7 ±0.1 | 92.8 ±0.3 | 94.6 ±0.1 | 90.6 ±0.4 |

| Model / Corruption | LN | | | | Mix | | | |
|---|---|---|---|---|---|---|---|---|
| | Stand. | SE | SEqSI | AffEq | Stand. | SE | SEqSI | AffEq |
| Original | 93.3 ±0.1 | 93.5 ±0.3 | 94.1 ±0.1 | 92.4 ±0.3 | 95.4 ±0.1 | 95.4 ±0.1 | 95.5 ±0.2 | 93.7 ±0.1 |
| **Affine transformations (Aff.)** | | | | | | | | |
| Shift | 56.4 ±0.8 | 59.5 ±0.6 | 94.1 ±0.1 | 92.4 ±0.3 | 94.6 ±0.1 | 94.6 ±0.1 | 95.5 ±0.2 | 93.7 ±0.1 |
| Scale (< 1) | 79.2 ±0.9 | 92.9 ±0.3 | 91.8 ±0.4 | 92.2 ±0.3 | 95.2 ±0.1 | 95.2 ±0.1 | 94.6 ±0.2 | 93.6 ±0.2 |
| Scale (> 1) | 92.4 ±0.2 | 93.5 ±0.3 | 94.1 ±0.1 | 92.4 ±0.3 | 95.4 ±0.1 | 95.4 ±0.1 | 95.4 ±0.2 | 93.7 ±0.1 |
| Affine | 66.5 ±1.1 | 68.3 ±0.8 | 93.5 ±0.2 | 92.4 ±0.3 | 93.2 ±0.2 | 93.4 ±0.1 | 95.2 ±0.1 | 93.7 ±0.2 |
| **Non-Affine transformations (NAff.)** | | | | | | | | |
| Shift saturated | 73.8 ±1.0 | 77.7 ±0.7 | 80.9 ±0.5 | 79.2 ±0.7 | 81.4 ±0.4 | 81.7 ±0.3 | 81.6 ±0.3 | 79.2 ±0.4 |
| Scale (> 1) saturated | 78.9 ±0.4 | 79.5 ±0.7 | **80.1** ±0.3 | 77.8 ±0.6 | 79.5 ±0.6 | 79.8 ±0.4 | 79.8 ±0.4 | 76.3 ±0.8 |
| Affine saturated | 66.3 ±1.5 | 68.0 ±0.8 | **74.3** ±0.6 | 72.2 ±1.1 | 73.4 ±0.8 | 74.2 ±0.9 | 74.2 ±0.4 | 70.5 ±0.6 |
| Spatially-varying Affine | 47.3 ±2.0 | 51.8 ±2.5 | 82.6 ±0.9 | 73.5 ±3.2 | 89.4 ±0.3 | 89.5 ±0.2 | **94.9** ±0.2 | 44.6 ±1.0 |
| Contrast Inversion | 11.4 ±1.5 | 10.4 ±2.4 | 63.4 ±0.8 | 55.1 ±2.6 | 66.7 ±0.3 | 67.2 ±0.5 | **71.7** ±0.4 | 66.8 ±0.8 |
| Noise (low) | 91.1 ±0.3 | 91.7 ±0.3 | 92.1 ±0.2 | 91.2 ±0.3 | **93.1** ±0.1 | **93.1** ±0.1 | 92.3 ±0.3 | 91.1 ±0.1 |
| Noise (high) | 15.5 ±0.6 | 18.0 ±1.5 | 13.9 ±1.0 | 23.1 ±3.0 | 13.6 ±1.5 | 13.7 ±0.7 | 12.2 ±0.4 | 21.0 ±1.2 |
| Gamma (darken) | 75.7 ±0.4 | 80.4 ±0.7 | 83.7 ±0.4 | 81.8 ±0.6 | **86.6** ±0.1 | 86.4 ±0.2 | 86.3 ±0.4 | 82.6 ±0.4 |
| Gamma (lighten) | 90.9 ±0.3 | 91.2 ±0.5 | 93.3 ±0.1 | 91.7 ±0.3 | **94.8** ±0.2 | **94.8** ±0.1 | **94.8** ±0.1 | 92.9 ±0.2 |

## C.8 Demonstrating Scalability on a More Complex Task: Oxford-IIIT Pets and Stanford Cars Classification

To demonstrate the scalability of our *SEqSI* architecture to more complex and higher-resolution computer vision tasks, we conducted experiments on the Oxford-IIIT Pet (Parkhi et al., 2012) and Stanford Cars (Krause et al., 2013) datasets.

**Scaling the Challenge: From CIFAR-10 to Oxford-IIIT Pets and Stanford Cars.**   The transition from CIFAR-10 (Krizhevsky et al., 2009) to the Oxford-IIIT Pets and Stanford Cars datasets represents a significant increase in difficulty, providing a robust test for scalability:

- **Image Resolution:** The input images are scaled from $32 \times 32$ pixels for CIFAR-10 to $224 \times 224$ pixels, which is a nearly 50-fold increase in the number of pixels. Image resolution can vary, but all images are resized to $224 \times 224$ pixels. This requires an architecture capable of handling a much larger receptive field and spatial hierarchy.
- **Task Complexity:** The classification task becomes significantly more fine-grained, moving from 10 general object classes in CIFAR-10 to 37 distinct pet breeds in Oxford-IIIT Pets, and an even more challenging 196 classes of car models (differentiated by make, model, and year) in Stanford Cars.
- **Image Variability:** Both datasets feature greater intra-class variation in object scale, pose, and background clutter compared to CIFAR-10, posing a greater challenge for generalization.

**Scaling the Architecture: From ResNet-20 to ResNet-18.**   To accommodate this increase in scale, we adapted the backbone of the model from a ResNet-20-like structure to a standard ResNet-18 architecture (He et al., 2015), a common choice for processing $224 \times 224$ images. The core principles of our model families (*Standard*, *SEq*, *SEqSI*, *AffEq*) are maintained, but the underlying architecture is scaled up as follows:

- **Aggressive Initial Downsampling:** The initial layer is adapted for high-resolution inputs. It uses a large $7 \times 7$ convolution with a stride of 2, followed by a $3 \times 3$ max-pooling layer with a stride of 2. This combination efficiently reduces the spatial resolution from $224 \times 224$ to $56 \times 56$, a standard practice to manage computational cost and build a hierarchy of features.

- **Deeper and Wider Structure:** The network features four residual stages instead of three. The number of channels progresses from 64 to 128, 256, and finally 512, creating a deeper and wider feature hierarchy suitable for a more complex task. The spatial resolution is successively halved at each stage, down to a final $7 \times 7$ feature map.

- **Adapted Classification Head:** The classification head operates on the $7 \times 7 \times 512$ feature map. After global average pooling, the resulting 512-dimensional vector is projected to the appropriate number of classes (37 for Pets, 196 for Cars).

Crucially, the fundamental design of each model family remains unchanged. For instance, the *SEqSI* model still employs a shift-invariant convolution as its first layer and bias-free convolutions throughout the rest of the network. While the overall structure is scaled up, the internal design of the residual blocks and the application of architectural constraints are conceptually identical to those used for the CIFAR-10 experiments (see Appendix C.1).

**Results and Discussion.**   The primary objective of these experiments on the Oxford-IIIT Pets and Stanford Cars datasets was not to achieve state-of-the-art accuracy, but rather to demonstrate the scalability of our constrained architectures. For these comparisons, it is important to consider the *Standard* model as the baseline, as it represents an unconstrained CNN. The key point is that both *SEqSI* and *AffEq* models, despite their significant architectural constraints, preserve their ability to learn the complex and fine-grained features required for these challenging high-resolution tasks.

The results, presented in Table 13 and Table 14, serve as strong evidence for this. First, they confirm that **the theoretical guarantees of our models are preserved when scaled to a deeper ResNet-18 backbone**: both *SEqSI* and *AffEq* maintain perfect invariance to affine transformations, a feat the

*Standard* models cannot achieve even with data augmentation. More importantly, this **robustness does not come at the cost of learning capability**.

On the Oxford-IIIT Pets dataset (Table 13), without data augmentation, *SEqSI* **(44.4% accuracy) is highly competitive with the *Standard* model** (44.5% accuracy) on original images, **while demonstrating superior robustness to affine corruptions** (e.g., 44.4% for *SEqSI* vs 12.4% for *Standard* under a shift). **With data augmentation (Train Aug. = All), *SEqSI* (31.7% accuracy on original images) largely outperforms all other models**, including *Standard* (14.6% accuracy), especially on non-affine transformations like "Spatially-varying Affine" (25.2% for *SEqSI* vs 10.0% for *Standard*).

For the even more complex Stanford Cars dataset (Table 14), both *SEqSI* **(33.2% accuracy) and *AffEq* (44.8% accuracy) significantly outperform the *Standard* model** (27.3% accuracy) on clean data without any augmentation. **With data augmentation (Train Aug. = All), *SEqSI* (22.8% accuracy) and *AffEq* (20.8% accuracy) remain clearly superior to *Standard*** (6.9% accuracy) on original images, and maintain this advantage across various corruptions (e.g., for "Gamma (darken)", *SEqSI* achieves 13.5% and *AffEq* 12.8% compared to *Standard* 5.1% accuracy).

It is important to note two key observations from these experiments:

- The significant drop in performance for all models when trained with strong data augmentation (Train Aug. = All, using parameters from Appendix C.2) suggests that the photometric perturbations applied for augmentation are too aggressive for these higher-resolution datasets. This motivated our complementary study using more moderate perturbations, detailed in Appendix C.9.

- In the most complex scenario of the Stanford Cars dataset, *AffEq* consistently shows a clear advantage, often achieving the highest accuracy despite being the most constrained network. This suggests that the architecture properties can be beneficial for generalization and convergence on challenging tasks.

**These findings validate the scalability of our approach.** In conclusion, these findings validate that the principles behind *SEqSI* **and *AffEq* are indeed scalable.** They can be successfully integrated into standard, deeper backbones to tackle complex, real-world computer vision problems, offering a robust and principled alternative to standard architectures. The absolute accuracy scores should be viewed in the context of a relatively shallow ResNet-18 model trained for a limited number of epochs (200). The crucial point is the demonstrated ability of these constrained models to learn effectively at scale while providing certified robustness. This is further supported by experiments with more moderate perturbations, as detailed in Appendix C.9.

Table 13: **Robustness to photometric corruptions on the Oxford-IIIT Pet test set.** Test accuracy (%) (mean ± std over 5 seeds) of four ResNet-18 architectures. Models are grouped by the training augmentation strategy used: none (∅), affine (**Aff.**), non-affine (**NAff.**), or all combined (**All**). Each row corresponds to a different perturbation applied at evaluation. Within each training strategy (column group), the best result is highlighted in gray. The overall best accuracy for each perturbation is in **bold**. Models were trained for 200 epochs.

| Model / Corruption | Train Aug. = ∅ | | | | Train Aug. = Aff | | | |
|---|---|---|---|---|---|---|---|---|
| | Stand. | SEq | SEqSI | AffEq | Stand. | SEq | SEqSI | AffEq |
| Original | **44.5** ±1.5 | 35.9 ±0.7 | 44.4 ±1.3 | 38.1 ±0.9 | 24.8 ±6.7 | 26.6 ±4.8 | 35.9 ±2.1 | 32.5 ±0.4 |
| **Affine transformations (Aff.)** | | | | | | | | |
| Shift | 12.4 ±0.7 | 10.3 ±0.3 | **44.4** ±1.3 | 38.1 ±0.9 | 23.5 ±6.6 | 25.3 ±4.8 | 35.9 ±2.1 | 32.5 ±0.4 |
| Scale ($<1$) | 28.1 ±0.7 | 35.9 ±0.7 | **44.4** ±1.3 | 38.2 ±0.9 | 22.7 ±6.3 | 26.6 ±4.8 | 35.9 ±2.1 | 32.5 ±0.4 |
| Scale ($>1$) | 37.9 ±0.4 | 35.9 ±0.7 | **44.4** ±1.3 | 38.1 ±0.9 | 24.5 ±6.7 | 26.6 ±4.8 | 35.9 ±2.1 | 32.5 ±0.4 |
| Affine | 15.9 ±0.9 | 15.8 ±0.4 | **44.4** ±1.3 | 38.2 ±0.9 | 22.3 ±6.3 | 25.1 ±4.3 | 35.9 ±2.1 | 32.5 ±0.4 |
| **Non-Affine transformations (NAff.)** | | | | | | | | |
| Shift saturated | 22.0 ±0.4 | 18.5 ±0.7 | **28.9** ±0.8 | 27.0 ±0.8 | 17.9 ±4.9 | 19.0 ±3.1 | 23.7 ±1.2 | 23.5 ±0.8 |
| Scale ($>1$) saturated | 27.6 ±1.0 | 22.7 ±0.4 | 27.9 ±0.9 | 27.7 ±0.4 | 18.1 ±4.5 | 18.7 ±3.3 | 23.6 ±0.7 | 23.7 ±0.2 |
| Affine saturated | 13.5 ±1.0 | 10.3 ±0.5 | 25.4 ±0.3 | 25.7 ±0.6 | 16.2 ±3.9 | 17.5 ±2.8 | 22.0 ±0.8 | 22.3 ±0.1 |
| Spatially-varying Affine | 6.2 ±0.5 | 7.4 ±0.6 | **31.1** ±1.8 | 16.4 ±0.9 | 16.8 ±4.7 | 22.4 ±4.0 | 25.3 ±1.0 | 13.8 ±1.0 |
| Contrast Inversion | 2.4 ±0.3 | 2.6 ±0.3 | 6.9 ±0.4 | 5.8 ±0.3 | 3.9 ±0.6 | 3.7 ±0.2 | 6.3 ±0.6 | 4.8 ±0.6 |
| Noise (low) | **44.3** ±1.5 | 35.9 ±0.7 | 43.0 ±0.9 | 37.9 ±0.9 | 24.6 ±6.6 | 26.7 ±4.8 | 34.7 ±1.8 | 32.5 ±0.4 |
| Noise (high) | 29.6 ±1.8 | 28.0 ±1.6 | 5.3 ±1.3 | 16.1 ±3.4 | 13.5 ±3.9 | 16.1 ±2.9 | 3.9 ±1.2 | 16.2 ±1.4 |
| Gamma (darken) | 18.8 ±0.5 | 16.5 ±0.5 | 21.5 ±1.0 | 21.5 ±0.5 | 13.8 ±3.6 | 14.1 ±2.0 | 17.5 ±1.3 | 19.0 ±0.5 |
| Gamma (lighten) | 31.6 ±1.2 | 24.1 ±0.8 | **38.0** ±1.0 | 35.1 ±1.0 | 20.2 ±5.5 | 22.6 ±3.9 | 30.2 ±1.8 | 29.8 ±0.5 |

| Model / Corruption | Train Aug. = NAff | | | | Train Aug. = All | | | |
|---|---|---|---|---|---|---|---|---|
| | Stand. | SEq | SEqSI | AffEq | Stand. | SEq | SEqSI | AffEq |
| Original | 20.0 ±8.1 | 18.4 ±3.7 | 34.7 ±1.1 | 30.1 ±1.3 | 14.6 ±3.4 | 13.9 ±2.0 | 31.7 ±1.8 | 25.6 ±2.3 |
| **Affine transformations (Aff.)** | | | | | | | | |
| Shift | 13.2 ±3.9 | 13.7 ±1.8 | 34.7 ±1.1 | 30.1 ±1.3 | 13.4 ±3.1 | 12.0 ±1.9 | 31.7 ±1.8 | 25.6 ±2.3 |
| Scale ($<1$) | 14.9 ±6.0 | 18.4 ±3.7 | 34.7 ±1.1 | 30.1 ±1.3 | 12.2 ±2.6 | 13.9 ±2.0 | 31.7 ±1.8 | 25.6 ±2.3 |
| Scale ($>1$) | 17.6 ±7.8 | 18.4 ±3.7 | 34.7 ±1.1 | 30.1 ±1.3 | 14.2 ±3.5 | 13.9 ±2.0 | 31.7 ±1.8 | 25.6 ±2.3 |
| Affine | 12.8 ±4.5 | 14.8 ±2.3 | 34.7 ±1.1 | 30.1 ±1.3 | 12.7 ±3.1 | 12.3 ±1.7 | 31.7 ±1.8 | 25.6 ±2.3 |
| **Non-Affine transformations (NAff.)** | | | | | | | | |
| Shift saturated | 16.8 ±6.5 | 14.9 ±2.8 | 27.9 ±0.8 | 24.5 ±0.6 | 11.9 ±2.5 | 11.2 ±1.3 | 24.4 ±1.5 | 20.6 ±1.6 |
| Scale ($>1$) saturated | 16.7 ±6.6 | 16.1 ±2.9 | **30.3** ±1.0 | 26.3 ±1.4 | 12.6 ±2.7 | 11.6 ±1.9 | 26.0 ±1.2 | 22.0 ±2.0 |
| Affine saturated | 14.9 ±6.3 | 14.3 ±2.2 | **28.2** ±1.2 | 25.1 ±1.6 | 11.1 ±2.0 | 10.7 ±1.4 | 24.2 ±0.9 | 20.9 ±1.7 |
| Spatially-varying Affine | 10.7 ±3.1 | 11.3 ±1.5 | 29.7 ±1.0 | 18.6 ±1.1 | 10.0 ±2.1 | 10.8 ±1.6 | 25.2 ±1.3 | 14.1 ±1.8 |
| Contrast Inversion | 17.6 ±8.4 | 15.2 ±3.3 | **18.6** ±2.9 | 9.7 ±1.1 | 10.4 ±2.2 | 7.8 ±1.7 | 11.8 ±0.5 | 5.8 ±1.4 |
| Noise (low) | 20.0 ±8.2 | 18.3 ±3.7 | 34.8 ±1.0 | 30.1 ±1.3 | 14.6 ±3.4 | 13.9 ±2.1 | 31.5 ±2.0 | 25.6 ±2.3 |
| Noise (high) | 18.0 ±8.2 | 16.0 ±3.0 | **30.2** ±1.7 | 27.2 ±1.8 | 11.8 ±2.6 | 10.9 ±1.6 | 25.1 ±2.3 | 23.2 ±1.9 |
| Gamma (darken) | 15.9 ±6.2 | 12.5 ±2.3 | **23.1** ±1.6 | 21.6 ±0.7 | 10.8 ±1.6 | 9.1 ±1.0 | 18.8 ±0.9 | 17.4 ±1.5 |
| Gamma (lighten) | 18.1 ±8.2 | 17.2 ±3.2 | 33.4 ±1.2 | 29.3 ±1.6 | 13.2 ±2.8 | 12.8 ±2.0 | 28.7 ±1.7 | 24.7 ±1.8 |

Table 14: **Robustness to photometric corruptions on the Stanford Cars test set.** Test accuracy (%) (mean ± std over 5 seeds) of four ResNet-18 architectures. Models are grouped by the training augmentation strategy used: none (∅), affine (**Aff.**), non-affine (**NAff.**), or all combined (**All**). Each row corresponds to a different perturbation applied at evaluation. Within each training strategy (column group), the best result is highlighted in gray. The overall best accuracy for each perturbation is in **bold**. Models were trained for 200 epochs.

| Corruption \ Model | Train Aug. = ∅ | | | | Train Aug. = Aff | | | |
|---|---|---|---|---|---|---|---|---|
| | Stand. | SEq | SEqSI | AffEq | Stand. | SEq | SEqSI | AffEq |
| Original | 27.3 ±3.4 | 24.9 ±2.0 | 33.2 ±2.7 | **44.8** ±1.4 | 18.0 ±5.2 | 18.5 ±1.4 | 24.7 ±2.8 | 25.3 ±0.6 |
| **Affine transformations (Aff.)** | | | | | | | | |
| Shift | 4.6 ±0.7 | 4.5 ±0.2 | 33.2 ±2.7 | **44.8** ±1.4 | 17.6 ±5.2 | 18.1 ±1.5 | 24.7 ±2.8 | 25.3 ±0.6 |
| Scale (< 1) | 17.0 ±2.2 | 24.9 ±2.0 | 33.2 ±2.7 | **44.8** ±1.4 | 16.4 ±4.8 | 18.5 ±1.4 | 24.7 ±2.8 | 25.3 ±0.6 |
| Scale (> 1) | 24.6 ±3.3 | 24.9 ±2.0 | 33.2 ±2.7 | **44.8** ±1.4 | 17.6 ±5.1 | 18.5 ±1.4 | 24.7 ±2.8 | 25.3 ±0.6 |
| Affine | 7.4 ±1.0 | 7.5 ±0.5 | 33.2 ±2.7 | **44.8** ±1.4 | 16.9 ±5.0 | 17.7 ±1.5 | 24.7 ±2.8 | 25.3 ±0.6 |
| **Non-Affine transformations (NAff.)** | | | | | | | | |
| Shift saturated | 10.6 ±1.6 | 10.1 ±0.9 | 21.8 ±1.7 | **31.5** ±1.0 | 12.1 ±3.5 | 12.5 ±1.0 | 16.3 ±1.8 | 18.0 ±0.6 |
| Scale (> 1) saturated | 17.7 ±2.3 | 16.1 ±1.6 | 21.4 ±1.8 | **31.9** ±0.9 | 12.2 ±3.1 | 12.4 ±0.9 | 16.6 ±2.3 | 18.4 ±0.6 |
| Affine saturated | 3.8 ±0.5 | 3.9 ±0.4 | 20.0 ±1.8 | **29.4** ±0.8 | 11.2 ±3.0 | 11.2 ±0.9 | 15.2 ±2.0 | 17.1 ±0.5 |
| Spatially-varying Affine | 2.1 ±0.5 | 2.5 ±0.2 | 28.8 ±2.4 | **36.3** ±1.0 | 14.0 ±4.3 | 16.7 ±1.3 | 21.0 ±1.9 | 18.7 ±0.5 |
| Contrast Inversion | 0.5 ±0.0 | 0.5 ±0.0 | 1.3 ±0.1 | 7.5 ±1.5 | 0.7 ±0.1 | 0.8 ±0.1 | 1.1 ±0.1 | 3.4 ±0.3 |
| Noise (low) | 27.3 ±3.4 | 24.9 ±1.9 | 33.0 ±2.8 | **44.6** ±1.3 | 17.9 ±5.2 | 18.5 ±1.3 | 24.6 ±2.8 | 25.2 ±0.6 |
| Noise (high) | 11.8 ±1.6 | 11.5 ±1.4 | 7.0 ±2.1 | 21.4 ±2.4 | 10.6 ±1.7 | 11.3 ±0.7 | 7.0 ±2.2 | 15.2 ±1.0 |
| Gamma (darken) | 11.1 ±1.7 | 9.3 ±0.8 | 17.2 ±1.6 | **25.7** ±0.7 | 10.1 ±2.6 | 9.7 ±0.4 | 12.8 ±1.4 | 14.4 ±0.5 |
| Gamma (lighten) | 17.1 ±2.5 | 15.7 ±1.1 | 29.4 ±2.5 | **42.2** ±1.1 | 16.1 ±4.6 | 16.7 ±1.2 | 21.8 ±2.7 | 23.7 ±0.5 |

| Corruption \ Model | Train Aug. = NAff | | | | Train Aug. = All | | | |
|---|---|---|---|---|---|---|---|---|
| | Stand. | SEq | SEqSI | AffEq | Stand. | SEq | SEqSI | AffEq |
| Original | 6.1 ±4.6 | 17.4 ±1.9 | 32.0 ±1.5 | 31.0 ±1.4 | 6.9 ±4.4 | 12.6 ±2.9 | 22.8 ±3.1 | 20.8 ±1.9 |
| **Affine transformations (Aff.)** | | | | | | | | |
| Shift | 4.8 ±3.5 | 13.2 ±1.7 | 32.0 ±1.5 | 31.0 ±1.4 | 6.4 ±4.3 | 11.8 ±2.7 | 22.8 ±3.1 | 20.8 ±1.9 |
| Scale (< 1) | 5.0 ±3.7 | 17.4 ±1.9 | 32.0 ±1.5 | 31.0 ±1.4 | 6.1 ±3.9 | 12.6 ±2.9 | 22.8 ±3.1 | 20.8 ±1.9 |
| Scale (> 1) | 5.7 ±4.3 | 17.4 ±1.9 | 32.0 ±1.5 | 31.0 ±1.4 | 6.6 ±4.3 | 12.6 ±2.9 | 22.8 ±3.1 | 20.8 ±1.9 |
| Affine | 4.7 ±3.4 | 13.6 ±1.8 | 32.0 ±1.5 | 31.0 ±1.4 | 6.0 ±3.8 | 11.7 ±2.5 | 22.8 ±3.1 | 20.8 ±1.9 |
| **Non-Affine transformations (NAff.)** | | | | | | | | |
| Shift saturated | 5.0 ±3.6 | 13.4 ±1.5 | 24.8 ±1.3 | 23.9 ±1.0 | 5.4 ±3.3 | 9.5 ±2.2 | 17.2 ±2.2 | 16.2 ±1.4 |
| Scale (> 1) saturated | 5.0 ±3.7 | 14.4 ±1.6 | 27.3 ±1.4 | 26.1 ±1.4 | 5.6 ±3.4 | 9.9 ±2.2 | 18.8 ±2.4 | 17.2 ±1.3 |
| Affine saturated | 4.6 ±3.3 | 13.0 ±1.5 | 25.7 ±1.4 | 24.4 ±1.3 | 5.2 ±3.3 | 9.1 ±1.9 | 17.5 ±2.3 | 16.3 ±1.3 |
| Spatially-varying Affine | 4.2 ±3.1 | 11.9 ±1.4 | 29.8 ±1.4 | 28.2 ±1.1 | 5.4 ±3.5 | 10.8 ±2.4 | 19.8 ±2.7 | 17.4 ±1.8 |
| Contrast Inversion | 4.8 ±3.6 | 13.7 ±1.5 | 19.5 ±1.9 | **22.9** ±1.1 | 3.7 ±2.2 | 3.8 ±1.5 | 3.8 ±0.9 | 12.0 ±1.2 |
| Noise (low) | 6.0 ±4.6 | 17.4 ±2.0 | 32.0 ±1.5 | 31.0 ±1.4 | 6.9 ±4.4 | 12.6 ±2.8 | 22.8 ±3.1 | 20.7 ±1.9 |
| Noise (high) | 5.5 ±4.0 | 15.9 ±1.7 | **28.0** ±1.4 | 25.2 ±1.1 | 6.3 ±3.9 | 10.8 ±2.3 | 19.4 ±2.4 | 16.6 ±1.5 |
| Gamma (darken) | 4.7 ±3.4 | 10.9 ±1.3 | 20.1 ±1.3 | 19.1 ±1.0 | 5.1 ±3.2 | 7.5 ±1.5 | 13.5 ±1.7 | 12.8 ±1.3 |
| Gamma (lighten) | 5.8 ±4.4 | 17.1 ±1.9 | 31.4 ±1.4 | 30.9 ±1.3 | 6.5 ±4.2 | 11.9 ±2.5 | 21.7 ±3.0 | 20.3 ±1.8 |

## C.9 Details on 'Moderate' Photometric Perturbations for Scalability Experiments

The photometric transformations used for the scalability experiments on Oxford-IIIT Pet and Stanford Cars are detailed below. The parameters were adjusted to a "moderate" intensity, less aggressive than those for CIFAR-10 (Appendix C.2), to better assess model performance on these higher-resolution datasets.

**Transformation Groups.** Transformations are categorized into two main groups:

**Affine Transformations (Aff):** This group consists of 4 non-saturated global affine intensity transformations.

- **Global Additive Shift:** $T(\boldsymbol{x}) = \boldsymbol{x} + \mu$, with $\mu \sim \mathcal{U}[-0.5, 0.5]$.
- **Global Scaling (Compression):** $T(\boldsymbol{x}) = \lambda\boldsymbol{x}$, with $\lambda \sim \mathcal{U}[0.5, 1.0]$.
- **Global Scaling (Dilation):** $T(\boldsymbol{x}) = \lambda\boldsymbol{x}$, with $\lambda \sim \mathcal{U}[1.0, 2.0]$.
- **Global Affine:** $T(\boldsymbol{x}) = \lambda\boldsymbol{x} + \mu$, with $\lambda \sim \mathcal{U}[0.5, 2.0]$ and $\mu \sim \mathcal{U}[-0.5, 0.5]$.

**Non-Affine and Saturated Transformations (NAff):** This group includes 9 transformations that are either non-affine, spatially-varying, or saturated.

- **Saturated Additive Shift:** $T(\boldsymbol{x}) = \mathrm{clip}(\boldsymbol{x} + \mu, 0, 1)$, with $\mu \sim \mathcal{U}[-0.3, 0.3]$.
- **Saturated Scaling:** $T(\boldsymbol{x}) = \mathrm{clip}(\lambda\boldsymbol{x}, 0, 1)$, with $\lambda \sim \mathcal{U}[1.0, 1.8]$.
- **Saturated Affine:** $T(\boldsymbol{x}) = \mathrm{clip}(\lambda\boldsymbol{x} + \mu, 0, 1)$, with $\lambda \sim \mathcal{U}[0.8, 1.2]$ and $\mu \sim \mathcal{U}[-0.2, 0.2]$.
- **Spatially-Varying Affine (Linear):** $T(\boldsymbol{x})(u, v) = \lambda(u, v)\boldsymbol{x}(u, v) + \mu(u, v)$, with $\lambda(u, v) \sim \mathcal{U}[0.5, 1.0]$ and $\mu(u, v) \sim \mathcal{U}[-0.5, 0.5]$ for each pixel $(u, v)$.
- **Contrast Inversion (Linear):** A spatially-varying scaling $T(\boldsymbol{x})(u, v) = \lambda(u, v)\boldsymbol{x}(u, v)$, with $\lambda(u, v) \sim \mathcal{U}[-1.0, -0.5]$ for each pixel $(u, v)$.
- **Additive Gaussian Noise (Low):** $T(\boldsymbol{x}) = \mathrm{clip}(\boldsymbol{x} + n, 0, 1)$, where $n \sim \mathcal{N}(0, \sigma^2)$ and $\sigma \sim \mathcal{U}[0.0, 0.03]$.
- **Additive Gaussian Noise (High):** $T(\boldsymbol{x}) = \mathrm{clip}(\boldsymbol{x} + n, 0, 1)$, where $n \sim \mathcal{N}(0, \sigma^2)$ and $\sigma \sim \mathcal{U}[0.1, 0.15]$.
- **Gamma Correction (Lighten):** $T(\boldsymbol{x}) = \boldsymbol{x}^\gamma$, with $\gamma \sim \mathcal{U}[0.5, 1.0]$.
- **Gamma Correction (Darken):** $T(\boldsymbol{x}) = \boldsymbol{x}^\gamma$, with $\gamma \sim \mathcal{U}[1.0, 2.5]$.

**Validation of the results.** The results of these experiments, presented in Tables 15 and 16, show that the 'moderate' data augmentation strategy is better suited for these datasets, leading to improved overall performance. For instance, the performance of the *SEqSI* model trained with all augmentations (Train Aug. = All) on the original test data improves significantly, increasing from 31.7% (Table 13) to 45.7% on Oxford-IIIT Pets, and from 22.8% (Table 14) to 37.0% on Stanford Cars. Crucially, the observations regarding the superior performance and robustness of *SEqSI* and *AffEq* models remain consistent, confirming the conclusions from Appendix C.8 about the scalability and validity of our approach.

Table 15: **Robustness to 'moderate' photometric corruptions on the Oxford-IIIT Pet test set.** Test accuracy (%) (mean ± std over 5 seeds) of four ResNet-18 architectures. Models are grouped by the training augmentation strategy used: none (∅), affine (**Aff.**), non-affine (**NAff.**), or all combined (**All**). Each row corresponds to a different perturbation applied at evaluation. Within each training strategy (column group), the best result is highlighted in gray. The overall best accuracy for each perturbation is in **bold**. Models were trained for 200 epochs.

| Model \ Corruption | Train Aug. = ∅ | | | | Train Aug. = Aff | | | |
|---|---|---|---|---|---|---|---|---|
| | Stand. | SEq | SEqSI | AffEq | Stand. | SEq | SEqSI | AffEq |
| Original | 44.5 ±1.5 | 35.9 ±0.7 | 44.4 ±1.3 | 38.2 ±0.9 | 41.4 ±3.9 | 40.6 ±1.6 | 43.0 ±1.4 | 37.6 ±0.5 |
| **Affine transformations (Aff.)** | | | | | | | | |
| Shift | 43.4 ±1.5 | 35.1 ±0.8 | 44.4 ±1.3 | 38.2 ±0.9 | 41.4 ±3.8 | 40.5 ±1.6 | 43.0 ±1.4 | 37.6 ±0.5 |
| Scale (< 1) | 44.4 ±1.4 | 35.9 ±0.7 | 44.4 ±1.3 | 38.2 ±0.9 | 41.4 ±3.9 | 40.6 ±1.6 | 43.0 ±1.4 | 37.6 ±0.5 |
| Scale (> 1) | 44.2 ±1.4 | 35.9 ±0.7 | 44.4 ±1.3 | 38.2 ±0.9 | 41.3 ±4.0 | 40.6 ±1.7 | 43.0 ±1.4 | 37.6 ±0.5 |
| Affine | 43.3 ±1.4 | 35.1 ±0.8 | 44.4 ±1.3 | 38.2 ±0.9 | 41.3 ±4.0 | 40.4 ±1.6 | 43.0 ±1.4 | 37.6 ±0.5 |
| **Non-Affine transformations (NAff.)** | | | | | | | | |
| Shift saturated | 43.4 ±1.5 | 34.8 ±0.7 | 44.2 ±1.3 | 38.0 ±0.9 | 41.4 ±3.9 | 40.2 ±1.7 | 42.7 ±1.1 | 37.4 ±0.6 |
| Scale (> 1) saturated | 44.1 ±1.6 | 35.5 ±0.8 | 44.0 ±1.4 | 37.9 ±0.8 | 41.1 ±3.8 | 40.3 ±1.7 | 42.6 ±1.2 | 37.6 ±0.6 |
| Affine saturated | 11.5 ±0.9 | 8.4 ±0.9 | 25.7 ±0.8 | 27.0 ±0.6 | 25.9 ±2.8 | 21.7 ±1.6 | 24.6 ±1.0 | 26.6 ±0.7 |
| Spatially-varying Affine | 43.8 ±1.5 | 35.3 ±0.6 | 44.5 ±1.4 | 38.1 ±0.8 | 41.3 ±4.0 | 40.4 ±1.6 | 42.9 ±1.4 | 37.6 ±0.5 |
| Contrast Inversion | 2.7 ±0.5 | 2.6 ±0.3 | 6.9 ±0.5 | 5.8 ±0.2 | 4.5 ±0.4 | 3.9 ±0.6 | 7.1 ±0.6 | 5.5 ±0.4 |
| Noise (low) | 44.3 ±1.5 | 35.9 ±0.7 | 43.0 ±0.9 | 37.9 ±0.9 | 41.3 ±4.0 | 40.4 ±1.7 | 42.1 ±1.5 | 37.5 ±0.6 |
| Noise (high) | 42.2 ±1.2 | 34.9 ±1.0 | 24.4 ±2.4 | 32.5 ±2.3 | 37.1 ±5.5 | 38.0 ±1.5 | 21.4 ±2.1 | 33.0 ±1.1 |
| Gamma (darken) | 41.0 ±1.6 | 33.8 ±0.4 | 42.3 ±1.9 | 36.5 ±0.9 | 39.5 ±4.1 | 38.9 ±2.1 | 40.8 ±1.6 | 36.0 ±0.6 |
| Gamma (lighten) | 43.5 ±1.5 | 34.4 ±1.0 | 44.0 ±1.0 | 38.2 ±0.9 | 41.0 ±3.8 | 39.9 ±1.6 | 42.6 ±1.6 | 37.4 ±0.6 |

| Model \ Corruption | Train Aug. = NAff | | | | Train Aug. = All | | | |
|---|---|---|---|---|---|---|---|---|
| | Stand. | SEq | SEqSI | AffEq | Stand. | SEq | SEqSI | AffEq |
| Original | 42.4 ±1.6 | 31.7 ±4.5 | 44.2 ±1.5 | 33.8 ±1.2 | 37.6 ±4.5 | 34.1 ±2.3 | **45.7** ±0.9 | 35.2 ±0.6 |
| **Affine transformations (Aff.)** | | | | | | | | |
| Shift | 42.2 ±1.7 | 31.7 ±4.7 | 44.2 ±1.5 | 33.8 ±1.2 | 37.6 ±4.5 | 34.0 ±2.4 | **45.7** ±0.9 | 35.2 ±0.6 |
| Scale (< 1) | 42.4 ±1.8 | 31.7 ±4.5 | 44.2 ±1.5 | 33.8 ±1.2 | 37.7 ±4.5 | 34.1 ±2.3 | **45.7** ±0.9 | 35.2 ±0.5 |
| Scale (> 1) | 42.2 ±1.7 | 31.7 ±4.5 | 44.2 ±1.5 | 33.8 ±1.2 | 37.6 ±4.5 | 34.1 ±2.3 | **45.7** ±0.9 | 35.2 ±0.6 |
| Affine | 42.3 ±1.6 | 31.7 ±4.7 | 44.2 ±1.5 | 33.8 ±1.2 | 37.5 ±4.5 | 34.0 ±2.2 | **45.7** ±0.9 | 35.2 ±0.6 |
| **Non-Affine transformations (NAff.)** | | | | | | | | |
| Shift saturated | 42.4 ±1.8 | 31.6 ±4.5 | 44.0 ±1.5 | 33.7 ±1.2 | 37.7 ±4.4 | 34.1 ±2.2 | **45.5** ±0.9 | 35.3 ±0.4 |
| Scale (> 1) saturated | 42.3 ±1.6 | 31.6 ±4.6 | 44.1 ±1.4 | 33.8 ±1.2 | 37.4 ±4.3 | 34.0 ±2.3 | **45.5** ±0.9 | 35.2 ±0.7 |
| Affine saturated | 32.2 ±1.5 | 23.4 ±3.5 | **35.1** ±1.2 | 27.7 ±0.9 | 26.9 ±3.8 | 22.6 ±1.9 | 35.0 ±1.0 | 28.5 ±0.6 |
| Spatially-varying Affine | 42.5 ±1.6 | 31.7 ±4.7 | 44.2 ±1.5 | 33.6 ±1.1 | 37.8 ±4.5 | 34.1 ±2.4 | **45.7** ±0.9 | 35.2 ±0.6 |
| Contrast Inversion | **40.0** ±1.8 | 29.4 ±4.2 | 22.4 ±1.9 | 12.4 ±0.7 | 31.4 ±6.8 | 29.8 ±3.0 | 16.1 ±0.6 | 10.5 ±0.2 |
| Noise (low) | 42.4 ±1.8 | 31.7 ±4.6 | 44.1 ±1.6 | 33.7 ±1.1 | 37.7 ±4.5 | 34.1 ±2.3 | **45.7** ±0.9 | 35.3 ±0.6 |
| Noise (high) | 42.0 ±1.6 | 30.8 ±4.8 | 43.3 ±1.7 | 33.2 ±1.0 | 36.2 ±4.9 | 32.8 ±1.8 | **44.2** ±1.5 | 34.5 ±0.5 |
| Gamma (darken) | 41.8 ±1.4 | 30.6 ±4.5 | 42.8 ±1.4 | 32.7 ±1.0 | 36.2 ±4.6 | 32.5 ±2.1 | **44.2** ±1.0 | 34.1 ±0.7 |
| Gamma (lighten) | 42.3 ±1.5 | 31.8 ±4.8 | 43.9 ±1.3 | 33.7 ±1.0 | 37.6 ±4.6 | 33.9 ±2.2 | **45.4** ±0.8 | 35.3 ±0.3 |

Table 16: **Robustness to 'moderate' photometric corruptions on the Stanford Cars test set.** Test accuracy (%) (mean ± std over 5 seeds) of four ResNet-18 architectures. Models are grouped by the training augmentation strategy used: none (∅), affine (**Aff.**), non-affine (**NAff.**), or all combined (**All**). Each row corresponds to a different perturbation applied at evaluation. Within each training strategy (column group), the best result is highlighted in gray. The overall best accuracy for each perturbation is in **bold**. Models were trained for 200 epochs.

| Model / Corruption | Train Aug. = ∅ | | | | Train Aug. = Aff | | | |
|---|---|---|---|---|---|---|---|---|
| | Stand. | SEq | SEqSI | AffEq | Stand. | SEq | SEqSI | AffEq |
| Original | 27.3 ±3.4 | 23.3 ±2.7 | 33.1 ±2.7 | **44.8 ±1.4** | 19.9 ±2.4 | 26.7 ±0.7 | 33.1 ±2.1 | 41.1 ±1.1 |
| **Affine transformations (Aff.)** | | | | | | | | |
| Shift | 14.0 ±2.3 | 12.4 ±1.2 | 33.1 ±2.7 | **44.8 ±1.4** | 19.7 ±2.3 | 26.3 ±0.7 | 33.1 ±2.1 | 41.1 ±1.1 |
| Scale (< 1) | 25.7 ±3.1 | 23.3 ±2.7 | 33.1 ±2.7 | **44.8 ±1.4** | 19.5 ±2.3 | 26.7 ±0.7 | 33.1 ±2.1 | 41.1 ±1.1 |
| Scale (> 1) | 26.4 ±3.4 | 23.3 ±2.7 | 33.1 ±2.7 | **44.8 ±1.4** | 19.8 ±2.2 | 26.7 ±0.7 | 33.1 ±2.1 | 41.1 ±1.1 |
| Affine | 15.6 ±2.4 | 14.1 ±1.3 | 33.1 ±2.7 | **44.8 ±1.4** | 19.4 ±2.2 | 26.5 ±0.6 | 33.1 ±2.1 | 41.1 ±1.1 |
| **Non-Affine transformations (NAff.)** | | | | | | | | |
| Shift saturated | 19.9 ±2.9 | 17.4 ±1.8 | 30.4 ±2.4 | **41.9 ±1.1** | 18.5 ±2.1 | 25.0 ±0.6 | 30.5 ±2.1 | 38.5 ±0.9 |
| Scale (> 1) saturated | 24.1 ±3.3 | 20.5 ±2.5 | 29.0 ±2.4 | **40.6 ±1.1** | 17.6 ±1.9 | 24.1 ±0.8 | 29.3 ±2.0 | 37.0 ±1.1 |
| Affine saturated | 2.7 ±0.3 | 2.8 ±0.3 | 20.1 ±1.8 | 30.3 ±1.0 | 11.3 ±1.0 | 15.9 ±0.9 | 21.0 ±1.6 | 27.6 ±0.7 |
| Spatially-varying Affine | 13.7 ±2.2 | 12.5 ±1.2 | 32.9 ±2.8 | **44.3 ±1.2** | 19.4 ±2.3 | 26.3 ±0.8 | 32.8 ±2.0 | 40.5 ±1.0 |
| Contrast Inversion | 0.5 ±0.1 | 0.5 ±0.0 | 1.2 ±0.1 | 7.5 ±1.4 | 0.9 ±0.1 | 1.2 ±0.2 | 1.2 ±0.2 | 6.3 ±0.7 |
| Noise (low) | 27.3 ±3.4 | 23.4 ±2.7 | 32.9 ±2.8 | **44.6 ±1.3** | 19.9 ±2.4 | 26.6 ±0.7 | 32.8 ±2.3 | 40.9 ±1.1 |
| Noise (high) | 19.7 ±2.4 | 16.9 ±2.2 | 16.6 ±4.0 | 33.4 ±1.0 | 13.4 ±1.0 | 19.2 ±0.7 | 16.5 ±3.6 | 31.4 ±1.6 |
| Gamma (darken) | 19.1 ±2.6 | 15.1 ±2.0 | 26.6 ±2.3 | **37.5 ±1.2** | 16.3 ±1.9 | 21.3 ±0.7 | 26.6 ±1.9 | 34.3 ±0.6 |
| Gamma (lighten) | 23.6 ±3.3 | 20.2 ±1.8 | 32.2 ±2.6 | **44.2 ±1.2** | 19.0 ±2.1 | 26.2 ±0.7 | 32.2 ±2.4 | 40.5 ±1.2 |

| Model / Corruption | Train Aug. = NAff | | | | Train Aug. = All | | | |
|---|---|---|---|---|---|---|---|---|
| | Stand. | SEq | SEqSI | AffEq | Stand. | SEq | SEqSI | AffEq |
| Original | 13.5 ±1.4 | 22.3 ±1.8 | 35.0 ±2.3 | 40.3 ±1.8 | 14.6 ±1.0 | 23.9 ±2.7 | 37.0 ±1.5 | 40.2 ±0.4 |
| **Affine transformations (Aff.)** | | | | | | | | |
| Shift | 12.7 ±1.3 | 21.1 ±1.6 | 35.0 ±2.3 | 40.3 ±1.8 | 14.2 ±0.9 | 23.2 ±2.5 | 37.0 ±1.5 | 40.2 ±0.4 |
| Scale (< 1) | 12.7 ±1.3 | 22.3 ±1.8 | 35.0 ±2.3 | 40.3 ±1.8 | 14.2 ±0.8 | 23.9 ±2.7 | 37.0 ±1.5 | 40.2 ±0.4 |
| Scale (> 1) | 13.0 ±1.3 | 22.3 ±1.8 | 35.0 ±2.3 | 40.3 ±1.8 | 14.3 ±1.0 | 23.9 ±2.7 | 37.0 ±1.5 | 40.2 ±0.4 |
| Affine | 12.2 ±1.2 | 21.0 ±1.6 | 35.0 ±2.3 | 40.3 ±1.8 | 14.0 ±0.9 | 23.1 ±2.5 | 37.0 ±1.5 | 40.2 ±0.4 |
| **Non-Affine transformations (NAff.)** | | | | | | | | |
| Shift saturated | 12.8 ±1.2 | 20.9 ±1.6 | 33.3 ±2.4 | 38.1 ±2.0 | 13.8 ±0.9 | 22.6 ±2.6 | 35.1 ±1.3 | 37.9 ±0.4 |
| Scale (> 1) saturated | 12.1 ±1.2 | 20.6 ±1.7 | 33.2 ±2.6 | 38.0 ±1.9 | 13.1 ±0.9 | 21.9 ±2.6 | 35.1 ±1.5 | 37.8 ±0.3 |
| Affine saturated | 9.3 ±1.1 | 16.2 ±1.6 | 26.8 ±2.1 | **31.0 ±2.1** | 9.9 ±0.8 | 17.1 ±2.2 | 28.2 ±1.2 | 30.1 ±0.4 |
| Spatially-varying Affine | 12.3 ±1.3 | 21.0 ±1.7 | 34.9 ±2.4 | 39.8 ±2.0 | 13.9 ±1.0 | 22.9 ±2.6 | 36.7 ±1.4 | 40.0 ±0.6 |
| Contrast Inversion | 11.6 ±1.3 | 18.9 ±1.4 | 24.6 ±2.8 | **32.4 ±1.4** | 11.2 ±1.0 | 19.1 ±2.1 | 18.9 ±2.4 | 27.4 ±0.5 |
| Noise (low) | 13.4 ±1.3 | 22.3 ±1.8 | 35.0 ±2.4 | 40.0 ±1.8 | 14.5 ±1.0 | 23.9 ±2.7 | 37.0 ±1.5 | 39.9 ±0.4 |
| Noise (high) | 12.4 ±1.0 | 20.5 ±1.7 | 32.8 ±2.4 | **35.5 ±1.5** | 13.2 ±0.8 | 21.8 ±2.2 | 34.0 ±1.2 | 35.1 ±0.6 |
| Gamma (darken) | 12.1 ±1.2 | 18.3 ±1.5 | 29.4 ±2.3 | 33.7 ±1.8 | 12.9 ±1.1 | 19.6 ±2.2 | 31.1 ±1.1 | 33.6 ±0.3 |
| Gamma (lighten) | 13.1 ±1.3 | 22.1 ±1.7 | 34.8 ±2.4 | 40.3 ±1.9 | 14.3 ±1.1 | 23.6 ±2.7 | 36.8 ±1.5 | 40.2 ±0.3 |

## C.10 TRANSFERT LEARNING FROM STANDARD MODEL

**Experimental Setup.** The transfer learning experiment was conducted on the Stanford Cars dataset (Krause et al., 2013). We used a ResNet-18 architecture with weights pre-trained on ImageNet-1K (`ResNet18_Weights.IMAGENET1K_V1` from PyTorch). The goal was to assess the practicality and compatibility of *SEqSI* architectures when fine-tuned from a standard model.

**Model Adaptation from Pre-trained Weights.** Since the pre-trained model is a *Standard* architecture, adapting it for our constrained models required significant modifications, creating a structural handicap for *SEq*, *SEqSI*, and *AffEq*.

- *Standard*: The pre-trained weights were loaded directly. Only the final fully-connected layer was replaced with a new one, randomly initialized, to match the 196 classes of the Stanford Cars dataset.

- *SEq*: All bias parameters were removed from the pre-trained layers to enforce scale-equivariance. The final classifier was also replaced.

- *SEqSI*: In addition to removing all biases like in *SEq*, the weights of the first convolutional layer were re-projected to be zero-sum. This was done by subtracting the mean from the pre-trained weight tensor of that layer, thus enforcing shift-invariance at the input.

- *AffEq*: This model required the most drastic changes. All biases were removed, all ReLU activation functions were replaced with 'SortPool', and the weights of every convolutional layer were re-projected to sum to one. These extensive modifications, which create a significant departure from the original learned representations, are the likely cause of the observed training divergence.

**These adaptations highlight a key practical challenge: leveraging standard pre-trained models with architectures that have specific structural constraints.**

**Fine-Tuning Protocol.** All models were fine-tuned for 200 epochs on the Stanford Cars training set using a batch size of 32. The learning rate was set to 1e-2 for the *Standard*, *SEq*, and *SEqSI* models, and decayed using a cosine annealing schedule. For the *AffEq* model, a lower learning rate of 1e-3 was necessary to prevent training divergence. For all models, the entire network was fine-tuned (no layers were frozen). The different convergence speeds noted in the main text (e.g., 10 epochs for *Standard* vs. 50 for *SEq/SEqSI* to pass 60% accuracy) were observed during this process, with *AffEq* failing to converge to a meaningful accuracy level.

**Results.** The fine-tuning results, presented in Table 17, demonstrate the practical viability of *SEqSI*. Despite the initial handicap from modifying the pre-trained weights, *SEqSI* proves its compatibility with transfer learning by achieving competitive performance. It reaches an accuracy of 82.3% on clean data, close to the advantaged *Standard* model (85.6%). Crucially, it provides certified robustness to all affine corruptions where the performance of *Standard* model collapses (e.g., to 60.6% under a simple shift), while also showing superior generalization on several challenging non-affine corruptions like "Spatially-Varying Affine" (82.1% vs. 67.2%). In contrast, the *AffEq* architecture fails to converge effectively in this transfer learning scenario, likely due to the extensive modifications required to adapt the pre-trained weights, resulting in poor performance across all evaluation conditions.

**Conclusion.** **This study demonstrates that *SEqSI* is a practical and viable architecture for transfer learning. It can successfully adapt features from standard, pre-trained models.** Despite a structural modification leading to slower initial convergence, it achieves competitive accuracy while preserving its architectural properties. In contrast, the *AffEq* architecture proves too rigid for this common and important workflow. This positions *SEqSI* as a principled yet pragmatic choice for fine-tuning pre-trained models in applications where photometric robustness is critical.

Table 17: **Robustness in a transfer learning scenario on Stanford Cars.** Test accuracy (%) of four ResNet-18 architectures fine-tuned from an ImageNet pre-trained model. Results are the mean $\pm$ std over 5 seeds. The best result in each group is in gray; the overall best is in **bold**.

| Model
Eval.
Perturb. | Train Aug. = ∅ | | | |
|---|---|---|---|---|
| | Stand. | SEq | SEqSI | AffEq |
| Original | 85.6 ±0.2 | 82.1 ±0.4 | 82.3 ±0.1 | 37.3 ±1.5 |
| **Affine transformations (Aff.)** | | | | |
| Shift | 60.6 ±0.9 | 78.6 ±1.1 | 82.3 ±0.1 | 37.3 ±1.5 |
| Scale ($< 1$) | 84.1 ±0.3 | 82.1 ±0.4 | 82.3 ±0.1 | 37.3 ±1.5 |
| Scale ($> 1$) | 66.2 ±2.3 | 82.1 ±0.4 | 82.3 ±0.1 | 37.3 ±1.5 |
| Affine | 49.4 ±0.9 | 78.8 ±1.0 | 82.3 ±0.1 | 37.3 ±1.5 |
| **Non-Affine transformations (NAff.)** | | | | |
| Shift saturated | 81.1 ±0.3 | 79.2 ±0.5 | 80.2 ±0.4 | 35.0 ±1.4 |
| Scale ($> 1$) saturated | 81.1 ±0.3 | 79.2 ±0.8 | 79.3 ±0.3 | 33.8 ±1.4 |
| Affine saturated | 50.1 ±1.2 | 51.0 ±1.0 | 66.5 ±0.9 | 24.8 ±0.8 |
| Spatially-varying Affine | 67.2 ±1.3 | 78.4 ±0.9 | 82.1 ±0.1 | 36.2 ±1.6 |
| Contrast Inversion | 0.6 ±0.0 | 8.5 ±1.9 | 31.7 ±2.4 | 2.4 ±0.2 |
| Noise (low) | 84.0 ±0.3 | 81.9 ±0.4 | 81.9 ±0.3 | 36.1 ±1.5 |
| Noise (high) | 13.2 ±2.2 | 66.6 ±1.7 | 46.8 ±2.6 | 3.7 ±0.6 |
| Gamma (darken) | 81.1 ±0.4 | 76.3 ±0.7 | 77.3 ±0.5 | 30.4 ±1.2 |
| Gamma (lighten) | 84.5 ±0.3 | 81.2 ±0.4 | 81.9 ±0.2 | 36.6 ±1.4 |

# D  EXPERIMENTS IN MACROMOLECULE CLASSIFICATION - APPENDIX

## D.1  DETAILS ON MODELS IMPLEMENTATION AND DATA PROCESSING

The original CZI challenge involves both localizing and identifying macromolecules. We simplify this to a pure classification task on pre-extracted patches. This simplification is used below since: the full localization-and-identification pipeline is highly complex and often requires a separate, non-trivial post-processing algorithm for connected component analysis. By decoupling the tasks, we separate the identification module, allowing for a more direct and controlled evaluation of our models robustness. This is particularly important for assessing the performance of architectures like *SEqSI* on the highly corrupted and variable cryo-ET images, enabling to draw clean conclusions about the benefits of our architectural properties under such challenging conditions.

**Dataset Details.**  We use the dataset from the CZI cryo-ET challenge (Harrington et al., 2024), which consists of 7 tomograms containing 1,269 annotated particle centroids belonging to 6 different macromolecule classes. From these annotations, we extract 3D patches of size $64 \times 64 \times 64$ centered on each particle. The dataset presents two main sources of variation. First, the challenge provides four different preprocessing pipelines for each tomogram, resulting in significant domain shifts for the same underlying data, as illustrated in Supp. Fig. 6. Second, there is substantial variability between the tomograms themselves due to different acquisition conditions (e.g. samples thickness, cryo quality...). Supp. Fig. 7, we show an example of this inter-tomogram variation. In Supp. Table 18, we summarize the voxel intensity distributions for each class across the splits, quantitatively confirming this significant variability.

These variations stem from three main sources:

- **Inter-preprocessing variability:**  The dataset provides four different preprocessing pipelines for each tomogram, creating severe domain shifts as shown in Supp. Fig. 6. All tomograms are reconstructed from tilt series using WBP (Weighted Back Projection) as the base method (Peck et al., 2024):
  – WBP: reconstructed by Weighted Back Projection (no additional processing)
  – CTF deconvolved: WBP + local Contrast Transfer Function (CTF) correction (deconvolution)
  – Denoised: WBP + DenoisET denoising
  – IsoNet Corrected: WBP + IsoNet correction (denoising / missing-wedge compensation / contrast correction)
- **Inter-tomogram and intra-class variability :** Even within a single preprocessing technique, there is substantial variability between different tomograms due to varying acquisition conditions (e.g., sample thickness, ice quality), causing appearance changes for particles of the same class (see Fig. 7.

In Supp. Table 18, we quantitatively summarize the voxel intensity distributions, confirming these significant photometric shifts. For example, the minimum intensity for a virus-like particle can vary by more than a factor of two between the training set ($-15.78 \times 10^{-5}$) and the validation set ($-7.08 \times 10^{-5}$), while the median intensity for a ribosome even changes sign between training ($0.06 \times 10^{-5}$) and testing ($-0.01 \times 10^{-5}$). This complex, multi-level variation creates a realistic and challenging testbed for out-of-distribution generalization.

**Data Preprocessing and Augmentation.**  During training, we only apply geometric data augmentation to enhance model generalization. This includes: (1) random flips along each of the three spatial axes (depth, height, width), each with a 50% probability, and (2) random rotations of 0, 90, 180, or 270 degrees in the height-width plane.

**Model Architectures.**  For this task, we compare a *Standard* 3D ResNet with *SEq*, *SEqSI*, *Affeq* variant. All models are based on a 3D ResNet architecture inspired by ResNet-18 (He et al., 2015), adapted for $64 \times 64 \times 64$ input patches. They share the same overall structure but differ in their specific layer configurations to reflect their respective properties. To preserve its theoretical guarantees, the the *SEqSI* and *AffEq* models use reflection padding, whereas the *Standard* and *SEq* models use standard zero-padding.

OVERALL ARCHITECTURE.   The network begins with an initial convolutional layer, followed by four stages of residual blocks, and concludes with a classification head. The **initial layer** is a '$3 \times 3 \times 3$' convolution that maps the single-channel input to 64 feature maps. For the *SEqSI* model, this is a shift-invariant convolution ('InvConv3d'), while the *Standard* model uses a standard convolution followed by a 'BatchNorm3d' layer (in the case of *SEq*, *SEqSI* and *AffEq* no 'Batch-Norm3d' layer were used as it breaks invariant properties). In both cases, a 'LeakyReLU' activation is applied. The network then proceeds through four **residual stages**, each composed of 2 residual blocks. Stage 1 operates on 64-channel feature maps at a '$64 \times 64 \times 64$' resolution. Stages 2, 3, and 4 double the channel count (to 128, 256, and 512, respectively) while halving the spatial resolution at each step (to '$32 \times 32 \times 32$','$16 \times 16 \times 16$',and '$8 \times 8 \times 8$'). The **classification head** processes the final '$8 \times 8 \times 8 \times 512$' feature map with a global average pooling layer to produce a 512-dimensional vector. This vector is then projected to 6 output logits by a fully connected layer, which includes a bias term for the *Standard* model but not for the *SEq*, *SEqSI* and *AffEq* models. (for recall, *SEq*, *SEqSI* and *AffEq* do not use any bias term, as it breaks scale equivariant properties)

RESIDUAL BLOCK STRUCTURE.   Each residual block contains a main path with two '3x3x3' bias-free convolutions and a shortcut connection. The data flow within the main path is 'Conv $\rightarrow$ BN (if Standard) $\rightarrow$ LeakyReLU $\rightarrow$ Conv $\rightarrow$ BN (if Standard)'. The output of this path is added to the shortcut connection's output, and a final 'LeakyReLU' activation is applied to the sum. When downsampling is required (at the beginning of stages 2, 3, and 4), the shortcut connection uses a '3x3x3' convolution with a stride of 2 to match the main path's output dimensions.

**Training Details.**   All models are trained using the AdamW optimizer with an initial learning rate of 1e-4 and the cross-entropy loss function. The learning rate is adjusted at each step using a cosine annealing schedule with a minimum rate of 1e-6. The batch size was set to 8. The number of training epochs was adapted to the training data: 300 epochs for the main experiment on the noisy *WBP* data, and 200 epochs for the supplementary experiment on the cleaner *Denoised* data. All models were implemented in PyTorch Lightning and trained on a single NVIDIA A40 GPU using 'bf16-mixed' precision. To ensure robust and reproducible results, we repeated each experiment with 5 distinct random seeds (5, 7, 42, 137, 181). For each run, the final model was selected by taking the checkpoint that achieved the highest validation accuracy.

**Supplementary Results: Training on Denoised Data.**   We conducted a supplementary experiment where models were trained exclusively on the cleaner *Denoised* data. The results, presented in Table 20, show that while the generalization gap is smaller when training on less noisy data, the *SEqSI* model still significantly outperforms the *Standard* baseline on two of the three out-of-distribution domains. This confirms that the choice of input data influences the out-of-distribution (OOD) capabilities, but the architectural properties of *SEqSI* provides a practical advantage.

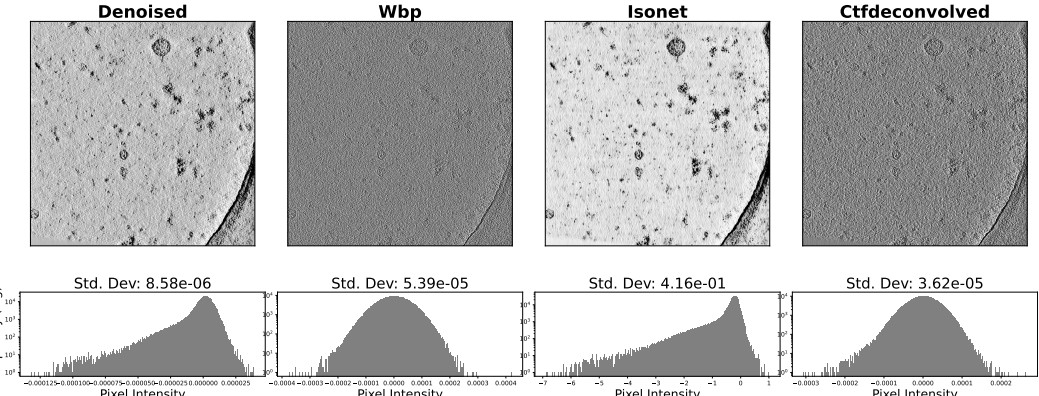

Figure 6: **Domain Shift Induced by Tomogram Preprocessing.** A representative central slice from a single raw tomogram is shown after four distinct preprocessing pipelines. The top row presents the visual appearance, while the bottom row displays the corresponding pixel intensity histograms (log-frequency scale). Each histogram is annotated with the standard deviation of its pixel intensities ($\sigma_I$), a direct measure of contrast, calculated as $\sigma_I = \sqrt{\frac{1}{N}\sum_{i=1}^{N}(I_i - \bar{I})^2}$, where $\bar{I}$ is the mean pixel intensity. The stark variations in contrast and pixel distributions illustrate the severe domain shift a classification model is exposed to, motivating the development of robust methods.

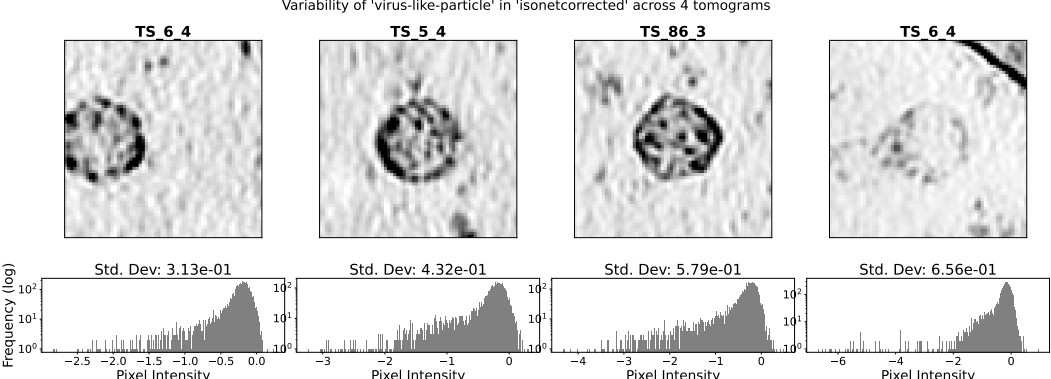

Figure 7: **Intra-Class Variability within a Single Preprocessing Domain.** This figure illustrates the appearance diversity for a single particle class (here, virus-like-particle) extracted from different tomograms, all processed with the same pipeline (isonetcorrected). The samples are selected to span the full spectrum of observed contrast, from lowest (left) to highest (right). The top row shows a central slice of each particle, while the bottom row presents its pixel intensity histogram, annotated with its standard deviation ($\sigma_I$). This significant intra-class variability, even within a supposedly homogeneous domain, poses a fundamental challenge for the generalization of classification models.

Table 18: Voxel intensity distribution per macromolecule across the training, validation, and test tomograms. All values are scaled by $10^5$. The statistics highlight the significant photometric shifts between the different data splits.

| Macromolecule | Dataset | Voxel Intensity ($\times 10^{-5}$) | | | | |
| | | Min | Q1 (25%) | Median | Q3 (75%) | Max |
| --- | --- | --- | --- | --- | --- | --- |
| 'apo-ferritin' | Train | -14.17 | -0.29 | 0.10 | 0.45 | 4.32 |
| | Val | -7.79 | -0.27 | 0.08 | 0.41 | 3.15 |
| | Test | -8.94 | -0.33 | 0.07 | 0.43 | 3.63 |
| 'beta-amylase' | Train | -9.47 | -0.24 | 0.12 | 0.45 | 3.67 |
| | Val | -8.21 | -0.26 | 0.09 | 0.41 | 2.88 |
| | Test | -7.73 | -0.29 | 0.08 | 0.42 | 2.99 |
| 'beta-galactosidase' | Train | -11.34 | -0.26 | 0.10 | 0.44 | 3.63 |
| | Val | -7.35 | -0.26 | 0.10 | 0.42 | 3.48 |
| | Test | -9.04 | -0.26 | 0.11 | 0.44 | 3.63 |
| 'ribosome' | Train | -11.04 | -0.46 | 0.06 | 0.46 | 3.75 |
| | Val | -8.39 | -0.48 | 0.03 | 0.43 | 3.48 |
| | Test | -9.89 | -0.59 | -0.01 | 0.41 | 3.63 |
| 'thyroglobulin' | Train | -15.04 | -0.25 | 0.11 | 0.44 | 3.84 |
| | Val | -8.29 | -0.27 | 0.09 | 0.42 | 3.15 |
| | Test | -9.04 | -0.30 | 0.08 | 0.42 | 3.63 |
| 'virus-like-particle' | Train | -15.78 | -0.30 | 0.09 | 0.44 | 3.98 |
| | Val | -7.08 | -0.30 | 0.07 | 0.40 | 3.15 |
| | Test | -8.46 | -0.42 | 0.03 | 0.40 | 3.29 |

Table 19: Performance comparison on CZI test tomograms. Models were trained on *WBP* data, evaluated over 5 random seeds, for 300 epochs. Results are reported as mean $\pm$ std. Best results for each metric are in **bold**.

| Data Type | Standard Acc. (%) | SEqSI Acc. (%) |
| --- | --- | --- |
| WBP (in-distribution) | **87.2 $\pm$ 1.7** | 85.2 $\pm$ 4.0 |
| Denoised | 22.4 $\pm$ 7.4 | **74.5 $\pm$ 4.8** |
| IsoNet Corrected | 16.0 $\pm$ 1.6 | **73.2 $\pm$ 4.7** |
| CTF Deconvolved | 48.9 $\pm$ 12.3 | **66.5 $\pm$ 10.9** |

Table 20: Performance comparison on CZI test tomograms. Models were trained on *Denoised* data, evaluated with 5 seeds, for 200 epochs. Results are reported as mean $\pm$ std. Best results for each metric are in **bold**.

| Data Type | Standard Acc. (%) | SEqSI Acc. (%) |
| --- | --- | --- |
| Denoised (in-distribution) | 88.0 $\pm$ 1.9 | **88.6 $\pm$ 5.0** |
| IsoNet Corrected | 16.7 $\pm$ 0.0 | **76.7 $\pm$ 9.7** |
| CTF Deconvolved | 16.4 $\pm$ 0.4 | **21.9 $\pm$ 4.7** |
| WBP | **17.1 $\pm$ 1.1** | 16.9 $\pm$ 0.4 |

## D.2 ARCHITECTURAL PROPERTIES VS. NORMALIZATION PREPROCESSING, FOR DOMAIN SHIFT ROBUSTNESS

**Motivation.** While the min-max normalization provides invariance to global affine photometric transformations B.4, this experiment investigates the limitations of min-max normalization in handling domain shifts, contrasting it with the inherent robustness of *SEqSI* and *AffEq* architectures.

**Experimental Setup.** A *Standard* model trained and test with the min-max normalization is compared against the *Standard*, *SEq*, *AffEq* and *SEqSI* models trained and test without normalization.

**Results and Analysis.** Table 21 demonstrates that the performance of MinMax normalized *Standard* model drops significantly on out-of-distribution domains (16.98% on *IsoNet Corrected* and 17.84% on *Denoised*) compared to 84.81% on *WBP*. **This indicates that the min-max normalization does not ensure generalization across different data distributions.**

**Conclusion.** The experiment highlights the limitations of pre-processing for robustness. Embedding invariance and equivariance directly into the network architecture, as with *SEqSI* and *AffEq*, is more effective for handling domain shifts, enabling better generalization to unseen data domains.

Table 21: Performance comparison on test set. Standard baseline is trained with and without the MinMax normalization, during both train and test. Models were trained on *WBP* data. Results are reported as mean accuracy (Acc.) $\pm$ std. Best results for each metric are in **bold**.

| Train/Test Norm. | *MinMax* | Ø | | | |
|---|---|---|---|---|---|
| **Data Type** | **Standard** Acc. (%) | **Standard** Acc. (%) | **SEq** Acc. (%) | **SEqSI** Acc. (%) | **AffEq** Acc. (%) |
| WBP (in-distrib.) | $84.81 \pm 1.87$ | $87.17 \pm 1.50$ | **87.69$\pm$0.82** | $85.15\pm3.59$ | $79.26 \pm 4.04$ |
| CTF Deconvolved | $41.34 \pm 11.20$ | $48.91 \pm 11.04$ | $65.21\pm6.08$ | $66.51 \pm 9.75$ | **79.07 $\pm$ 5.04** |
| Denoised | $17.84 \pm 1.96$ | $22.36 \pm 6.59$ | $28.42\pm7.68$ | **74.53$\pm$4.25** | $61.79 \pm 5.97$ |
| IsoNet Corrected | $16.98 \pm 0.63$ | $15.95 \pm 1.44$ | $16.67\pm0.00$ | **73.21$\pm$4.18** | $46.37 \pm 5.98$ |

# E  EXPERIMENTS IN OBJECT LOCALIZATION - APPENDIX

## E.1  DETAILS ON THEORETICAL CONTRIBUTIONS

### E.1.1  PROOF OF INVARIANCE FOR OUR SCORE MAP GENERATION APPROACH

For *AffEq* network:

$$
\begin{aligned}
\mathcal{Z}(f(T_{\lambda,\mu}(\boldsymbol{x}))) = \mathcal{Z}(T_{\lambda,\mu}(f(\boldsymbol{x}))) &\quad \text{by equivariance properties} \\
&= \frac{\lambda f(\boldsymbol{x}) + \mu - \mathbb{E}[\lambda f(\boldsymbol{x}) + \mu]}{\sigma(\lambda f(\boldsymbol{x}) + \mu)} \\
&= \frac{\lambda f(\boldsymbol{x}) + \mu - \lambda \mathbb{E}[f(\boldsymbol{x})] - \mu}{\lambda \sigma(f(\boldsymbol{x}))} \quad \text{by mean and std properties} \\
&= \mathcal{Z}(f(\boldsymbol{x})).
\end{aligned}
$$

For *SEqSI* network:

$$
\begin{aligned}
\mathcal{Z}(f(T_{\lambda,\mu}(\boldsymbol{x}))) = \mathcal{Z}(T_{\lambda,0}(f(\boldsymbol{x}))) &\quad \text{by equivariance and invariance properties} \\
&= \frac{\lambda f(\boldsymbol{x}) - \mathbb{E}[\lambda f(\boldsymbol{x})]}{\sigma(\lambda f(\boldsymbol{x}))} \\
&= \frac{\lambda f(\boldsymbol{x}) - \lambda \mathbb{E}[f(\boldsymbol{x})]}{\lambda \sigma(f(\boldsymbol{x}))} \quad \text{by mean and std properties} \\
&= \mathcal{Z}(f(\boldsymbol{x})).
\end{aligned}
$$

### E.1.2  CONSISTENCY IN RESULTS REGARDING THE CHOICE OF THE NORMALIZATION POST NETWORK

Common normalization functions include min-max scaling to $[0,1]$ ($T_1 : \boldsymbol{x} \mapsto \frac{\boldsymbol{x} - \min(\boldsymbol{x})}{\max(\boldsymbol{x}) - \min(\boldsymbol{x})}$), scaling to $[-1,1]$ ($T_2 : \boldsymbol{x} \mapsto 2T_1(\boldsymbol{x}) - 1$), and standardization ($T_3 : \boldsymbol{x} \mapsto \frac{\boldsymbol{x} - \mathbb{E}[\boldsymbol{x}]}{\sigma(\boldsymbol{x})}$).

For the centroid detection task, the strategy we apply imply to normalize the output of the *AffEq* and *SEqSI* networks. We have decided to apply a standardization strategy which is equivalent to function $T_3$. More precisely the standardization of the output corresponds to the affine normalization $T_{1/\sigma(f(\boldsymbol{x})), \mathbb{E}[f(\boldsymbol{x})]/\sigma(f(\boldsymbol{x}))}(f(\boldsymbol{x}))$, whose inverse transformation is the normalization $T_{\sigma(f(\boldsymbol{x})), -E[f(\boldsymbol{x})]}$ denoted $h_3$.

We show that rescaling the outputs using $T_1$ would lead to an equivalent thresholding strategy and an identical proof can be given for $T_2$.

The normalization in $[0,1]$ using $T_1$ corresponds to the affine normalization $T_{1/(\max - \min)(f(\boldsymbol{x})), \min(f(\boldsymbol{x}))/(\max - \min)(f(\boldsymbol{x}))}(f(\boldsymbol{x}))$, whose inverse transformation is the normalization $T_{(\max - \min)(f(\boldsymbol{x})), -\min(f(\boldsymbol{x}))}$ denoted $h_1$.

We consider that a network $f$ is applied on an image $\boldsymbol{x}$ giving $\boldsymbol{y} = f(\boldsymbol{x})$.

Let $\gamma_3$ be a threshold that separates the regions of interest from the background on the normalized output channel of the network using $T_3$.

Hence, the threshold $\gamma_1 = T_1(h_3(\gamma_3))$ provides the exact same separation for the normalization $T_1$.

$$
T_3(f(\boldsymbol{x})) \leq \gamma_3 \iff f(\boldsymbol{x}) \leq h_3(\gamma_3) \iff T_1(f(\boldsymbol{x})) \leq T_1(h_3(\gamma_3)) = \gamma_1
$$

This holds because all the normalizations studied are increasing functions.

Experimentally, we obtain our thresholds by optimizing a given metric. So, if $\gamma_3$ is the optimal threshold for normalization $T_3$, then $\gamma_1$ is optimal for $T_1$ and that will be the chosen threshold for that normalization.

Moreover, the demonstration provided in Supp. E.1.1 holds for $T_1$ and $T_2$ using $\min$ and $\max$ properties instead of mean and std properties.

### E.1.3 INTUITION ON NORMALIZATION BENEFITS IN ALTERATION ROBUSTNESS

We can wonder why our strategy is better than the classical approach as we still apply a normalization after the network and it is only moved from "before the network" to "after the network". In the standard approach, the input data is scaled to fit a chosen range or distribution, then fed to the network to provide the output channels and score map. But some photometric alterations can affect the normalization and modify the range of the really meaningful data, leading to wrong prediction from the network as the data is not properly scaled/shifted accordingly to the training data. For example, if an image contains very bright artifacts, the normalization in a specific range will be affected and the rest of the data will not get the same value after normalization that it would have without artifacts. In our paradigm, the network being invariant or equivariant, the learned features do not rely on the input range and so does the output of the network. It is then robust to various transformations that could alter the normalization range in the classical approach. This output is then normalized to fit a desired range or distribution for the sake of interpretation, then thresholded. Applying this strategy provides robust results, as well as increased generalization.

### E.2 Details on experimental setup

#### E.2.1 Evaluated architectures

The four variants of U-Net Ronneberger et al. (2015) that we consider have similar architecture: the initial number of channels equals to 64 for 2D, 32 for 3D models, maximum depth of 4, 3 convolution with kernel of size $3 \times 3$ at each depth, max-pooling dividing the dimension of channel by 2 along each dimension and doubling the number of channels.

The specificity of each model are the following ones:

- **Standard:** our baseline, an unconstrained CNN with standard convolutions using bias and ReLU activation functions,
- **SEq:** bias free convolutions with unconstrained kernel weights and ReLU activation functions,
- **AffEq:** bias free convolutions with kernel weights constrained to sum to 1 and *Sort-pool* activation functions,
- **SEqSI:** the first layer of the U-Net generating the 32 channels from the input image is shift invariant (weights constrained to sum to 0), the rest of the network is composed of unconstrained bias free convolution and ReLU activations.

To implement the models for the object location task, we applied the "telescopic" strategy (see App. C.5) to constraint the weights.

#### E.2.2 Object location datasets

**Data Science Bowl 2018**    The Data Science Bowl 2018 (DSB) dataset (Goodman et al., 2018) is a dataset of 2D real microscopy images from a Kaggle challenge. We focused on the subset containing 497 fluorescence images[6] used to train StarDist 2D (Schmidt et al., 2018). Each image contains dozens of objects. The set is split in 427 training images, 25 validation images, 50 testing images. The test set contain in total approximately 2500 objects representing a wide range of scenario of shape and illumination conditions that can be observed in fluorescence microscopy. Some example are shown in Supp. Fig. 8.

**3D synthetic dataset**    Our 3D synthetic dataset is composed of 10 training images, 3 validation images and 5 testing images. Each image contains approximately 400 objects which are ellipsoids with random size, orientation, intensity and the possibility to have a darker inside as it can sometimes be observed on microscopy image. Each image is blurred and Poisson-Gaussian noise was added to mimic real imaging conditions (Boulanger et al., 2010). An example of such image is shown in Supp. Fig. 9.

#### E.2.3 Definition of the accuracy metric

We can compute a distance matrix $\mathbf{D} \in \mathbb{R}^{N_{GT} \times N_{pred}}$, that stores the distance between all pairs of GT-predicted points: $\mathbf{D}_{i,j} = ||\mathbf{c}_i - \hat{\mathbf{c}}_j||_2$ is the distance between the GT location $\mathbf{c}_i$ of index $i$ and the predicted location $\hat{\mathbf{c}}_j$ of index $j$. The pairing between points is done by iteratively associating together the points that correspond to the minimum of $\mathbf{D}$ and removing the corresponding column and row to prevent the process to reassign those points. We also investigated the solution that consists in solving an assignment problem between the points sets that minimize the sum of the distances of all pairings, as it is often done for other tasks such as segmentation. But, in our case, minimizing a global term sometimes results in associating a point with another that is far away instead of one really close as it minimize the overall process. We do not want this behavior to occur (see Fig. 10), thus we keep the greedy pairing approach depicted first. For each matching between a predicted and a GT center, we know the distance $d$ in pixels (or voxels in 3D) between the two objects.

In what follows, we define:

---

[6]https://github.com/stardist/stardist/releases/download/0.1.0/dsb2018.zip

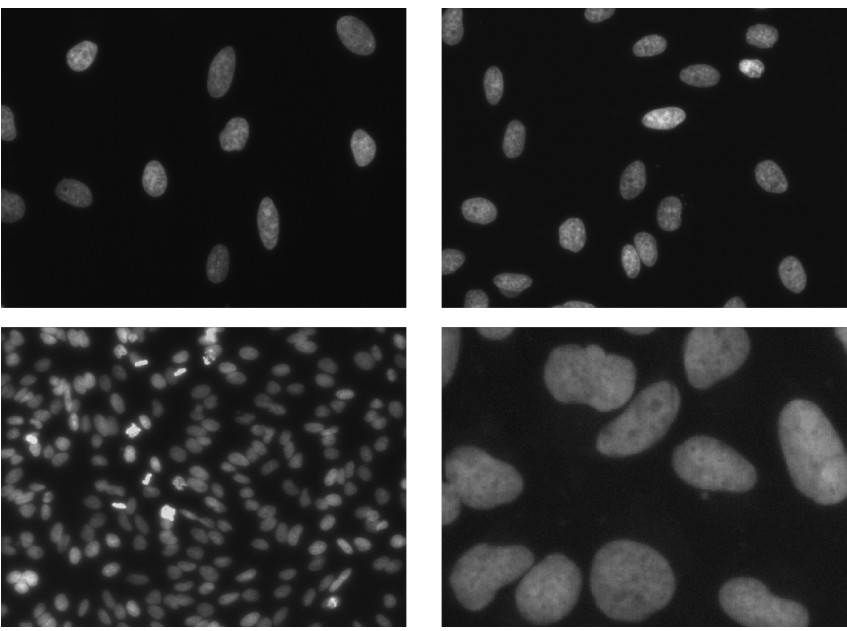

Figure 8: Examples of images from the test set of DSB dataset.

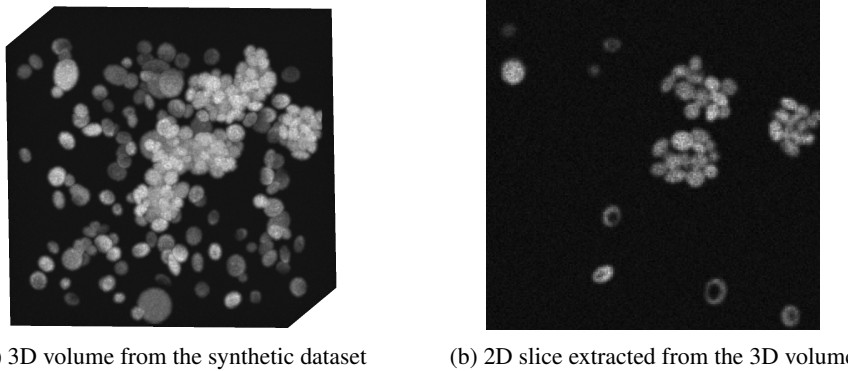

(a) 3D volume from the synthetic dataset      (b) 2D slice extracted from the 3D volume

Figure 9: Illustration of the 3D synthetic dataset mimicking fluorescence microscopy

- $TP(d)$ (True Positives): the number of matched GT and predicted centers separated by less than $d$ pixels/voxels apart;

- $FP(d)$ (False Positives): the number of predicted centers unmatched or at a distance greater than or equal to $d$ pixels/voxels from their paired GT center;

- $FN(d)$ (False Negatives): the number of GT centers unmatched or at a distance greater than or equal to $d$ pixels/voxels from their paired predicted center.

We evaluate the performances of the tested approaches using the accuracy metric:

$$acc(d) = \frac{TP(d)}{TP(d) + FN(d) + FP(d)}.$$

Eventually, our final score is provided by the area under the accuracy curve, integrated between 0 and a maximum distance $D$:

$$score = \frac{1}{D} \int_0^D acc(d)\mathrm{d}d.$$

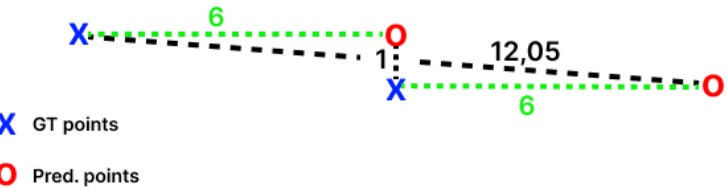

Figure 10: We illustrate the two pairings strategy between predicted and GT locations. The green pairing minimizes the global cost corresponding to the sum of all pairing distances. However, it does not pair together the two points that are at a distance of one which is the behavior that we expect (the left blue point is a FN, the right red point is a FP and center points correspond to a match). Thus, we prefer the greedy pairing approach that consider the black pairs. The very high distance pair will be filtered afterward anyway as we only consider pairings above a certain distance threshold $d$ to compute the accuracy.

In the experiments, to choose the value of $D$ that we use, we consider that a pairing for which the distance between objects is equal to the mean radius of the objects in the set can be considered as a wrong matching (the predicted point is probably outside the object or almost outside it). Thus we fix $D$ equal to the mean radius for each set. In the 2D set, the mean radius is equal to 11.99 pixels, we fix the threshold to $D = 12$. In 3D, the radius is approximately 5.53 voxels, thus we fix the distance threshold to $D = 6$. In the Tables, we report the scores for those $D$.

Table 22: **Measure of the invariance for the object localization task under various affine intensity transformations. The invariance measure is equivalent to computing the accuracy for** $d = 1$**, comparing new locations with reference locations obtained in the original range [0,1]). A value of 1 corresponds to an invariant location estimation**. Results in gray are invariant.

| | Setting | w. sigmoid and BCE | | | | w. standardization and ZMSE | | | |
| --- | --- | --- | --- | --- | --- | --- | --- | --- | --- |
| | | Standard | SEq | AffEq | SEqSI | Standard | SEq | AffEq | SEqSI |
| **Shift** | $\mu = -2$ | 0.0 | 0.0 | 0.112 | 1.0 | 0.0 | 0.001 | 1.0 | 1.0 |
| | $\mu = 0.5$ | 0.076 | 0.037 | 0.832 | 1.0 | 0.085 | 0.101 | 1.0 | 1.0 |
| | $\mu = 2$ | 0.001 | 0.0 | 0.74 | 1.0 | 0.003 | 0.002 | 1.0 | 1.0 |
| | $\mu = 10$ | 0.0 | 0.0 | 0.22 | 1.0 | 0.002 | 0.0 | 1.0 | 1.0 |
| **Scale** | $\lambda = 0.5$ | 0.264 | 0.97 | 0.959 | 0.984 | 0.365 | 1.0 | 1.0 | 1.0 |
| | $\lambda = 3.0$ | 0.157 | 0.869 | 0.897 | 0.927 | 0.238 | 1.0 | 1.0 | 1.0 |
| | $\lambda = 255$ | 0.003 | 0.014 | 0.012 | 0.012 | 0.032 | 1.0 | 1.0 | 1.0 |
| **Affine** | $\mu = -2, \lambda = 10$ | 0.029 | 0.055 | 0.526 | 0.309 | 0.016 | 0.001 | 1.0 | 1.0 |
| | $\mu = 5, \lambda = 0.1$ | 0.0 | 0.0 | 0.101 | 0.953 | 0.002 | 0.0 | 1.0 | 1.0 |
| | $\mu = 5, \lambda = 3$ | 0.001 | 0.0 | 0.67 | 0.927 | 0.003 | 0.002 | 1.0 | 1.0 |

## E.3 EXPERIMENTAL RESULTS

### E.3.1 VALIDATION OF OUR TRAINING AND INFERENCE STRATEGY TO GENERATE INVARIANT CENTROID DETECTION ON DSB2018

This experiment was conducted using 2D U-Net models. We trained the models for 1000 epochs, using batch of size 64, using AdamW optimizer with a learning rate (lr) of $10^{-4}$ and image patches of size $128 \times 128$, each patch being ranged in the range $[0, 1]$. For each approach, we selected the weights that minimized the validation loss during the training to avoid taking a model overfitting the training set. We optimized the thresholding value by maximizing the area under the accuracy curve on the validation set.

We perform a first experiment to verify empirically the theoretical invariance of our approach. We evaluated 8 models: the 4 architectures *Stand.*, *SEq*, *AffEq* and *SEqSI* using either the standard thresholding and loss (BCE) strategy or our approach based on standardization and ZMSE loss. The additional combinations: *Stand.+ZMSE*, *SEq+ZMSE*, *AffEq+BCE* and *SEqSI+BCE* were added here to show that neither the network nor the post-processing alone guarantee the invariance and both are necessary.

We evaluate the invariance by first estimating the objects locations for image scaled in $[0, 1]$ range to obtain reference locations. Then, we re-estimate the object positions for different values of scale, shift and affine normalization. We then compute an invariance measure, equivalent to the accuracy for $d = 1$ between the new locations and the locations estimated in $[0, 1]$. If the measure is not equal to 1, it is either because: i) there are missing or additional location for that transformation creating either FP or FN; ii) there are the same number of objects which are not at the same positions (as we set $d = 1$ we consider only matching of objects whose positions are distant by less than 1). Thus, an invariant location estimation corresponds to a value of 1.

Invariance results are given in Supp. Table 22. As expected, only *AffEq* and *SEqSI* are completely invariant when paired with the standardization and ZMSE loss.

We also provide additional results assessing the performances of each of these methods on this dataset. We assess each of the above mentioned methods in $[0, 1]$ and other ranges that are either a scale, a shift or both. Supp. Fig. 11 shows the performances (accuracy curve) of each method for these different ranges of normalization. On the training range, all approaches provide similar and very satisfying results with very close accuracy curves. However, we observe that, as expected, the non invariant strategy have major drops in performances for other ranges:

- Shift ($\mu = 1$), both *AffEq+ZMSE* and *SEqSI+ZMSE* provide invariant results. *SEqSI+BCE* provides invariant results too as the network is shift-invariant by itself and do not rely on the post-processing to remove the shift. Surprisingly, *AffEq+BCE* shows good performances

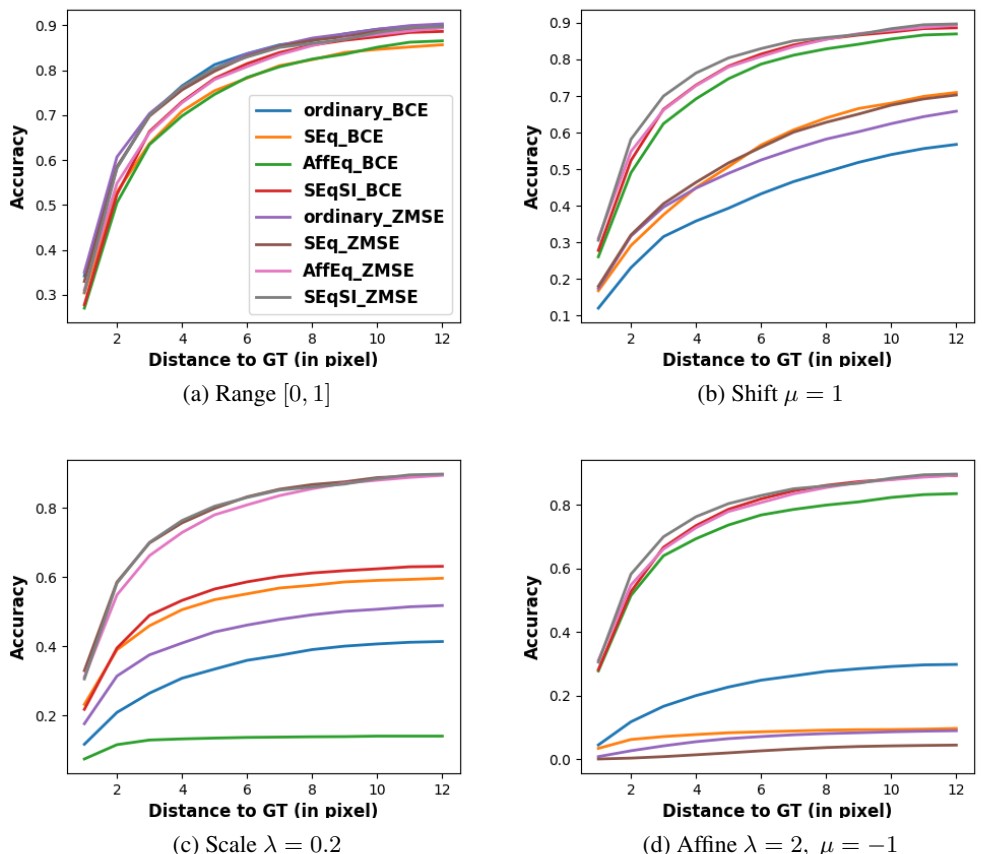

Figure 11: Accuracy curves for various range of normalization evaluated on DSB2018 test set. The range used to train the networks was $[0, 1]$.

    too, but it cannot be explained by any invariant property. The performances of the other methods drop.

- Scale ($\lambda = 0.2$), *AffEq+ZMSE*, *SEqSI+ZMSE* and *SEq+ZMSE* provide invariant results as the three networks are scale-equivariant and the thresholding strategy makes the location invariant. Other methods performances drop.

- Affine ($\lambda = 2, \mu = 1$), *AffEq+ZMSE* and *SEqSI+ZMSE* provide invariant results. *AffEq+BCE* and *SEqSI+BCE* provide non-invariant, but still good results. We can explain this by the fact that even if the thresholding strategy does not provide invariant results, the output of the networks being equivariant, the maps conserve the pixel values order. Thus, if the thresholding is not too sensitive to the chosen threshold (for maps that provide very different values for background and foreground, different threshold could provide similar results) and the network output not shift in the sigmoid tails (where it is almost 0 or 1 everywhere), the location can still be accurate even if it is not invariant. But this results could not be generalize to any affine transformation. All other methods show important accuracy drops.

On Supp. Figures 12, we present the output score maps and predicted locations for each method, on the same image, for the normalization range $[-1, 1]$. These results illustrate how the non-invariant methods score maps can be affected by a change in the input normalization range. Obviously, this perturbations of the score map significantly affects the object localization.

In general the normalization range is the same during training and inference and that experiment was mainly dedicated to empirically prove our theoretical results. In the following experiment, we

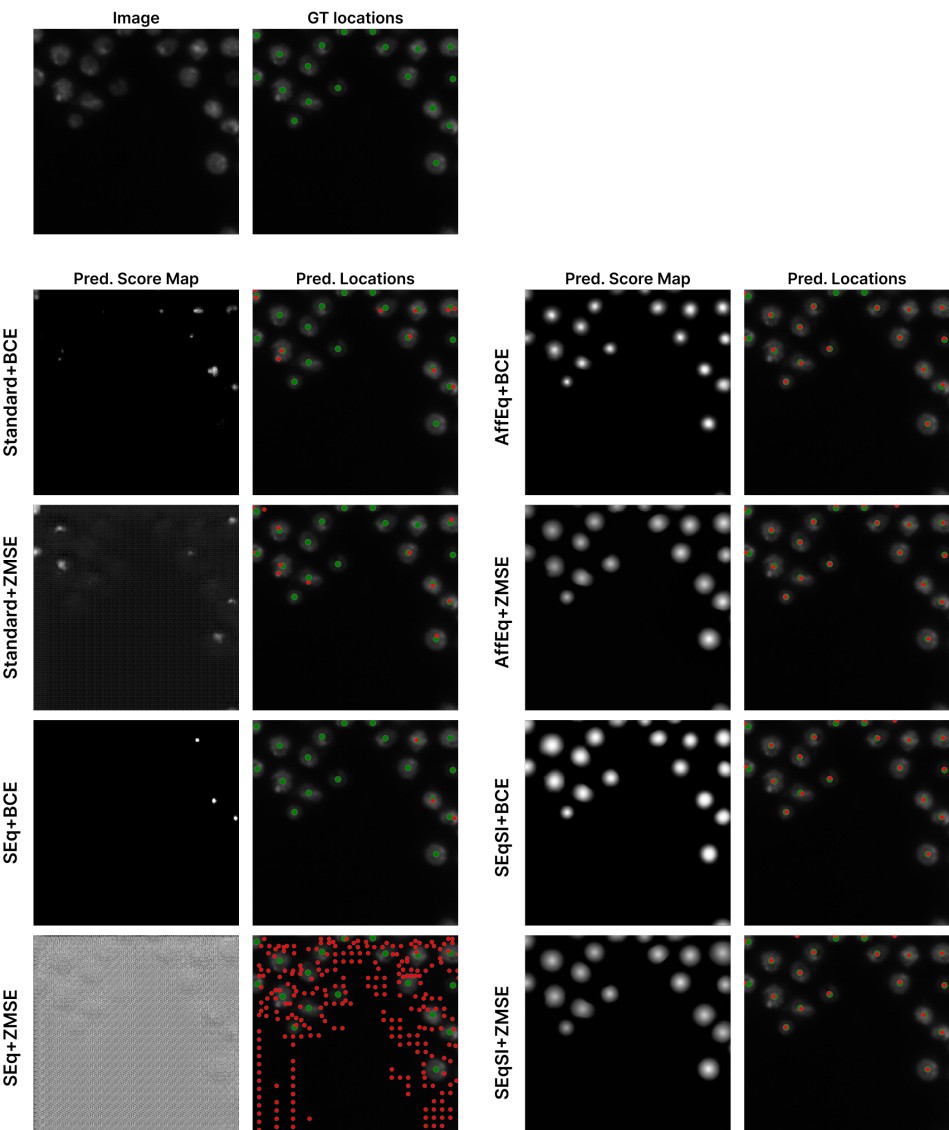

Figure 12: Score maps and object location for the different methods in the range $[-1, 1]$. The GT location are represented with the large green dots and the predicted location are represented using the smaller red dots.

evaluate if, for the same range of normalization, invariant strategies can improve the robustness over non-affine perturbation.

### E.3.2 ROBUSTNESS OF *Standard* WITH NORMALIZATION AND *SEqSI* AGAINST SPATIALLY-VARYING SHIFTS

In the Methods Section (see Section 4) we present our approach that makes a convolutional network provably invariant to a global affine photometric transformation applied to the image. Other widespread methods lead to similar results. Most of the normalizations commonly applied in image processing eliminate global affine photometric corruptions. For example, the transformation $T_1 : \boldsymbol{x} \mapsto \frac{\boldsymbol{x} - \min(\boldsymbol{x})}{\max(\boldsymbol{x}) - \min(\boldsymbol{x})}$, which normalizes the image to the range [0,1], removes the shift by subtracting $\min(\boldsymbol{x})$ and the scale by dividing by $\max(\boldsymbol{x}) - \min(\boldsymbol{x})$. Similarly, the normalization

$T_2 : \boldsymbol{x} \mapsto \frac{\boldsymbol{x} - \mathbb{E}[\boldsymbol{x}]}{\sigma(\boldsymbol{x})}$, which centers and reduces the intensity distribution of the image, also eliminates a global shift and scale.

Thus, at first glance, one might therefore think that our approach has little value if normalizing input images provides a simple solution to the same problem.

In this section, we demonstrate how our approach differs from simple normalization, particularly for images affected by affine intensity transformations that vary spatially (cases for which theoretical guarantees are weakened but, in our view, still meaningful). We describe two examples of local intensity shifts for which our approach theoretically guarantees better results than simple normalization, and we illustrate it with experimental results.

For the first example, we consider the very simple case of a two-piece shift $\boldsymbol{\mu}$: the upper half of the image undergoes no transformation, while the lower half undergoes a shift. Specifically, considering the domain of positions on the image defined as $[0, H-1] \times [0, W-1]$ for an image of size $(H, W)$, $\forall (i, j) \in [0, H-1] \times [0, W-1]$, $\mu_{i,j} = 0$ if $i \leq H/2$ and $\mu_{i,j} = \mu$ if $i > H/2$. In this configuration, applying the transformation $T_1$ to rescale the original image to [0,1] will, at best, behave adequately on one half of the image (depending on where the minimum and maximum values of the image were located before and after the transformation) and, at worst, behave poorly everywhere (meaning the normalized image values will differ from those of the original image at every location). Similar considerations apply if $T_2$ is used.

In the case of the *SEqSI* network, the shift is removed by the first layer of the network through a convolution that acts locally. Thus, the shift is correctly eliminated almost everywhere, except at the shift discontinuity. The size of the affected area after the first layer depends on the size of the convolution kernel. In our case ($3 \times 3$), only two rows of pixels are impacted. Depending on the number of subsequent convolutions performed by the network, the error will propagate to a slightly larger region, but a significant portion of the image will remain unaffected by this non-global shift. In light of this observation, we can note that our network provides guarantees of near-invariance to piecewise constant shifts.

To illustrate this observation, we present in Supp Fig. 13 the object localization results for the case where the image is corrupted with a shift of $\mu = 0.25$ applied only to the lower half of the image. We evaluate the *Standard* network from the experiment in Section E.3.1, and we apply a normalization to [0,1] (namely $T_1$) as a pre-processing to match the range used during training. We compare with the *SEqSI* network from the same experiment.

We can clearly observe that, for the *Standard* network, the score map is significantly affected across the entire lower half, even far from the junction. In contrast, the score map for *SEqSI* appears to show differences only in the immediate vicinity of the separation. Regarding position predictions, we note that *SEqSI* produces very similar position predictions between the clean image and the corrupted image. Some positions are missing, particularly because the thresholding depends on global values of the score map (mean and standard deviation), which are inevitably affected by the differences at the junction. On the other hand, *Standard* misses a large portion of the centers in the lower half, and those that are predicted are generally slightly displaced compared to the positions predicted on the clean image. This difference in behavior explains the reasonable invariance measurement for *SEqSI*, whereas that of *Standard* is very low.

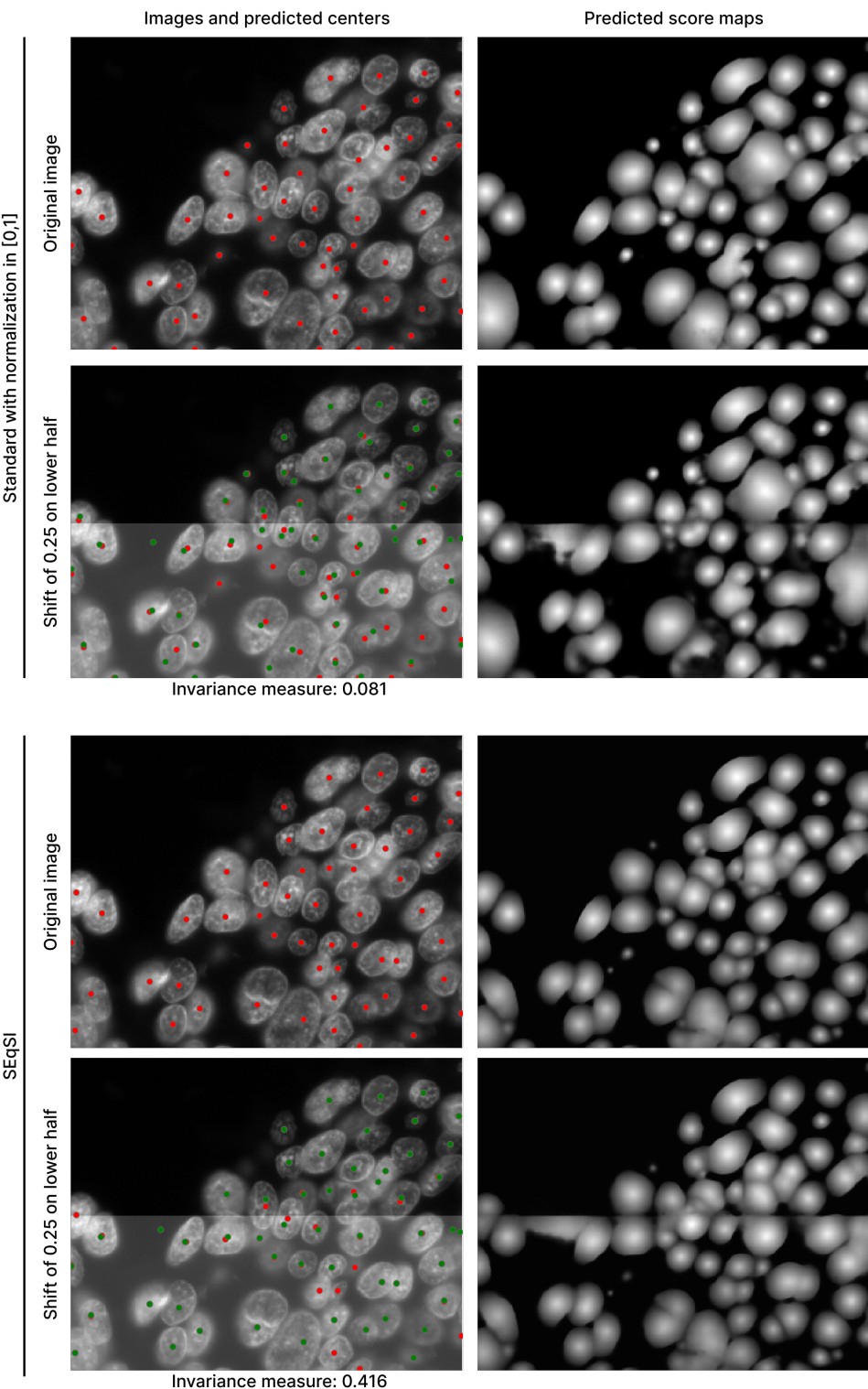

Figure 13: Object localization results for a spatially-varying shift corresponding to a shift of $\mu = 0.25$ only on the second half of the image. The red dots correspond to the positions estimated on the clean image, the green one are obtained on the corrupted one. On the corrupted image, both red and green are displayed: the non-visible red correspond to green that perfectly overlap (invariant estimation). The "invariance measure" correspond to an accuracy for $d = 1$ comparing green and red positions (the closer it is to 1, the more invariant the estimation is).

The second example we consider is that of a shift that evolves linearly according to the position on the image. Specifically, we define a shift $\boldsymbol{\mu}$ such that $\forall (i,j) \in [0, H-1] \times [0, W-1]$, $\mu_{i,j} = \kappa(\frac{i+1}{H} + \frac{j+1}{W})$, where $\kappa$ is a parameter controlling the magnitude of the shift. This kind of shift can be observed frequently in biological images, for example where the thickness of the imaged tissue is not constant.

Just as in the previous experiment, basic normalization to [0,1] will not achieve the desired behavior (removing the shift) anywhere.

For our approach, since the shift is not piecewise constant, the argument put forward for the previous experiment no longer holds. However, we can examine the form of the shift in the neighborhood of a position $(i, j)$ on the image. For $(i+1, j)$, the shift is $\mu_{i+1,j} = \mu_{i,j} + \frac{\kappa}{H}$. For any other pixel $(k, l)$ in the neighborhood, the shift can also be decomposed into the sum of a common component $\mu_{i,j}$ depending on $(i, j)$ and a residual component $\delta_{i-k,j-l}$ depending on the location in the neighborhood: $\mu_{i,j} + \delta_{k-i,l-j}$. Thus, the convolution (which sums to zero over the pixels in the neighborhood) will eliminate the common component $\mu_{i,j}$ related to the position on the image and will have an indeterminate effect on the residual part. In cases where the residual part is negligible, particularly when $\kappa$ is small compared to the dimensions of the image (or when the shift varies slowly in a more general, non synthetic setting), the proposed approach therefore appropriately removes most of the shift.

In Figure 14, we illustrate the localization results for the *Standard* network using [0,1] normalization as pre-processing, alongside those provided by our *SEqSI*, for a spatially-varying shift with $\kappa = 0.3$.

The results are even more striking for this experiment: the score map of the *Standard* network is heavily impacted, especially in the bottom-right corner where the shift is strongest. As a result, position prediction no longer works at all, and the invariance measure is almost zero. For *SEqSI*, the network predicts nearly identical results, explaining the very high invariance measure.

A more real life example of spatially-varying shifts that are handled very well by our approach is the bright-artifact case mentioned at the end of Section 7.2 and more developed in Supp. E.3.4. In that case we generated very local and high intensity shift applied in small regions of the image to generate artifacts that can be observed in microscopy images.

In light of the various points discussed in this section, it seems to us that, although our solution may appear simple at first glance, it addresses a much broader range of corruptions than most of the normalizations commonly used in deep learning tasks. Exploring its applicability across multiple applications by testing its robustness to numerous non-affine and/or non-global corruptions seems highly relevant.

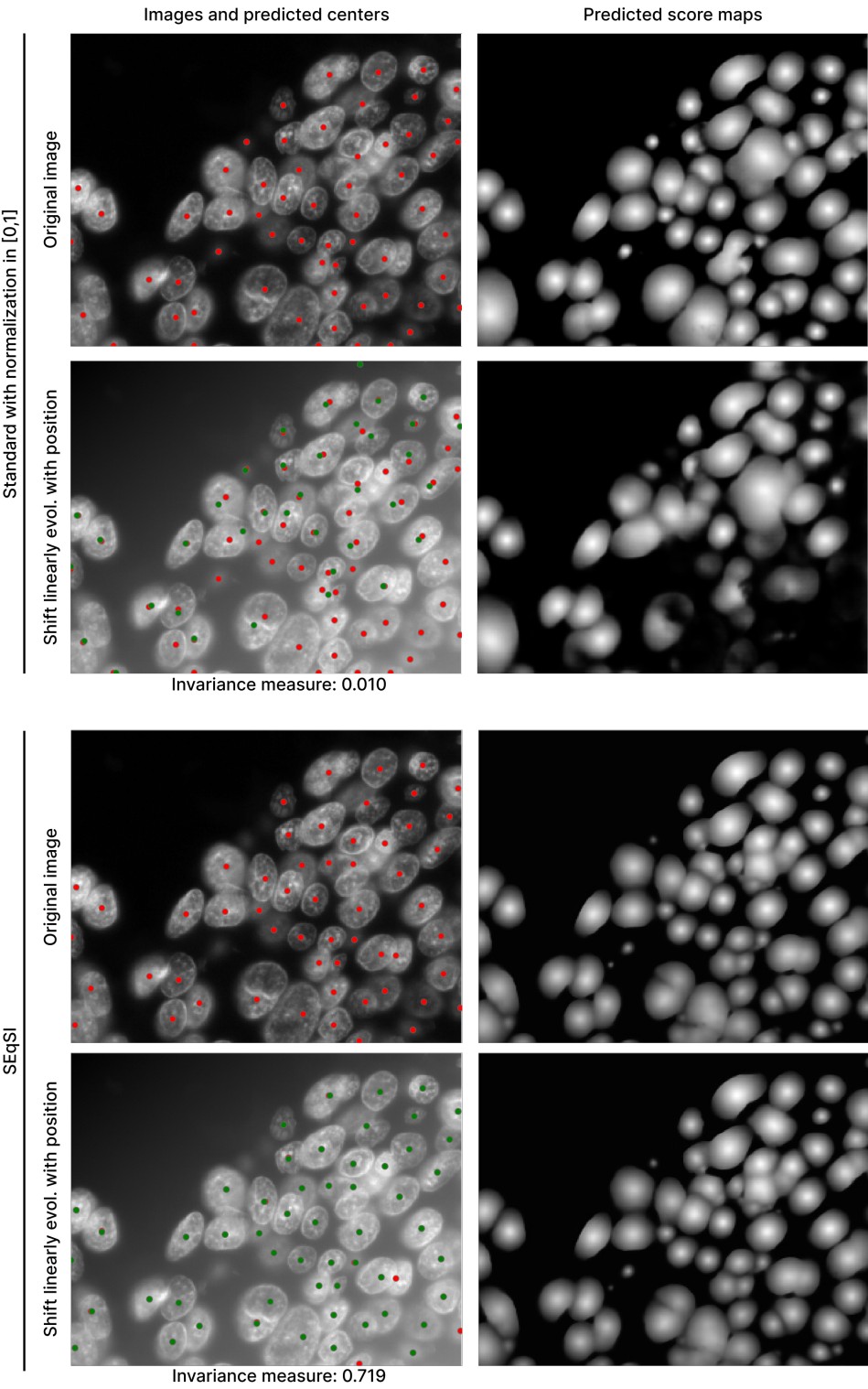

Figure 14: Object localization results for a spatially-varying shift with $\kappa = 0.3$. The red dots correspond to the positions estimated on the clean image, the green one are obtained on the corrupted one. On the corrupted image, both red and green are displayed: the non-visible red correspond to green that perfectly overlap (invariant estimation). The "invariance measure" correspond to an accuracy for $d = 1$ comparing green and red positions (the closer it is to 1, the more invariant the estimation is).

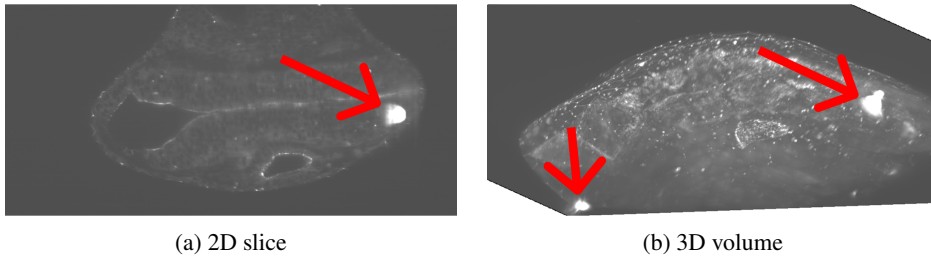

(a) 2D slice                     (b) 3D volume

Figure 15: Example of bright artifacts (red arrows) on a 3D biological image representing a zebra fish larva. The maximum contrast has been limited to allow visualization because the artifact is 250 times brighter than other pixel values.

### E.3.3 Robustness to affine and Non-Affine photometric Corruptions

This experiment was conducted using 3D U-Net models. We trained them for 6000 epochs, using batch of size 4, using AdamW optimizer with a learning rate (lr) of $10^{-4}$ and image patches of size $90 \times 90 \times 90$ voxels. Each patch is ranged in the range $[0, 1]$.

If a data augmentation strategy is applied (Aff., NAff. or All, details in Supp. Sec. C.2), the patch is corrupted by a image transformation picked in that augmentation category. The parameters of the corruption are picked in the same interval than those presented in the classification experiment.

Gradient clipping is applied to prevent exploding gradient. We observe that the *AffEq* method is particularly hard to train in 3D for this task and often creates exploding gradients. Clipping partially solved the problem but this tendency to create high gradient can explain the bad results of *AffEq*, especially when combined with data augmentations that perturbs the training even more.

For every approach, we selected the weights that minimized the validation loss during the training to avoid taking an over-fitted model.

We optimized the thresholding value used in the local maxima detection by maximizing the score on the validation set.

To assess the robustness over each perturbation category, we inferred the models on the test set with a random alteration picked for each image of the set (with parameters picked in the same interval than for training). We fixed the random seeds to guarantee that each model sees identical transformations.

For each experiment and each model, we launched 5 training using 5 different seeds ($\{1, 2, 3, 4, 5\}$) to ensure statistical significance of the results. We provide results corresponding to the mean of the 5 models score. They are provided with standard deviations in Supp. Table 23.

### E.3.4 Robustness to bright artifacts

Bright artifacts, caused by sensor saturation arise frequently on fluorescence microscopy images. They correspond to very bright zones in the image that are sometimes thousands times higher in intensity than the maximum value of the signal (see example in Supp. Fig. 15). When the image is normalized in a particular range (e.g. $[0, 1]$) without particular concern on corrupted zones, the rest of the signal is squeezed. Thus, rescaling the image in $[0, 1]$, locally correspond to a scaling compressing the signal in parts of the images with no artifact.

Therefore, we assume that even if these alterations are neither affine nor global, our invariant approach can handle them correctly, given that a convolutional network processes an image locally and that the score maps are per-voxel results. In presence of artifacts, the overall result will not be invariant as expected: i) the results will differ at locations corresponding to artifacts voxels, ii) the mean and std of the predicted output will be impacted, modifying the score map after standardization. But if the artifacts represent a small region of the image, the mean and std could vary slightly, still providing a good localization even without any data-augmentation.

To evaluate this hypothesis, we corrupt our test images with artifacts with various intensities:

Table 23: **Robustness to perturbations at inference on the 3D synthetic test set of fluorescence microscopy (mean±std over 5 models).** Test score for $D = 6$ of four architectures under various photometric corruptions. Models are grouped by the training augmentation strategy used: none (Ø), affine (Aff.), non-affine (NAff.), or all combined (All). Each row corresponds to a different perturbation applied at evaluation. For each combination of model+augmentation strategy, 5 models were trained, we give the mean score and the std computed among 5 models. Within each training strategy (column group), the best result is highlighted in gray. The overall best score for each perturbation is in **bold**.

| Model \ Corruption | Train Aug. = Ø | | | | Train Aug. = Aff | | | |
|---|---|---|---|---|---|---|---|---|
| | Stand. | SEq | SEqSI | AffEq | Stand. | SEq | SEqSI | AffEq |
| Original | 0.868±0.015 | 0.605±0.33 | **0.886±0.002** | 0.87±0.007 | 0.868±0.003 | 0.842±0.011 | 0.881±0.006 | 0.395±0.017 |
| **Affine transformations (Aff.)** | | | | | | | | |
| Shift | 0.152±0.12 | 0.117±0.135 | **0.886±0.002** | 0.87±0.007 | 0.864±0.005 | 0.837±0.016 | 0.881±0.006 | 0.395±0.017 |
| Scale ($< 1$) | 0.612±0.177 | 0.558±0.319 | **0.886±0.002** | 0.87±0.007 | 0.866±0.005 | 0.718±0.153 | 0.881±0.006 | 0.395±0.017 |
| Scale ($> 1$) | 0.491±0.142 | 0.506±0.287 | **0.886±0.002** | 0.87±0.007 | 0.864±0.005 | 0.835±0.02 | 0.881±0.006 | 0.395±0.017 |
| Affine | 0.149±0.109 | 0.093±0.067 | **0.886±0.002** | 0.87±0.007 | 0.843±0.028 | 0.799±0.066 | 0.881±0.006 | 0.395±0.017 |
| **Non-Affine transformations (NAff.)** | | | | | | | | |
| Shift saturated | 0.074±0.073 | 0.066±0.087 | 0.069±0.059 | 0.359±0.179 | 0.457±0.177 | 0.298±0.22 | 0.322±0.21 | 0.264±0.063 |
| Scale ($> 1$) saturated | 0.51±0.136 | 0.444±0.272 | 0.876±0.008 | 0.842±0.019 | 0.844±0.016 | 0.792±0.014 | 0.846±0.017 | 0.299±0.036 |
| Affine saturated | 0.124±0.069 | 0.094±0.063 | 0.113±0.068 | 0.456±0.1 | 0.558±0.063 | 0.374±0.197 | 0.398±0.179 | 0.297±0.041 |
| Noise low | 0.633±0.125 | 0.505±0.289 | **0.885±0.001** | 0.736±0.122 | 0.868±0.003 | 0.841±0.013 | 0.882±0.006 | 0.385±0.017 |
| Noise high | 0.233±0.141 | 0.045±0.05 | 0.048±0.077 | 0.238±0.127 | 0.603±0.236 | 0.42±0.186 | 0.08±0.117 | 0.102±0.052 |
| Gamma (darken) | 0.535±0.136 | 0.462±0.214 | 0.424±0.193 | 0.704±0.09 | 0.838±0.015 | 0.676±0.104 | 0.669±0.175 | 0.388±0.017 |
| Gamma (lighten) | 0.512±0.23 | 0.39±0.278 | 0.883±0.002 | 0.805±0.066 | 0.865±0.004 | 0.835±0.012 | 0.876±0.007 | 0.382±0.02 |
| **Additional experiment on artifacts** | | | | | | | | |
| Arti. low | 0.561±0.165 | 0.571±0.342 | 0.842±0.016 | 0.858±0.007 | 0.857±0.008 | 0.714±0.134 | 0.87±0.009 | 0.387±0.02 |
| Arti. medium | 0.274±0.199 | 0.53±0.342 | 0.814±0.058 | 0.845±0.007 | 0.852±0.011 | 0.48±0.358 | **0.869±0.009** | 0.376±0.023 |
| Arti. high | 0.064±0.09 | 0.081±0.066 | 0.693±0.146 | 0.733±0.046 | 0.846±0.017 | 0.293±0.293 | **0.863±0.01** | 0.291±0.054 |

| Model \ Corruption | Train Aug. = NAff | | | | Train Aug. = All | | | |
|---|---|---|---|---|---|---|---|---|
| | Stand. | SEq | SEqSI | AffEq | Stand. | SEq | SEqSI | AffEq |
| Original | 0.847±0.051 | 0.702±0.211 | 0.883±0.003 | 0.483±0.037 | 0.877±0.004 | 0.853±0.013 | 0.881±0.004 | 0.377±0.006 |
| **Affine transformations (Aff.)** | | | | | | | | |
| Shift | 0.599±0.25 | 0.289±0.168 | 0.883±0.003 | 0.483±0.037 | 0.874±0.008 | 0.836±0.022 | 0.881±0.004 | 0.377±0.006 |
| Scale ($< 1$) | 0.796±0.123 | 0.59±0.281 | 0.883±0.003 | 0.483±0.037 | 0.874±0.006 | 0.789±0.084 | 0.881±0.004 | 0.377±0.006 |
| Scale ($> 1$) | 0.835±0.067 | 0.723±0.217 | 0.883±0.003 | 0.483±0.037 | 0.875±0.005 | 0.855±0.01 | 0.881±0.004 | 0.377±0.006 |
| Affine | 0.631±0.09 | 0.24±0.133 | 0.883±0.003 | 0.483±0.037 | 0.837±0.055 | 0.803±0.076 | 0.881±0.004 | 0.377±0.006 |
| **Non-Affine transformations (NAff.)** | | | | | | | | |
| Shift saturated | 0.508±0.135 | 0.385±0.118 | 0.564±0.199 | 0.234±0.104 | **0.567±0.161** | 0.405±0.212 | 0.555±0.162 | 0.244±0.053 |
| Scale ($> 1$) saturated | 0.83±0.069 | 0.716±0.184 | **0.881±0.003** | 0.368±0.073 | 0.87±0.009 | 0.823±0.023 | 0.878±0.005 | 0.284±0.029 |
| Affine saturated | 0.596±0.094 | 0.423±0.242 | 0.607±0.1 | 0.303±0.049 | **0.639±0.058** | 0.44±0.226 | 0.61±0.097 | 0.287±0.033 |
| Noise low | 0.846±0.053 | 0.703±0.211 | 0.884±0.004 | 0.471±0.036 | 0.878±0.004 | 0.853±0.012 | 0.881±0.004 | 0.372±0.01 |
| Noise high | 0.79±0.085 | 0.62±0.244 | 0.735±0.162 | 0.074±0.036 | **0.844±0.009** | 0.763±0.056 | 0.78±0.086 | 0.08±0.07 |
| Gamma (darken) | 0.798±0.127 | 0.581±0.312 | 0.864±0.018 | 0.438±0.03 | **0.873±0.005** | 0.752±0.067 | 0.777±0.021 | 0.376±0.01 |
| Gamma (lighten) | 0.843±0.054 | 0.718±0.199 | **0.884±0.003** | 0.471±0.037 | 0.876±0.004 | 0.85±0.013 | 0.879±0.003 | 0.364±0.013 |
| **Additional experiment on artifacts** | | | | | | | | |
| Arti. low | 0.816±0.078 | 0.569±0.364 | 0.86±0.026 | 0.477±0.036 | 0.862±0.004 | 0.819±0.042 | **0.872±0.004** | 0.374±0.006 |
| Arti. medium | 0.777±0.108 | 0.521±0.358 | 0.787±0.148 | 0.472±0.034 | 0.859±0.006 | 0.789±0.064 | 0.868±0.007 | 0.364±0.01 |
| Arti. high | 0.672±0.181 | 0.343±0.318 | 0.737±0.144 | 0.426±0.015 | 0.855±0.007 | 0.618±0.128 | 0.823±0.084 | 0.273±0.059 |

- **low:** the artifacts maximum intensity is 3 times higher than the original signal maximum intensity,

- **medium:** the artifacts maximum intensity, is 5 times higher than the original signal maximum intensity

- **high:** the artifacts maximum intensity is 10 times higher than the original signal maximum intensity.

We illustrate the corruptions in Supp. Fig. 16.

We evaluate the models trained for the precedent experiment. The images are rescaled in the $[0, 1]$ range before being provided to the network to match the training range. The results are given in the last rows of Supp. Table 23 show that even with no augmentation, both *AffEq* and *SEqSI* provided extremely good results for artifacts up to 10 times brighter than the original signal while other methods score is close to 0. It shows that our approach addresses this type of corruption intrinsically. Adding data-augmentation improves even more *SEqSI* performances.

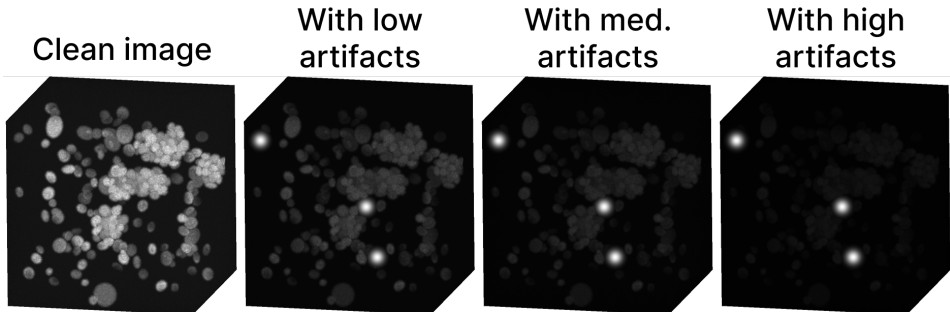

Figure 16: 3D volume from our test set of synthetic images corrupted with artifacts with various intensities: $3\times$ the maximum images intensity for low, $5\times$ for medium and $10\times$ for high, respectively.

The common strategy to address bright artifacts, based on normalization by p-value requires to know approximately the amount of pixels/voxels curated by artifacts while we propose an approach that automatically address this issue.

## F    STUDY OF THE BINARY SEGMENTATION TASK

We add a final experiment on the binary segmentation task, that can be considered as a sub-task of the object localization (as the later include a thresholding of a score map).

### F.1    DETAILS ON EXPERIMENTAL SETUP

**Evaluated Architectures.**    The architectures are rigorously the same than for the object localization task (see Supp. Sec. E.2.1 for details).

**Binary Segmentation Datasets.**    The **DSB 2018** set mentioned in Supp. Section E.2.2 provides instance segmentation masks that can be converted into binary segmentation masks, thus we decided to reuse this set for this task.

**Definition of the score.**    For this task, we compare the GT binary mask $B_{\mathrm{GT}}$ with the predicted binary mask $B_{\mathrm{pred}}$ using a Dice Score (DS).

$$DS = \frac{2|B_{\mathrm{GT}} \otimes B_{\mathrm{pred}}|}{|B_{\mathrm{GT}}| + |B_{\mathrm{pred}}|}, \tag{16}$$

where $\otimes$ is an element-wise product of the masks and $|B_{\mathrm{GT}}|$ count the number of values equal to 1 in $B_{\mathrm{GT}}$.

**Training details.**    Unlike the object localization task for which we generated dedicated score maps to train the models (with local maxima at the barycenter of each object), for binary segmentation the GT binary masks can directly be used as GT for the loss computation.

We trained the 2D U-Net models for 1500 epochs using a learning rate of $10^{-4}$, using the AdamW optimizer. The batch size is 32, the image patch are of size $128\times 128$ pixels.

Again, the thresholding value used to generate the binary masks is optimized for each model at the end of the training to maximize the DS over the validation set.

In Table 24, we report the Dice Scores on the DSB test set for each of the methods (averaged over 5 models trained for different seeds). In Fig. 17, we show examples of binary masks obtained on two images either on original or corrupted images. We observe that the shift makes the baselines either predict full foreground or full background while *AffEq* and *SEqSI* results remain unchanged.

On the quantitative results, we observe that, once again, *AffEq* and *SEqSI* are affine-invariant. For the experiment with no data-augmentation both models demonstrate better robustness than baseline

Table 24: **Robustness to perturbations at inference on the DSB dataset for binary segmentation.** The Dice score is given comparing predicted binary mask with GT one. Models are grouped by the training augmentation strategy used: none (∅), affine (Aff.), non-affine (NAff.), or all combined (All). Each row corresponds to a different perturbation applied during evaluation. Within each training strategy (column group), the best result is highlighted in gray. The overall best score for each perturbation is in **bold**. Results are averaged over 5 models trained on different seeds. The std. are given in Supp. Table 25.

| Corruption / Model | Train Aug. = ∅ | | | | Train Aug. = Aff | | | | Train Aug. = NAff | | | | Train Aug. = All | | | |
|---|---|---|---|---|---|---|---|---|---|---|---|---|---|---|---|---|
| | Stand. | SEq | SEqSI | AffEq | Stand. | SEq | SEqSI | AffEq | Stand. | SEq | SEqSI | AffEq | Stand. | SEq | SEqSI | AffEq |
| Original | **0.933** | **0.933** | 0.925 | 0.927 | 0.931 | 0.929 | 0.926 | 0.926 | 0.932 | 0.928 | 0.923 | 0.919 | 0.932 | 0.931 | 0.924 | 0.913 |
| **Affine transformations (Aff.)** | | | | | | | | | | | | | | | | |
| Shift | 0.339 | 0.289 | 0.925 | 0.927 | **0.931** | 0.929 | 0.926 | 0.926 | 0.634 | 0.442 | 0.923 | 0.919 | 0.93 | 0.929 | 0.924 | 0.913 |
| Scale (< 1) | 0.783 | 0.903 | 0.925 | 0.927 | **0.929** | 0.922 | 0.926 | 0.926 | 0.914 | 0.902 | 0.923 | 0.919 | 0.927 | 0.918 | 0.924 | 0.913 |
| Scale (> 1) | 0.917 | **0.934** | 0.925 | 0.927 | 0.931 | 0.929 | 0.926 | 0.926 | 0.893 | 0.933 | 0.923 | 0.919 | 0.932 | 0.932 | 0.924 | 0.913 |
| Affine | 0.366 | 0.353 | 0.925 | 0.927 | **0.928** | 0.917 | 0.926 | 0.926 | 0.611 | 0.449 | 0.923 | 0.919 | 0.919 | 0.922 | 0.924 | 0.913 |
| **Non-Affine transformations (NAff.)** | | | | | | | | | | | | | | | | |
| Shift saturated | 0.358 | 0.353 | 0.526 | 0.529 | 0.517 | 0.522 | 0.531 | 0.531 | **0.672** | 0.67 | 0.66 | 0.654 | 0.669 | 0.657 | 0.657 | 0.639 |
| Scale (> 1) saturated | 0.915 | 0.925 | 0.911 | 0.918 | 0.92 | 0.921 | 0.917 | 0.918 | 0.931 | **0.932** | 0.923 | 0.919 | 0.93 | 0.93 | 0.923 | 0.914 |
| Affine saturated | 0.394 | 0.386 | 0.587 | 0.587 | 0.575 | 0.578 | 0.585 | 0.588 | 0.705 | **0.708** | 0.697 | 0.691 | 0.702 | 0.695 | 0.694 | 0.68 |
| Noise low | 0.928 | 0.918 | 0.851 | 0.923 | 0.929 | 0.927 | 0.827 | 0.923 | **0.931** | 0.93 | 0.923 | 0.922 | 0.929 | 0.926 | 0.923 | 0.915 |
| Noise high | 0.6 | 0.489 | 0.146 | 0.715 | 0.708 | 0.748 | 0.15 | 0.746 | **0.894** | **0.894** | 0.884 | 0.881 | 0.878 | 0.861 | 0.873 | 0.852 |
| Gamma (darken) | 0.747 | 0.785 | 0.815 | 0.8 | 0.847 | 0.803 | 0.811 | 0.806 | 0.913 | 0.893 | 0.889 | 0.867 | **0.916** | 0.884 | 0.875 | 0.854 |
| Gamma (lighten) | 0.875 | 0.867 | 0.866 | 0.884 | 0.91 | 0.904 | 0.856 | 0.891 | 0.927 | **0.928** | 0.916 | 0.917 | 0.927 | 0.924 | 0.916 | 0.908 |

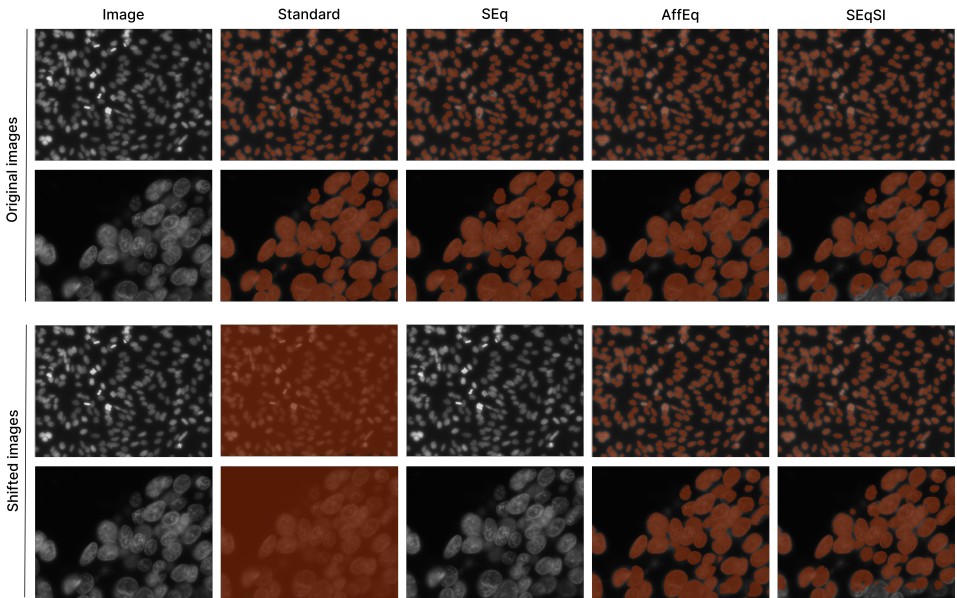

Figure 17: Binary segmentation results for different models trained without data-augmentation. The top images correspond to binary segmentation on original images. The bottom examples correspond to images corrupted with a shift.

models, even for multiple non-affine perturbations. Using data-augmentation increases the performance of all models on non-affine corruptions, with competitive and comparable results.

Table 25: **Robustness to perturbations at inference on the DSB dataset for binary segmentation (mean±std over 5 models).** The Dice score is given comparing predicted binary mask with GT one. Models are grouped by the training augmentation strategy used: none (Ø), affine (Aff.), non-affine (NAff.), or all combined (All). Each row corresponds to a different perturbation applied during evaluation. Within each training strategy (column group), the best result is highlighted in gray. The overall best score for each perturbation is in **bold**.

| Model / Corruption | Train Aug. = Ø | | | | Train Aug. = Aff | | | |
|---|---|---|---|---|---|---|---|---|
| | Stand. | SEq | SEqSI | AffEq | Stand. | SEq | SEqSI | AffEq |
| Original | **0.933±0.002** | **0.933±0.002** | 0.925±0.002 | 0.927±0.002 | 0.931±0.002 | 0.929±0.001 | 0.926±0.002 | 0.926±0.001 |
| **Affine transformations (Aff.)** | | | | | | | | |
| Shift | 0.339±0.035 | 0.289±0.096 | 0.925±0.002 | 0.927±0.002 | **0.931±0.002** | 0.929±0.002 | 0.926±0.002 | 0.926±0.001 |
| Scale (< 1) | 0.783±0.057 | 0.903±0.021 | 0.925±0.002 | 0.927±0.002 | **0.929±0.002** | 0.922±0.008 | 0.926±0.002 | 0.926±0.001 |
| Scale (> 1) | 0.917±0.006 | 0.934±0.002 | 0.925±0.002 | 0.927±0.002 | 0.931±0.002 | 0.929±0.001 | 0.926±0.002 | 0.926±0.001 |
| Affine | 0.366±0.059 | 0.353±0.091 | 0.925±0.002 | 0.927±0.002 | **0.928±0.004** | 0.917±0.015 | 0.926±0.002 | 0.926±0.001 |
| **Non-Affine transformations (NAff.)** | | | | | | | | |
| Shift saturated | 0.358±0.081 | 0.353±0.069 | 0.526±0.046 | 0.529±0.049 | 0.517±0.052 | 0.522±0.051 | 0.531±0.051 | 0.531±0.049 |
| Scale (> 1) saturated | 0.915±0.007 | 0.925±0.003 | 0.911±0.007 | 0.918±0.004 | 0.92±0.002 | 0.921±0.003 | 0.917±0.003 | 0.918±0.002 |
| Affine saturated | 0.394±0.095 | 0.386±0.04 | 0.587±0.02 | 0.587±0.029 | 0.575±0.025 | 0.578±0.028 | 0.585±0.026 | 0.588±0.028 |
| Noise low | 0.928±0.001 | 0.918±0.006 | 0.851±0.011 | 0.923±0.003 | 0.929±0.001 | 0.927±0.002 | 0.827±0.015 | 0.923±0.001 |
| Noise high | 0.6±0.125 | 0.489±0.186 | 0.146±0.006 | 0.715±0.068 | 0.708±0.042 | 0.748±0.04 | 0.15±0.008 | 0.746±0.043 |
| Gamma (darken) | 0.747±0.032 | 0.785±0.017 | 0.815±0.019 | 0.8±0.024 | 0.847±0.008 | 0.803±0.02 | 0.811±0.024 | 0.806±0.027 |
| Gamma (lighten) | 0.875±0.023 | 0.867±0.046 | 0.866±0.025 | 0.884±0.017 | 0.91±0.007 | 0.904±0.007 | 0.856±0.051 | 0.891±0.02 |

| Model / Corruption | Train Aug. = NAff | | | | Train Aug. = All | | | |
|---|---|---|---|---|---|---|---|---|
| | Stand. | SEq | SEqSI | AffEq | Stand. | SEq | SEqSI | AffEq |
| Original | 0.932±0.001 | 0.928±0.007 | 0.923±0.002 | 0.919±0.002 | 0.932±0.001 | 0.931±0.003 | 0.924±0.002 | 0.913±0.006 |
| **Affine transformations (Aff.)** | | | | | | | | |
| Shift | 0.634±0.167 | 0.442±0.049 | 0.923±0.002 | 0.919±0.002 | 0.93±0.002 | 0.929±0.004 | 0.924±0.002 | 0.913±0.006 |
| Scale (< 1) | 0.914±0.011 | 0.902±0.032 | 0.923±0.002 | 0.919±0.002 | 0.927±0.003 | 0.918±0.017 | 0.924±0.002 | 0.913±0.006 |
| Scale (> 1) | 0.893±0.02 | 0.933±0.001 | 0.923±0.002 | 0.919±0.002 | 0.932±0.001 | 0.932±0.002 | 0.924±0.002 | 0.913±0.006 |
| Affine | 0.611±0.117 | 0.449±0.072 | 0.923±0.002 | 0.919±0.002 | 0.919±0.014 | 0.922±0.011 | 0.924±0.002 | 0.913±0.006 |
| **Non-Affine transformations (NAff.)** | | | | | | | | |
| Shift saturated | **0.672±0.038** | 0.67±0.039 | 0.66±0.037 | 0.654±0.039 | 0.669±0.04 | 0.657±0.044 | 0.657±0.042 | 0.639±0.039 |
| Scale (> 1) saturated | 0.931±0.001 | **0.932±0.001** | 0.923±0.002 | 0.919±0.003 | 0.93±0.001 | 0.93±0.003 | 0.923±0.002 | 0.914±0.004 |
| Affine saturated | 0.705±0.027 | **0.708±0.022** | 0.697±0.027 | 0.691±0.03 | 0.702±0.028 | 0.695±0.028 | 0.694±0.028 | 0.68±0.026 |
| Noise low | **0.931±0.002** | 0.93±0.001 | 0.923±0.002 | 0.922±0.001 | 0.929±0.002 | 0.926±0.006 | 0.923±0.003 | 0.915±0.006 |
| Noise high | **0.894±0.006** | **0.894±0.003** | 0.884±0.005 | 0.881±0.008 | 0.878±0.018 | 0.861±0.031 | 0.873±0.011 | 0.852±0.031 |
| Gamma (darken) | 0.913±0.008 | 0.893±0.027 | 0.889±0.018 | 0.867±0.024 | **0.916±0.005** | 0.884±0.028 | 0.875±0.02 | 0.854±0.019 |
| Gamma (lighten) | 0.927±0.002 | 0.928±0.004 | 0.916±0.005 | 0.917±0.003 | 0.927±0.002 | 0.924±0.003 | 0.916±0.005 | 0.908±0.006 |

