# OpenReview forum: "Designing Affine-Invariant Neural Networks for Photometric Corruption Robustness and Generalization"
_ICLR.cc/2026/Conference — ICLR 2026 Poster_

### Official Review · Reviewer_Ssow · 2025-10-24

**Soundness:** 4
**Presentation:** 3
**Contribution:** 3
**Rating:** 6
**Confidence:** 3

**Summary:**

The paper introduces SEqSI, a convolutional network that guarantees invariance to global shifts and equivariance to global scales by design. The proposed SEqSI block ensures equivariance to scale and invariance to shifts by constraining the weights of the kernel to be zero-sum in the first block. Equivariance to scale is maintained through the rest of the network via bias-free networks. Notably, the network is significantly less constrained compared to [Herbreteau, 2023] by only requiring weight constraint at the first layer, instead of throughout the network. This yields a network that is significantly more efficient in terms of training, inference, and memory complexity while maintaining the core functionality and achieving on par or better performance. Furthermore, the paper introduces pipeline to extend the affine-equivariance to thresholding tasks such as object localization. This is achieved by replacing the conventional sigmoidal thresholding with a novel thresholding based on the standard deviation of the entire map, which is provably equivariance, as well as a novel Z-MSE loss function.

On classification tasks, SEqSI is shown to have better performance than baseline models (SEq, AffEq, etc) when no data augmentation is available and performance on par with baselines under augmentation. SEqSI also achieves strong performance to non-affine transformations that it is not specifically designed for, showcasing flexibility. These result showcases the benefits of designing for invariance and equivariance in a convolutional network.

On object localization tasks SEqSi significantly outperforms baseline methods, including a straight sweep for affine-transformation, both under and without augmentation. It achieves strong performance as well for non-affine transformations. This can be attributed to its novel thresholding scheme that preserves equivariance and invariance throughout the entire transformation.

Overall SEqSI represents an elegant solution to a problem with clear motivations and practical applications, such as biomedical imaging. My initial recommendation is a 6 (weak accept) which should be easily increased to an 8 (strong accept) with a few clarifications from the authors (please see weaknesses and questions section).

**Strengths:**

* The paper is well motivated and provides an elegant solution.
* SEqSI is provably equivariant to scale and invariant to shift. Furthermore, SEqSI achieves this with significantly less constraint than baseline line methods.
* SEqSI introduces a pipeline and loss function that extends equivariance to thresholding tasks, enabling wider application to tasks such as object localization and binary segmentation.
* SEqSI demonstrates strong performance compared to baseline methods on both classification and object localization task, both with and without data augmentation, and under affine and non-affine transformations.
* SEqSI has computational efficiency significantly better than fully affine-equivariant baselines and on par with standard models without any equivariance and only scale-equivariance models.

**Weaknesses:**

While I believe this is overall a good paper that meets all ICLR standards for acceptance, I do think there are a few weaknesses that I would like the authors to address during rebuttal.
1. The classification test in Table 3 is done on a relatively small and low resolution dataset (CIFAR-10). It would be more compelling if the authors could also present results on higher resolution (ie Stanford Cars, Oxford Pets, Caltech-101) and larger datasets (ImageNet). I don't think it is reasonable to run all experiments or the ImageNet dataset in the rebuttal time frame. If the authors can showcase results for non-augmented train set under only the four affine transformations for standard, SEqSI, and SEq for datasets such as Oxford Pets, that would convince me the scalability of the model.
2. I think it would be cool to showcase binary segmentation results on a small toy dataset such as Caltech-101, if time permits.
3. I am confused as to why Table 4 only includes non-equivariant baselines? While this demonstrates the benefits of equivariance, I don't think that was ever in question. Filling out the additional baselines would serve the paper much better.
4. I think the authors should spend more time talking about the architectural differences between SEqSI and [Herbreteau, 2023], as they appear very similar on first glance, in terms of what they want to achieve and the ways they go about achieving them.

**Questions:**

1. The authors mentioned that the proposed thresholding scheme can also be extended to tasks such as foreground-background segmentation. Would this hold for tasks requiring additional classes such as semantic segmentation? This may further extend the use case of the proposed method for situations such as autonomous driving where cameras are often subject to significantly interference with light sources such as the sun or metallic reflection.
2. While not completely related, since 2023 there have been works that deal with perceptual variation in images from the point of view of color equivariance. Specifically [1] first introduced hue-equivariance using Group CNNs [2] by rotating the RGB cube through (1,1,1). [3] achieves this using soft equivariance. [4] extends this idea to cover equivariance to hue, saturation, and luminance. I wonder if it would be worth while to talk about these in related works.

[1] Attila  Lengyel, et al. "Color Equivariant Convolutional Networks." NeurIPS 2023.\
[2] Taco Cohen, and Max Welling. "Group Equivariant Convolutional Networks." ICML 2016.\
[3] Hyunsu Kim, et al. "Variational Partial Group Convolutions for Input-Aware Partial Equivariance of Rotations and Color-Shifts." ICML 2024.\
[3] Yulong  Yang, et al. "Learning Color Equivariant Representations." ICLR 2024

---

> ### Author Response · Authors · 2025-11-21
> **Answer to Reviewer 3 (1/2)**
>
> Dear reviewer, thank you very much for your constructive feedback and recommendations to improve our paper. Below, we answer your questions and detail the additional experiments conducted to address the weaknesses.
>
> **W1** _"The classification test in Table 3 is done on a relatively small and low resolution dataset (CIFAR-10). [...]"_
>
> We agree that demonstrating performance on higher-resolution datasets is crucial, and we appreciate the suggestion to use datasets like Oxford Pets within a non-augmented training framework.
>
> Following your recommendation, we conducted a new scalability study, first on the Oxford-IIIT Pet dataset and then on the even more challenging Stanford Cars dataset to provide stronger evidence. This represents a significant leap in scale: image resolution increases from 32x32 pixels to 224x224 pixels, and the number of classes from 10 classes to 37 classes and 196 classes (for Oxford-IIIT Pet and Stanford Cars, respectively). The full results are now available in the appendix (Supp. C.8 and C.9).
>
> These new experiments confirm that our architectural priors scale effectively. On clean data and without augmentation, our models show compelling performance. For instance, on Oxford-IIIT Pet, SEqSI (44.4%) performs as good as the Standard model (44.5%). On the more complex Stanford Cars dataset, both AffEq (44.8%) and SEqSI (33.2%) significantly outperform the Standard (27.3%) and SEq (24.9%) baselines. This suggests that the constraints can act as a beneficial regularizer for generalization on such fine-grained tasks.
>
> We believe these new experiments, conducted as per your valuable recommendation, provide the compelling evidence you were looking for regarding the scalability and practical benefits of the SEqSI architecture.
>
> **W2** _"I think it would be cool to showcase binary segmentation results on a small toy dataset such as Caltech-101, if time permits."_
>
> Thank you for this valuable suggestion. Accordingly, to be able to provide an example of binary segmentation (that was also asked by Reviewer 1) within the time of the rebuttal, we decided to evaluate this task on the DSB set that is already mention in the paper for the object localization task. The instance segmentation masks are available for this set and can be converted to binary masks to evaluate if our approach can be applied on this task. We show in Supp. Section F that is it totally possible to use our thresholding paradigm in order to solve binary segmentation. Our approach offers the same formal guarantees of invariance and is competitive with other approaches, particularly in the context of no data-augmentation. A more exhaustive study of the equivariant and invariant models should be performed on other set (e.g., Caltech-101, Oxford-pet, ...) to draw general conclusions.
>
> **W3** _"I am confused as to why Table 4 only includes non-equivariant baselines? While this demonstrates the benefits of equivariance, I don't think that was ever in question. Filling out the additional baselines would serve the paper much better."_
>
> Thank you for pointing this out. Due to the density of the manuscript, we initially planned to focus the reader's attention on our primary contribution, SEqSI, as opposed to the standard baseline, in order to improve readability with regard to this specific applied Cryo-ET use case. However, in response to your feedback and that of Reviewer 1, we have now included the performance of the SEq and AffEq models in Table 2, in order to provide a more comprehensive evaluation. We now agree that including these models is essential for a comprehensive evaluation.
>
> The updated results confirm that while other architectural priors like AffEq offer robustness, SEqSI presents a more compelling overall trade-off. For instance, while AffEq excels on the CTF Deconvolved domain (79.07%), SEqSI achieves stronger generalization across the other out-of-distribution (OOD) domains, notably on Denoised (74.53% vs. 61.79%) and IsoNet Corrected (73.21% vs. 46.37%).
>
> These findings clarify that SEqSI strikes a a good compromise between performance and robustness for generalization in these challenging real-world scenarios. Thank you for pushing us to strengthen this part of our evaluation.

---

> ### Author Response · Authors · 2025-11-21
> **Answer to Reviewer 3 (2/2)**
>
> **W4** _"I think the authors should spend more time talking about the architectural differences between SEqSI and [Herbreteau, 2023], as they appear very similar on first glance, in terms of what they want to achieve and the ways they go about achieving them."_
>
>
> We thank the reviewer for this insightful question, which gives us the opportunity to clarify the key architectural differences between our proposed SEqSI model and the AffEq model from [Herbreteau, 2023]. While both models aim to achieve robustness to affine intensity transformations, they follow fundamentally different design principles with significant practical implications.
>
> The core distinction lies in the level of architectural constraint.
>
> The AffEq model enforces strict affine equivariance throughout the entire network. This requires every convolutional layer to be bias-free and have its weights constrained to sum to 1. This strong constraint also necessitates the use of specialized activation functions, "SortPool", as standard activations like ReLU would break the affine equivariance. This need for equivariance stems from the task for which this network was introduced: denoising.
>
> In contrast, for the tasks adressed in our paper, we do not recquire equivariance but invariance. Thus, our SEqSI model adopts a more pragmatic and less constrained approach to guarantee invariance if possible (for the shift) and ensure equivariance if invariance can be guaranteed (for the scale). We achieve the desired properties by strategically making only the very first layer shift-invariant (weights sum to 0, no bias). The rest of the network backbone only needs to be scale-equivariant, a much weaker constraint that simply requires the layers to be bias-free.
>
> We have insisted on these points at the end of Section 4.1 in the updated version of the paper.
>
> This architectural difference leads to two major advantages for SEqSI:
>
> Computational Efficiency: By isolating the shift-invariance property at the input, SEqSI remains fully compatible with standard, highly optimized activation functions like ReLU. This avoids the computational overhead associated with SortPool, making SEqSI significantly more efficient and easier to integrate into existing deep learning frameworks.
> Reduced Constraints: SEqSI is a far less constrained model. Only one layer has a weight sum constraint, whereas AffEq requires this for all convolutional layers. This provides the network with greater flexibility and expressive power, which, as our experiments show, translates into better performance, especially when combined with data augmentation.
> In summary, while both AffEq and SEqSI can produce a provably affine-invariant output after appropriate post-processing of the logits, SEqSI offers a more practical and efficient path to achieve this goal. It strikes a better balance between theoretical guarantees and practical utility, resulting in a robust, efficient, and high-performing architecture.
>
>
> **Q1** _"The authors mentioned that the proposed thresholding scheme can also be extended to tasks such as foreground-background segmentation. Would this hold for tasks requiring additional classes such as semantic segmentation? This may further extend the use case of the proposed method for situations such as autonomous driving where cameras are often subject to significantly interference with light sources such as the sun or metallic reflection."_
>
> The task of semantic segmentation can generally be viewed as pixel-wise classification: a softmax is applied at the end of the network to assign the most relevant class to each pixel. Therefore, all the considerations discussed in the classification section are directly applicable.
> In the case of localization, thresholding is the problematic step that requires special attention. However, semantic segmentation does not involve thresholding, making the problem simpler. As a result, our method can be used for this task.
>
>
> **Q2** _"While not completely related, since 2023 there have been works that deal with perceptual variation in images from the point of view of color equivariance. Specifically [1] first introduced hue-equivariance using Group CNNs [2] by rotating the RGB cube through (1,1,1). [3] achieves this using soft equivariance. [4] extends this idea to cover equivariance to hue, saturation, and luminance. I wonder if it would be worth while to talk about these in related works."_
>
> While these works address a topic slightly different from ours, we believe they are entirely relevant to the Related Works section, and we have therefore added references to them. Thanks for pointing it.

---

> ### Comment · Reviewer_Ssow · 2025-11-21
>
> I would like to thank the authors for the exhaustive rebuttal. The additional results presented by the authors have resolved all of my concerns and questions from my original review. I encourage the authors to add the additional experiments they were not able to run during the rebuttal period, especially binary segmentations on larger datasets which may help reader draw a more concrete conclusion. I think the authors should also consider extensions for semantic segmentation, especially for autonomous driving, where I believe the method may be of great advantage.
>
> I have decided to raise my score to 8 and my confidence to 4 in light of the additional experiments and explanation (as well as time to dwell on the paper). I believe this paper meets all acceptance criteria for ICLR and would be of value to the community. I wish the authors best of luck in their rebuttal.

---

### Official Review · Reviewer_tYPp · 2025-10-30

**Soundness:** 2
**Presentation:** 2
**Contribution:** 2
**Rating:** 2
**Confidence:** 4

**Summary:**

The paper presents a neural network architecture intended to be “scale-equivariant and shift-invariant” to photometric corruption, but the equivariance and invariance operate only on global pixel-intensity transformations rather than spatial changes. The method prepends a single convolutional layer with zero-sum weights and uses bias-free layers to enforce these properties, and then applies simple post-processing (e.g., standardization) to obtain affine-invariant predictions. The paper evaluates the approach on CIFAR-10, Cryo-ET classification, and microscopy localization, claiming improved robustness.

**Strengths:**

1. The paper is reasonably well written and clearly structured.
2. The authors conduct a large number of experiments across several tasks and datasets, providing extensive empirical evaluations.

**Weaknesses:**

1. The paper initially gives the impression that it addresses spatial affine transformations, but the invariance is only with respect to global intensity changes. Even spatial affine equivariance is already well studied and not particularly novel; restricting the scope further to global intensity scaling and shifting makes the contribution very limited. These transformations are trivial to handle with standard preprocessing or normalization. Removing biases to enforce scale equivariance, or imposing zero-sum weights in the first layer to enforce shift invariance, is not a substantial architectural idea and does not constitute a meaningful contribution.
2. Despite the many experiments, the paper does not seem to have compare against simple preprocessing baselines such as per-image normalization or mean subtraction. These standard techniques may provide the same robustness without any architectural modification. Without such comparisons, it is unclear that the proposed method offers any practical advantage. In fact, enforcing zero-sum filters in the initial layer appears effectively equivalent to preprocessing.
3. Overall, the contribution is very limited. The architectural modification is minimal and conceptually straightforward, and robustness to global intensity transforms can already be achieved with standard pipelines. The work does not demonstrate sufficient novelty or significance.

**Questions:**

Please see the above comments. My primary concern is the significance, novelty, and timeliness of the work. I would be interested to hear how the authors can clarify the architectural novelty (given that it appears to be a trivial modification) in a way that changes my assessment.

---

> ### Author Response · Authors · 2025-11-21
> **Answer to Reviewer 2 (1/3)**
>
> Dear reviewer, thank you for your feedback, which highlights the need to clarify certain aspects of our paper to better demonstrate the value of our method and contributions.
>
> **W1** _"The paper initially gives the impression that it addresses spatial affine transformations, but the invariance is only with respect to global intensity changes. Even spatial affine equivariance is already well studied and not particularly novel; restricting the scope further to global intensity scaling and shifting makes the contribution very limited."_
>
> Our paper exclusively addresses photometric affine transformations (intensity shifts and scales), not geometric affine transformations. We acknowledge the potential ambiguity of the terms "shift", "scale" and "affine" which in our context refers to additive shifting, and/or multiplicative scaling of pixel intensities. Our approach provides theoretical guarantees of full invariance in the case of global affine photometric transformations. Accordingly, we improved the manuscript to avoid confusion, by precising the terms "photometric", "intensity" and "global". However, it is of a particular significance that even if the theoretical guarantees hold only in the case of global affine intensity transformations, our approach is also robust to spatially varying affine transformations. This is what we attempted to highlight with the experiment on bright artifacts. To further emphasize this point of increased robustness to spatially varying transformations, we added several additional experiments on this topic (in Supp. D.2, Supp. E.3.2 and Supp. E.3.4, which we introduce in more detail when addressing your second point).

---

> ### Author Response · Authors · 2025-11-21
> **Answer to Reviewer 2 (2/3)**
>
> **W2** _"These transformations are trivial to handle with standard preprocessing or normalization. Removing biases to enforce scale equivariance, or imposing zero-sum weights in the first layer to enforce shift invariance, is not a substantial architectural idea and does not constitute a meaningful contribution."
> "Despite the many experiments, the paper does not seem to have compare against simple preprocessing baselines such as per-image normalization or mean subtraction. These standard techniques may provide the same robustness without any architectural modification. Without such comparisons, it is unclear that the proposed method offers any practical advantage. In fact, enforcing zero-sum filters in the initial layer appears effectively equivalent to preprocessing."_
>
>
> As you point out, simple and commonly used solutions in deep learning exist to handle global affine photometric transformations, such as intensity image normalization (e.g., scaling to [0,1] or standardizing to a zero-mean, unit-variance distribution). These approaches provide the same theoretical guarantees if applied as a preprocessing step to the images. At first glance, it might therefore seem that our approach offers little advantage compared to this simple trick. Nevertheless, we believe it is important to clarify several tricky points to demonstrate its utility.
> The most crucial point is that our approach (using zero-sum convolutional weights in the first layer) is not equivalent to the normalization techniques commonly used in deep learning because it operates locally. Its effect is indeed equivalent to normalization for global affine transformations (with full invariance in both cases). However, our approach, by removing shift using local information through convolution, performs much better for a wide range of local affine transformations, which are poorly handled by standard normalization techniques.
>
> For demonstration, we have added two examples of spatially varying intensity shifts in our paper (Fig. 4 of main text and Supp. Section E.3.2), where our approach significantly outperforms a Standard network using normalization as pre-processing to remove global affine transformation. In the first example, we show that our approach is almost invariant to piecewise constant local intensity shifts. In this case, we can guarantee near-perfect invariance everywhere except in the vicinity of the boundaries between different pieces. The second example involves a local shift that varies linearly in space. Here, standard normalization is ineffective, whereas our approach achieves near-perfect invariance. This is because, within the neighborhood of a pixel during convolution, the applied shift is almost the same for all surrounding pixels and is therefore effectively removed.
> These two basic cases (along with others, such as the artifacts already studied in the initial version of the paper Supp. E.3.4) are discussed in greater detail and supported by experiments. This (likely non-exhaustive) list of cases where normalization is less effective than our approach reinforces our belief that it is worth investigating thoroughly. This is why, throughout the paper, we conducted an extensive series of experiments on both affine and non-affine transformations, whether local or global, to test its applicability for various tasks.
>
> Finally, to demonstrate this superiority on a challenging domain shift task, we conducted an additional experiment on the real-world Cryo-ET dataset (Supp. D.2). We trained a Standard model using a min-max normalization pre-processing step and evaluated its out-of-distribution (OOD) generalization. The results are unequivocal: the performance collapses on unseen domains, proving that simple normalization is insufficient for handling real-world domain shifts. This stands in stark contrast to the performance of SEqSI, which successfully generalizes across all domains. These results reinforce our conviction in the superiority of an architectural prior like SEqSI over simple pre-processing for achieving photometric robustness.

---

> ### Author Response · Authors · 2025-11-21
> **Answer to Reviewer 2 (3/3)**
>
> **W3** _"Overall, the contribution is very limited. The architectural modification is minimal and conceptually straightforward, and robustness to global intensity transforms can already be achieved with standard pipelines. The work does not demonstrate sufficient novelty or significance."_
>
> While equivariance issues in networks have been thoroughly addressed in certain cases, such as geometric transformations (e.g., image or point cloud spatial distortions), there is still much work to be done in the context of photometric transformations. We respectfully argue this conceptual simplicity is a key strength and a significant advance over the current state-of-the-art. The AffEq model [Herbreteau, 2023], originally proposed as an equivariant solution for denoising tasks, requires complex and restrictive constraints on all layers. In this work, we adapt it to achieve invariance for the new tasks of classification and object localization. By contrast, SEqSI model is a more practical and computationally efficient solution that achieves certified invariance with far fewer constraints, making it more elegant and easier to implement. Our experiments demonstrate that this architectural prior is superior to standard normalization, especially in key scenarios where standard pipelines fail:
> - Spatially Varying Shifts: Our new experiments (Figure 4, Supp. E.3.2) show that a standard model with normalization fails, whereas SEqSI provides robust results.
> - Real-World Domain Shift: The Cryo-ET experiment (Supp. D.2) demonstrates that normalization is insufficient for out-of-distribution generalization, while the built-in prior of SEqSI is critical.
> - Bright Imaging Artifacts: Our model handles these common, local photometric corruptions by design, a scenario where global normalization is ineffective (App. E.3.4).
>
> Therefore, we believe that providing a simpler, more efficient, and more robust architectural solution to photometric variations, validated across a wide range of experiments where standard methods fail constitutes a significant contribution.
>
> Finally, we thank you for your feedback, which, in our view, has significantly improved our paper by clarifying its positioning and the technical challenges it aims to address. We provide strong theoretical guarantees in the case of global affine photometric transformations, additionnal weaker guarantees for certain spatially varying intensity affine cases, and empirical evaluation for various non-affine photometric transformations.

---

### Official Review · Reviewer_FUb7 · 2025-11-01

**Soundness:** 3
**Presentation:** 3
**Contribution:** 2
**Rating:** 6
**Confidence:** 3

**Summary:**

This is a paper which builds on the concept of scale equivariant (SEq) networks, adding shift invariance (SI). This leads to a network which is equivariant to scale operations on pixels in the image, and invariant to shifts in pixel values. In comparison to existing approaches for full affine equivariance, this approach is faster; centrally because it can leverage standard ReLU activations as opposed to the SortPool activations required for full equivariance.

The authors also propose an approach to preserve the logit outputs of their networks through various post-processing stages such as the argmax operation and thresholding.

The authors validate their approach for three tasks including classification and localisation. In all cases, the approach demonstrates strong overall performance combined with robustness both to affine and non-affine corruptions of the images.

**Strengths:**

The paper is well structured. The mathematics are clearly presented and easy to follow. The experiments are generally well motivated to demonstrate the benefits which the authors claim to achieve.

The results which are presented seem to me to be strong. The authors demonstrate the utility of their approach in a controlled setting (CIFAR-10), a macromolecule classification task and a localisation test. In all cases, the SEqSI framework outperforms performs either competitively or outperforms comparisons without degraded performance in the case of unwarped inputs compared to standard network architectures.

In particular, experiments where networks are compared with various different transforms at train time, and are evaluated under various shifts at test time, are particularly compelling for this architecture.

**Weaknesses:**

My central reservation about this paper is the magnitude of its technical contribution. In terms of the technical innovative step, it appears to me that the contribution is principally a zero-sum weight in the first layer of the neural network, and a different transformation of output logits to preserve the equivariance/invariance at output.

Nonetheless, I would support an acceptance of this paper owing to the results which support the view that the method has clear utility. My view would be strengthened if the authors were able demonstrate the effectiveness of their approach more extensively. While the authors do propose several test settings, and the results appear strong, my concerns are as follows:
- CIFAR-10 results appear good, and I would particularly credit the wide range of augmentations used within the evaluation framework. However, this dataset is relatively small and not so complicated, and so these results need to be supported by other results in the paper.
- Macromolecule classification results also appear strong, and this test setting is the most interesting in my view. However, I see that the Affine equivariant and SEq approaches are omitted here, and I cannot find an explanation for this in the paper. I feel that either the inclusion of these baselines, or provision of a reason for their exclusion is essential.
- The 3D localisation test set is also compelling, and the testing framework is strong here. However, as a synthetic dataset it would be useful to have a real-world comparison. This is presented in the form of the Data Science Bowl set (DSB). But in the main paper the DSB is only mentioned as demonstrating the invariance of the approach, whereas I feel that highlighting the robustness of the method to augmentation (Figure 10) was the most important outcome of this test for supporting the method proposed. I feel that if the DSB dataset is mentioned in the main paper, then the paper would be strengthened by including a table or chart of the main results from that dataset in the main paper (i.e. from Figure 10). Alternatively, I feel that demonstrating utility for a different real-world dataset for another task such as segmentation would strengthen the case for this method significantly.

I also found the presentation of the DSB results in the supplementary results to be confusing in some places. In particular I would highlight that the specification that $d=0$ seems to clash with the definition of TP(d) as "the number of pairs of centers *less than* d voxels apart" (emphasis added). I also found the term 'accuracy' in table 16 led me initially to think that this was showing a comparison with ground truth.

**Questions:**

1. In table 5, your SEqSI network outperforms a standard network both in the no augmentation at train/no augmentation at test, and in the affine at train / affine at test setting. Do you have a good heuristic for why this is the case? Is it because of the network's ability to efficiently learn over variations in the train set? It seems to be a curious result to me if the SEqSI approach is missing both layer biases and has a constraint on the first set of weights.

2. In figure 10, the selected values of $\mu$ and $\lambda$ are significant in comparison to the range of the original image. As I read it, the chosen value of $\mu$ would mean there was no overlap between the two distributions in the shift case. Is there a reason that such a significant perturbation was chosen?

---

> ### Author Response · Authors · 2025-11-21
> **Answer to Reviewer 1 (1/2)**
>
> Dear reviewer, thank you very much for the time you dedicated to reviewing our article and for your thoughtful and constructive feedback. We provide below our detailed responses to the weaknesses and questions you have raised.
>
> **W1** _"CIFAR-10 results appear good, and I would particularly credit [...]"_
>
> To address this relevant point, also raised by Reviewers 3, regarding the relative simplicity of the CIFAR-10 dataset, we have conducted additional experiments to demonstrate the scalability and effectiveness of our approach on more complex, higher-resolution datasets. We scaled both the image resolution (from 32x32 pixels to 224x224 pixels) and the number of classes (from 10 classes for CIFAR-10 to 37 classes for Oxford-IIIT Pet and 196 classes for Stanford Cars). A new section has been added to the appendix (Supp. C.8 and C.9, see "Demonstrating Scalability on a More Complex Task: Oxford-IIIT Pets and Stanford Cars Classification") presenting results on the Oxford-IIIT Pet and Stanford Cars datasets.
>
> These new experiments confirm that our constrained architecture preserves its capability to learn complex, fine-grained features at a larger scale. When trained without any data augmentation, our models show compelling performance on clean test data. On Oxford-IIIT Pet, SEqSI (44.4%) performs performs equivalently with the Standard model (44.5%). On the more challenging Stanford Cars dataset, both AffEq (44.8%) and SEqSI (33.2%) significantly outperform the Standard (27.3%) and SEq (24.9%) baselines. This strong performance, particularly on Stanford Cars, suggests that the architectural constraints are beneficial for generalization.
>
> We believe these additional results provide compelling evidence for the scalability and practical benefits of the SEqSI architecture, demonstrating its value for building models that are robust to photometric variations in complex, real-world scenarios.
>
> **W2** _"Macromolecule classification results also appear strong, and this test setting is the most interesting in my view. [...]"_
>
> Thank you for your valuable comment. Our initial decision to omit these models was a matter of presentation. Due to the density of the manuscript, we decided to simplify the comparison for this particular Cryo-ET application, directing the reader's attention towards our main contribution, SEqSI, as opposed to the standard baseline, in order to enhance readability. However, we agree that including these models is essential for a comprehensive evaluation.
>
> We have now addressed this concern by including the performance of the SEq and AffEq models in the CryoET macromolecule classification experiments in Table 2.
>
> The results indicate that the AffEq model also exhibits notable domain shift capabilities, even outperforming SEqSI on the CTF Deconvolved domain (79.07% vs. 66.51%). However, SEqSI appears to offer a more compelling trade-off, achieving strong generalization across all out-of-distribution (OOD) conditions while remaining competitive on in-distribution data. Specifically, SEqSI significantly outperforms AffEq on the WBP, Denoised and IsoNet Corrected datasets (85.15, 74.53% and 73.21% vs. 79.26, 61.79% and 46.37%, respectively). These findings underscore that while different architectural priors can provide robustness, SEqSI presents a more appealing balance for generalization in challenging, real-world imaging scenarios.
>
> **W3**  _"The 3D localisation test set is also compelling, and the testing framework is strong here. However, as a synthetic dataset it would be useful to have a real-world comparison. This is presented in the form of the Data Science Bowl set (DSB). [...]"_
>
> Initially, we chose to emphasize the synthetic dataset because it allowed us to highlight a 3D application case, whereas DSB focuses on 2D image analysis. As recommended, to showcase a case with real data we have performed a comparative study of the approaches for binary segmentation on the DSB set (which provides segmentation masks). This is briefly mentioned in the main paper and further developed in the appendix (see Supp. F).
> Additionally, we have incorporated into the main paper qualitative and quantitative examples obtained on DSB for the localization task (Fig. 4) to emphasize the versatility and robustness of our approach against spatially-varying intensity shift, as well as the higher performance when compared to usual normalization applied to the input image.

---

> ### Author Response · Authors · 2025-11-21
> **Answer to Reviewer 1 (2/2)**
>
> **W4** _"I also found the presentation of the DSB results in the supplementary results to be confusing in some places.[...]"_
>
> Thank you for pointing out that lack of precision. We have updated the definitions of TP(d), FP(d), and FN(d) in the revised manuscript in Supp. E.2.3. TP now accounts for matches between ground-truth and predicted centers whose distance is **strictly less than d**, while FP and FN use matches whose distances greater than or equal to d. Additionally, the term “accuracy” in the former Table 16 (now 21) was confusing. We now instead refer to the "invariance measure" computed with d=1, which is consistent with what we actually implemented in our experiments.
>
>
> **Q1** _"In table 5, your SEqSI network outperforms a standard network both in the no augmentation at train/no augmentation at test, and in the affine at train / affine at test setting. [...]"_
>
> You are absolutely right, it is surprising that a more constrained network performs better even when no corruption is applied. This scenario seems to be quite rare but not a unique case (e.g., for the Cryo-ET classification task, where SEq outperforms the baseline and Stanford Cars where AffEq outperform the baseline). Thus we advocate that these are probably task specific cases and that some architecture constraints and properties could be more suited for particular data and tasks.
>
>
> **Q2** _"In figure 10, the selected values of $\lambda$ and $\mu$ are significant in comparison to the range of the original image. [...]"_
>
> Indeed, the chosen values generate ranges that differ significantly from the training range, particularly for the shift transformation, where the range does not overlap at all with [0,1]. We made this choice because our primary goal here was to demonstrate the invariant property of our approach (and the lack of invariance in conventional approaches).

---

> > ### Comment · Reviewer_FUb7 · 2025-11-25
> > **Response to Authors**
> >
> > Thank you for these clarifying points, and for updating the results you illustrate in the paper. I have now had time to read and consider your responses, and would make the following remarks.
> >
> > **W1**: Thank you for providing further examples of your method on more complicated datasets. My remaining concern is that the classification accuracy you do report on these datasets seems low. While I appreciate you mention that you do not aim to achieve SOTA accuracy - particularly on the Stanford cars dataset, where you report **27.3%** for no training augmentation - in my view the level of these results undermines comparison with the SEqSI methodology. It seems to me that even with a ResNet-18, scores significantly higher should be attainable. For example, a cursory look suggests that a test accuracy of around 90% is possible with a ResNet-18 (https://www.kaggle.com/code/archanatrivedi/resnet18-on-stanford-car-dataset). And when the dataset was released in 2013, a score of 69.5% was achievable albeit with a different network (https://ai.stanford.edu/~jkrause/papers/fgvc13.pdf). If I have misinterpreted these results or the evaluation setup then please correct me.
> >
> > **W2**: Thank you for providing the comparisons with this approach. I feel now that this is a much more complete set of results, and highlights the relative strengths of the AffEq and SEqSI methods. I feel this is now resolved.
> >
> > **W3**: Thank you for providing this. I feel that these new results do resolve my concerns. Although the numerical results using the SEqSI methodology are not the highest across the baselines you test against, balancing this with the computational efficiency advantages SEqSI achieves over the AffEq approach does motivate a clear use case for this approach.
> >
> > **W4**: Thank you for this clarification. I feel this is now resolved.
> >
> > In light of the changes to the paper made in the rebuttal, specifically the comprehensive evaluation of this approach in several applications, I have decided to change my overall score to an 8. However, taking into consideration my remaining concern about **W1**, and about the magnitude of the overall technical contribution noted in my original review, I will keep my confidence score at a 3.
> >
> > One final suggestion I would make to the authors is to highlight further the computational advantages of this approach over the AffEq network. While this is explored in Supp. Table 6, I feel it could be highlighted more clearly and serves to offset the fact that evaluation scores are often similar between SEqSI results and AffEq results.

---

> > > ### Author Response · Authors · 2025-12-01
> > > **2nd Answer to Reviewer 1 (1/1)**
> > >
> > > **W1**  "Thank you for providing further examples of your method on more complicated datasets. My remaining concern is that the classification accuracy you do report on these datasets seems low [...]"
> > >
> > >
> > > We sincerely thank Reviewer 1 for their valuable feedback regarding the initial performance of our model on the Stanford Cars dataset. This comment lead us to deepen our analysis and strengthen the validation of our approach.
> > >
> > > The Reviewer correctly pointed out that the classification accuracy reported in our initial experiments on Stanford Cars (27.3% without augmentation) seemed low, especially when compared to results achievable with pre-trained models. We acknowledge this concern and would like to clarify that our initial experiments were intentionally conducted from scratch, without any pre-training, to isolate and evaluate the intrinsic architectural properties of SEqSI and its baselines. While this setup is common for fundamental architectural comparisons, it naturally led to lower absolute performance on complex datasets like Stanford Cars.
> > >
> > > To directly address this valid concern and demonstrate the practical applicability of our methodology in a more realistic high-performance scenario, we have conducted a new transfer learning experiment. This involved fine-tuning an ImageNet pre-trained ResNet-18, a common practice that significantly boosts performance on such datasets. The key findings from this new experiment (detailed in Supp. C.10) directly address the reviewer's point:
> > >
> > >
> > > - Achieving a Competitive Baseline Performance: By fine-tuning a standard ImageNet pre-trained ResNet-18, our Standard model now achieves a much more competitive accuracy of 85.6% on Stanford Cars. This aligns with the high-performance levels the reviewer rightly expected and provides a much stronger baseline for comparison.
> > >
> > > - Demonstrating the Practicality and Strength of SEqSI: This is where the value of SEqSI becomes clear. To adapt the pre-trained model to the SEqSI architecture, we had to perform significant structural adaptations, notably the removal of all bias parameters. Despite these initial structural adaptations, our SEqSI model achieves a highly competitive accuracy of 82.3%.
> > >
> > > - Robustness as a Key Differentiator: Crucially, SEqSI achieves this strong performance while providing certified, perfect robustness to all affine transformations. In contrast, the high-performing accuracy of Standard collapses under these same transformations (e.g., dropping from 85.6% to 60.6% under a simple shift). This demonstrates that SEqSI adds a critical layer of guaranteed robustness without a major sacrifice in performance, even in a realistic transfer learning scenario.
> > >
> > > - Highlighting Architectural Viability: Furthermore, this experiment shows that the fully-constrained AffEq model fails to converge in this transfer learning setting, highlighting its practical limitations and rigidity for this type of scenario. This underscores the superior design of SEqSI, which strikes an optimal balance between theoretical guarantees and practical applicability.
> > >
> > > In summary, this new experiment demonstrates convincingly that SEqSI is not only a theoretically sound and scalable architecture but also a highly practical one. It proves the compatibility of SEqSI with transfer learning from Standard architectures, successfully leveraging its learned representations despite the required structural adaptations, and achieving competitive performance with increased robustness to affine and non-affine perturbations. We have updated the manuscript (Supp C.10), as we believe it provides a much clearer and more compelling validation of our approach.
> > >
> > > We thank Reviewer 1 again for leading us to conduct this experiment, which has significantly strengthened the paper.

---

### Author Response · Authors · 2025-11-21
**General comment**

In response to the reviewers' valuable feedback, we have substantially revised the manuscript to clarify key aspects of our contributions. To facilitate the review process, all significant modifications have been highlighted in red. A summary of the major changes is provided below.

In the Main Text:
- Experiment 2 (Macromolecule Classification in Cryo-ET): We have included the AffEq and SEq models in our comparison to provide a more comprehensive evaluation.
- Experiment 3 (Object Localization): We have added a new Figure 4 to empirically demonstrate the superior robustness of SEqSI against spatially varying intensity shifts, a key scenario where standard intensity normalization fails.

In the Appendix:
- Supp B.4: Adding of a formal proof for the invariance of Min-Max Normalization.
- Supp. C.8 & C.9: Adding of a new scalability study on large-scale and more complex datasets (Oxford-IIIT Pets and Stanford Cars)
- Supp. D.4: Adding of a direct comparison between our architectural properties and intensity normalization pre-processing strategy to highlight the benefits for domain shift robustness.
- Supp. E.3.2: Adding of a detailed analysis of the differing behaviors between a Standard model with intensity normalization and SEqSI when facing spatially varying intensity shifts, complementing the new Figure 4.
- Supp. F: Including of a new study demonstrating the robustness of our method for binary segmentation in biological imaging.

A specific response to each reviewer is provided in the following comments.

---

### Author Response · Authors · 2025-12-02
**Summary for the new Area Chair**

Dear Area Chair,

To facilitate your final assessment, here is a concise summary of the review process for our submission.

**Initial Reviews & Positive Trajectory**: The initial reviews were highly constructive. Two reviewers (FUb7, Ssow) were supportive, highlighting the elegant solution and strong results, while suggesting we further demonstrate its scalability. A third reviewer (tYPp) questioned the significance of our work compared to standard normalization.

**Addressing All Concerns with New Experiments**: We conducted extensive new experiments to address every point raised:
- Scalability (Reviewers FUb7, Ssow): In response to Reviewers FUb7 and Ssow, we added scalability tests on more complex datasets, Oxford Pets and Stanford Cars. (Supp. C.8-9).
- Missing Baselines (Reviewers FUb7, Ssow): We completed the Cryo-ET experiments by including the AffEq and SEq models, providing a more thorough comparison (Table 2).
- Real-World Data and Task Variety (Reviewers FUb7, Ssow): We added a new study on binary segmentation using the real-world DSB dataset (Supp. F) and highlighted the localization results on this dataset in the main paper (Fig. 4).

These revisions were very well-received: **Reviewers FUb7 and Ssow both explicitly stated their concerns were resolved and updated their scores and confidence levels accordingly (details are in their official responses)**.

Reviewer FUb7 raised a final question regarding the absolute performance on complex datasets, while pointing out a Kaggle notebook using Transfer Learning. We addressed this by demonstrating the strong performance and compatibility of SEqSI with transfer learning from standard pre-trained models (Supp. C.10). The reviewer did not have the opportunity to comment on these new results.

**Demonstrating Novelty over Normalization**: To address Reviewer tYPp's main concern, we added experiments (new Fig. 4, Supp. D.4) that empirically prove a critical point: while input normalization handles global transforms, it fails on the spatially-varying corruptions that SEqSI adresses efficiently. This clarifies the novelty and significance of our architectural approach, highlting its inherent versatility compared to common practices. We regret not receiving feedbacks from this reviewer before the early end of the review period, as we are confident our response could have convinced them of the relevance of our work.

In conclusion, we would like to thank the reviewers for their constructive feedbacks, which significantly improved our paper. **We believe we have thoroughly addressed every question raised**, and the resulting manuscript presents a solution that is not only elegant and theoretically grounded but also extensively validated as a practical and robust approach for a persistent challenge in computer vision, earning strong support from the reviewers who engaged with the discussion.

We hope this summary will be helpful for your final evaluation.

---

### Meta-Review · Area_Chair_TpLC · 2026-01-03

**Summary:**

The submission introduces a simple way to achieve invariance to pointwise affine transformations (scales and shifts of pixel intensities), and demonstrates its usefulness in several applications.

Initial reviews were mostly positive with only tYPp recommending rejection. Reviewers valued the writing, well-motivated ideas and experiments, and strong results. Reviewers raised concerns about the significance of the contribution, lack of some baselines and lack of experiments on more challenging tasks.

The rebuttal answered most of the reviewers questions. My assessment is that the paper presents a valid though arguably small contribution with somewhat niche applications, but it is above the bar for ICLR so I recommend acceptance.

**Reviewer Concerns:**

The most serious concerns were raised by FUb7 and tYPp about the significance of the technical contributions. The method is indeed quite simple consisting of small modifications to existing methods, but this is also a strength since it has little computational overhead and can be made compatible with typical models, in contrast with prior work. tYPp concerns about the differences wrt simple input normalization and centering were resolved by the rebuttal -- the proposed method is local, so it is also robust to spatially-varying transformations. Ssow also raised a concern about the differences wrt AffEq, which the rebuttal resolved mentioning that it is invariant instead of equivariant, less constrained and more applicable.

Other concerns were about lack of some baselines and lack of challenging experiments. The rebuttal provided more results that were fine but not particularly impressive. Classification tasks on slightly larger datasets showed reasonable metrics in fair comparisons, but far from the current SoTA. Some of the lacking baselines ended up outperforming the submission in some tasks, though the submission was still superior in challenging out-of-distribution scenarios.

**Reviewer Scores:**

FUb7, Ssow were initially positive with a positive outlook so I believe they would increase or maintain their scores.

tYPp initially recommended rejection due to the lack of significant contributions. I believe this concern was resolved so they would increase the score to 4 or 6.

---

### Decision · Program_Chairs · 2026-01-26

Accept (Poster)